# Adaptive Experimentation When You Can't Experiment

**Yao Zhao**
University of Arizona
yaoz@arizona.edu

**Kwang-Sung Jun**
University of Arizona
kjun@cs.arizona.edu

**Tanner Fiez**
Amazon
fieztann@amazon.com

**Lalit Jain**
University of Washington
lalitj@uw.edu

## Abstract

This paper introduces the *confounded pure exploration transductive linear bandit* (CPET-LB) problem. As a motivating example, often online services cannot directly assign users to specific control or treatment experiences either for business or practical reasons. In these settings, naively comparing treatment and control groups that may result from self-selection can lead to biased estimates of underlying treatment effects. Instead, online services can employ a properly randomized encouragement that incentivizes users toward a specific treatment. Our methodology provides online services with an adaptive experimental design approach for learning the best-performing treatment for such *encouragement designs*. We consider a more general underlying model captured by a linear structural equation and formulate pure exploration linear bandits in this setting. Though pure exploration has been extensively studied in standard adaptive experimental design settings, we believe this is the first work considering a setting where noise is confounded. Elimination-style algorithms using experimental design methods in combination with a novel finite-time confidence interval on an instrumental variable style estimator are presented with sample complexity upper bounds nearly matching a minimax lower bound. Finally, experiments are conducted that demonstrate the efficacy of our approach.

## 1 Introduction

In this study, we present a methodology for adaptive experimentation in scenarios characterized by potential confounding. Online services routinely conduct thousands of A/B tests annually [23]. In most online A/B/N experimentation, meticulous user-level randomization is essential to ensure unbiased estimates of *treatment effects at the population level*, commonly known as average treatment effects (ATE). In this setting, firms are often interested in understanding the treatment with the highest average outcome if presented to all members of the population. However, in many settings firms may not be able to randomize, for example if a feature must be rolled out to all users for various business reasons [30]. In such instances, users may choose to engage with a feature or not based on potentially unobservable preferences. Thus the resulting measured outcome may be correlated with the decision to engage in a specific feature. I.e. the underlying characteristics of the user *confound* the relationship between the decision to use the feature being evaluated and the effect of the feature. Thus, naively comparing the average outcome for users who engage with a feature with those who do not suffers from a (selection) bias. This setting is captured in Figure 1.

A potential solution is for services to employ *encouragement designs* where users are presented with incentives that encourage users to engage with a specific feature [4, 7, 12]. As a concrete example, many online services have introduced membership levels with different offerings and

38th Conference on Neural Information Processing Systems (NeurIPS 2024).

prices available to all users. Given a set of membership level options, the service is interested in knowing the counterfactual of which level has the optimal outcome (e.g., total revenue) if every user chooses to join that membership level. In this setting, encouragements could be coupons or trials for corresponding membership levels. In these settings, the firm can use intent-to-treat (ITT) estimates for the treatment effect which naively compare the average outcomes between the groups given different encouragements. In practice, given an encouragement a user may not engage with the corresponding feature choosing either the control or a different feature. Hence, the resulting ITT estimate may be a diluted estimate of the ATE [3]. However, all is not lost: if the encouragement presented to a user is properly randomized, and the service guarantees that the encouragement only affects the outcome through the choice of user treatment, then the encouragement acts as an *instrumental variable*. Standard analysis from the econometrics and compliance literature show that two-stage least squares (2SLS) estimators can then be used to provide consistent estimates of treatment effects.

At the same time, firms are also increasingly utilizing adaptive experimentation techniques, often known as *pure exploration multi-armed bandits* (MAB) algorithms [26, 15], to accelerate traditional A/B/N testing. Pure exploration MAB techniques promise to deliver accurate inferences in a fraction of the time and cost as traditional methods. Similar to A/B testing, bandit methods assume users are properly randomized and can fail to learn the optimal treatment if naively used and may be sample inefficient if they fail to take the confounded structure into account.

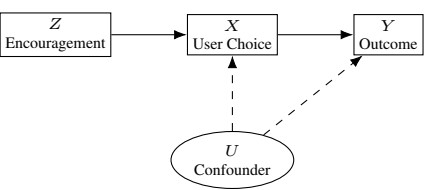

Figure 1: Causal graph of the model.

**Contributions.** In this work, we provide a methodology for experimenters seeking to use adaptive experimentation in settings with confounding where encouragements are available. We formulate this work in the more general and novel setting of *confounded pure exploration transductive linear bandits* (CPET-LB) (Section 1.1). We present algorithms using experimental design for the CPET-LB problem and analyze the resulting sample complexity. As we demonstrate, even in the simple multi-armed bandit setting described above, computing an effective sampling pattern requires using the machinery of linear bandits. Without knowledge of the underlying structural model, existing linear bandit approaches could lead to sub-optimal sampling. The main technical challenge we face is simultaneously improving our estimate of the structural model while designing with inaccurate estimates (Section 3). This approach crucially relies on our development of novel finite-time confidence bounds for two-stage least squares (2SLS) style estimators that may be of independent interest (Section 2.2). Moreover, we provide worst-case sample complexity lower bounds that are nearly matched by our sample complexity upper bounds (Appendix D). Though the goal of this work is primarily theoretical, we empirically show the efficacy of our method over existing solutions (Appendix E).

## 1.1 General Problem Formulation

A confounded pure exploration transductive linear bandits (CPET-LB) instance consists of finite collections of measurement vectors $\mathcal{Z} \subset \mathbb{R}^d$ and evaluation vectors $\mathcal{W} \subset \mathbb{R}^d$. At each time $t \in \mathbb{N}$, the learner selects $z_t \in \mathcal{Z}$ and observes a pair of noisy responses $x_t \in \mathcal{X} \subseteq \mathbb{R}^d$ and $y_t \in \mathbb{R}$ generated via the structural equation model

$$x_t = \Gamma^\top z_t + \eta_t, \quad y_t = x_t^\top \theta + \varepsilon_t, \tag{1}$$

where $\Gamma \in \mathbb{R}^{d \times d}$ and $\theta \in \mathbb{R}^d$ are model parameters.[1] We define the history $\mathcal{H}_{t-1} = \{(z_s, x_s, y_s)\}_{s<t}$ and $\mathbb{E}_{t-1}[\cdot] = \mathbb{E}[\cdot | \mathcal{H}_{t-1}]$ denoting the conditional expectation under the filtration generated by $\mathcal{H}_{t-1}$. The noise $\{\eta_t\}_{t=1}^\infty$ and $\{\varepsilon_t\}_{t=1}^\infty$ satisfy the following set of assumptions unless otherwise noted.

**Assumption 1.** We assume $\varepsilon_t | \mathcal{H}_{t-1}$ is 1-sub-Gaussian (and thus $\mathbb{E}[\varepsilon_t | \mathcal{H}_{t-1}] = 0$). Furthermore, $\eta_t | \mathcal{H}_{t-1}$ is $\sigma_\eta^2$-sub-Gaussian vectors (and thus $\mathbb{E}[\eta_t | \mathcal{H}_{t-1}] = 0$), i.e.,

$$\forall \beta \in \mathbb{R}, \max_{a: \|a\|_2 \leq 1} \mathbb{E}[\exp(\beta \langle a, \eta_t \rangle)] \leq \exp\left(\frac{\beta^2 \sigma_\eta^2}{2}\right).$$

---

[1]We assume throughout that $\Gamma \in \mathbb{R}^{d \times d}$ is an invertible matrix.

**Goal.** The objective is to identify $w^* := \arg\max_{w \in \mathcal{W}} w^\top \theta$ with probability at least $1 - \delta$ for $\delta \in (0, 1)$ while taking a minimum number of measurements.

In the setting where $\Gamma = I, \eta = 0$ and $\mathbb{E}_{t-1}[\varepsilon_t | x_t] = 0$, our setting reduces to the standard *pure exploration transductive linear bandit* problem [33, 15]. In general, the joint noise process $[\eta_t, \varepsilon_t]$ may be dependent across the entries. In particular, we are allowing for the data generating process to be *endogenous*, meaning that $\mathbb{E}_{t-1}[\varepsilon_t | x_t] \neq 0$. That is, $\varepsilon_t$ can affect not just $y_t$, but also $x_t$ given a choice of $z_t$. The presence of endogeneity is a key challenge in the CPET-LB problem.

**Assumption 2** (Exclusion Restriction). We assume that $\mathbb{E}_{t-1}[z_t \varepsilon_t] = 0$, or alternatively that $z_t$ is uncorrelated with $\varepsilon_t$.

The variable $z_t$ is commonly referred to as an *instrumental variable* [3]. We consider algorithms for the CPET-LB problem that stop at a $\mathcal{H}_t$-measurable time $\tau \in \mathbb{N}$, and produce a recommendation $\widehat{w} \in \mathcal{W}$. The goal is $\delta$-PAC algorithms with efficient sample complexity guarantees.

**Definition 1.1** ($\delta$-PAC). We say an algorithm is $\delta$-PAC for a CPET-LB problem with $\mathcal{W}, \mathcal{Z} \subset \mathbb{R}^d$ if for all $\theta \in \mathbb{R}^d$ and $\Gamma \in \mathbb{R}^{d \times d}$, it holds that $\mathbb{P}_{\theta, \Gamma}(\widehat{w} \neq w^*) \leq \delta$ for $\delta \in (0, 1)$.

### 1.2 Encouragement Designs

The CPET-LB feedback model generalizes the classical *compliance* setting.

**Compliance as a Special Case.** In *compliance* problems, a decision-maker has access to a set of treatment that can be offered to users, while the users themselves have the option to accept the treatment they are presented or instead opt-in to a different treatment. The goal is to identify the treatment with the optimal average outcome if all users were to accept it. Specifically, given a finite set $\mathcal{A} = \{1, 2, \ldots, d\}$, a decision-maker presents user $t \in \mathbb{N}$ with an encouragement for a treatment $i \in \mathcal{A}$, the user then selects treatment $j \in \mathcal{A}$ where potentially $j \neq i$, and an outcome $y_t$ results. To capture compliance with the CPET-LB framework, we set $\mathcal{Z} = \mathcal{X} = \mathcal{W} = \{e_1, \cdots, e_d\}$ and the parameter $\Gamma$ captures the probability of accepting a treatment given an encouragement for a potentially different treatment. Specifically, $\Gamma(i, j) = \mathbb{P}(x_t = e_i \mid z_t = e_j)$, and a straightforward computation shows that $x_t = \Gamma^\top z_t + \eta_t$ where $\mathbb{E}[\eta_t | z_t] = 0$ with

$$\eta_t = x_t - \left[ \mathbb{P}(e_1 \mid z_t), \cdots, \mathbb{P}(e_d \mid z_t) \right]^\top. \tag{2}$$

Moreover, $y_t = x_t^\top \theta + \varepsilon_t$ gives the resulting reward, which is clearly correlated with the user choice so that $\text{cov}(\eta_t, \varepsilon_t) \neq 0$. Finally, $e_i^\top \theta = \theta_i$ gives the expected value of treatment $i$ over the population and our goal is to identify $w^* = \arg\max_{e_i \in \mathcal{W}} e_i^\top \theta$. [2] Note that when $\Gamma = I$, we automatically have that $\eta_t = 0$ and there is no confounding. This reduces to the standard MAB setting.

**Motivating Compliance Example.** As a motivating compliance example representing the membership level discussion from the introduction, consider a location model that assumes each user $t \in \mathbb{N}$ arriving online has an underlying unobserved one-dimensional preference $u_t \sim \mathcal{N}(0, \sigma_u^2)$. If an algorithm presents the user with encouragement $z_{I_t} = e_{I_t}$ for $I_t \in \mathcal{A}$, then the user selects into the membership level given by $J_t = \min_{j \in \mathcal{A}} |I_t + u_t - j|$ so that $x_t = e_{J_t}$. This process captures a user being more likely to opt-in to membership levels that are closer to the encouragement that they were presented. The outcome is then given by $y_t = x_t^\top \theta + u_t$.

We conduct an experiment with this problem instance (see Fig. 2 and Appendix B for more details). Specifically, $d = 6, \theta = \begin{bmatrix} 1 & -0.95 & 0 & 0.45 & 0.95 & 0.99 \end{bmatrix}$, and $\sigma_u^2 = 0.35$. Observe that $w^* = e_1 = \arg\max_{w \in \mathcal{W}} w^\top \theta$. An upper confidence bound (UCB) selection strategy is simulated that maintains estimates of the mean reward of each encouragement $i \in \mathcal{A}$, namely $\widehat{\mu}_{i,t} = \sum_{s=1}^t \mathbf{1}\{z_t = e_i\} y_t$, and then pulls the one with the highest UCB. The UCB selection strategy is combined with a pair of recommendation strategies. The UCB-OLS algorithm estimates the mean reward of each treatment using an OLS estimator, namely $\widehat{\theta}_{\text{LS}}^{i,t} = \sum_{s=1}^t \mathbf{1}\{x_t = e_i\} y_t / \sum_{s=1}^t \mathbf{1}\{x_t = e_i\}$, and recommends $\arg\max_{a \in \mathcal{A}} \widehat{\theta}_{\text{LS}}^{i,t}$. Moreover, the UCB-IV algorithm uses an instrumental variable-estimator (see the next section) that incorporates knowledge of $\Gamma$ similar to 2SLS to deconfound

---

[2]Our setting differs slightly from the traditional compliance setting based on a potential outcomes framework [3]. Our setting is equivalent to one where we assume that there is a constant treatment effect. See Chapter 4 of [3] for further discussion of the differences. Thus in our setting, learning $\theta_i$ and the local average treatment effect (LATE) on compilers are the same.

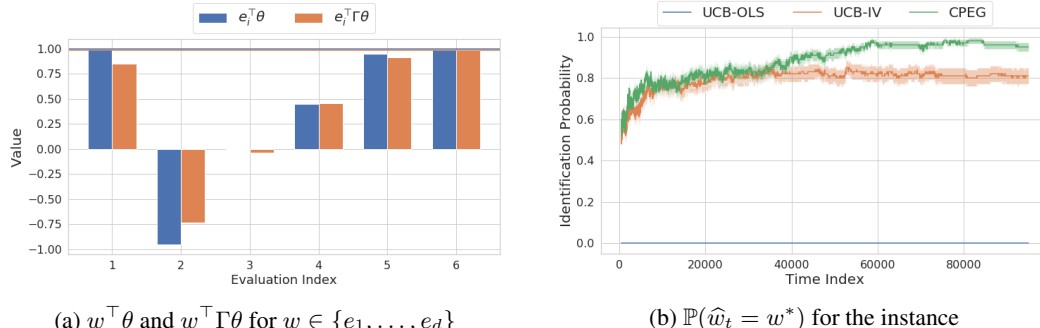

(a) $w^\top\theta$ and $w^\top\Gamma\theta$ for $w \in \{e_1, \ldots, e_d\}$   (b) $\mathbb{P}(\widehat{w}_t = w^*)$ for the instance

Figure 2: (a) A bar chart showing $\mathbb{E}[y|x = w] = w^\top\theta$ and $\mathbb{E}[y|z = w] = z^\top\Gamma\theta$ for all $w \in \mathcal{W}$. This chart shows that the optimal evaluation vector is $w^* = e_1 = \arg\max_{w \in \mathcal{W}} \mathbb{E}[y|x = w]$, while $e_6 = \arg\max_{w \in \mathcal{W}} \mathbb{E}[y|z = w]$ and consequently estimation based on this quantity is problematic. (b) The probability of identifying $w^* = e_1$ for a collection of algorithms on the CPET-LB instance from Section 1.2.

estimates of the mean rewards of each treatment and recommends the treatment with the maximum estimate. The results over 100 simulations are shown in Fig. 2. UCB-OLS completely fails to identify $\theta_1 = \arg\max_{i \in d} \theta_i$ due to a biased estimate, whereas UCB-IV does better. However, UCB-IV methods seem to have a constant probability of error. To see why, note that the expected reward from pulling $z = e_i$ is $e_i^\top\Gamma\theta$. These values are plotted in orange in Figure 2a. In particular, with some constant probability, UCB zeroes in on arm 6 becuase of the mean estimates on the $z$'s, and as a result fails to give enough samples to learn that arm 1 is indeed the best. In contrast, our proposed method CPEG, Algorithm 1 manages to find the best arm with significantly higher probability.

**Notation.** Let $\Delta(\mathcal{Z}) = \{\lambda \in \mathbb{R}^{|\mathcal{Z}|} : \lambda \geq 0, \sum_{z \in \mathcal{Z}} \lambda_z = 1\}$ denote the set of probability distributions over the set $\mathcal{Z}$. Given a distribution $\lambda \in \Delta(\mathcal{Z})$ and matrix $\Gamma \in \mathbb{R}^{d \times d}$, define the operator $A(\lambda, \Gamma) := \sum_{z \in \mathcal{Z}} \lambda_z \Gamma^\top z z^\top \Gamma$. Given $Z \in \mathbb{R}^{T \times d}$ and $\Gamma \in \mathbb{R}^{d \times d}$, define the operator $\bar{A}(Z, \Gamma) := \sum_{t=1}^{T} \Gamma^\top z_t z_t^\top \Gamma = \Gamma^\top Z^\top Z\Gamma$ where $z_t \in \mathbb{R}^d$ denotes row $t$ of $Z$. Given a vector $x \in \mathbb{R}^d$ and a symmetric positive-definite matrix $A \in \mathbb{R}^{d \times d}$ we let $\|x\|_A^2 = x^\top A x$. We adopt the standard notation that $(a \vee b) \equiv \max\{a, b\}$ and $(a \wedge b) \equiv \min\{a, b\}$ for $a, b \in \mathbb{R}$. $\sigma_{\min}(A), \sigma_{\max}(A)$ denote the minimum and maximum singular value of a matrix $A$. We denote by $\text{polylog}(x_1, \ldots, x_n)$ any polylogarithmic factors of $x_1, \ldots, x_n$.

## 1.3   Related Works

Our work is at the intersection of several parallel tracks of literature, pure exploration linear bandits, causal bandits, and econometrics. The most relevant work on pure exploration in linear bandits is the RAGE algorithm of [15, 33]. RAGE is nearly instance optimal for linear bandits in the non-confounded setting. Extensions of RAGE to various noise models including logistic and heteroskedastic noise have been considered [35, 20]. Other algorithms for pure exploration linear bandits have been proposed - and we leave it for future work to extend the ideas of this paper to those settings [27, 10].

Confounding in bandits was first considered in the regret minimization setting by [5]. They introduces the Multi-armed bandit with unobserved confounders (MABUC) problem. They empirically demonstrate traditional bandit algorithms can have linear regret in this setting and provide an algorithm that effectively employs observed intuition. The early work of [21] also assumes there is an additional unobserved latent class at each time that determines confounding in a compliance setting. They provide novel notions of regret, relative to the instrument with the highest reward ($\arg\max Z^\top\Gamma^\top\theta$ in our notation), the highest treatment ($\arg\max_w w^\top\theta$), regret relative to the best latent class at each time, and regret on the set of "compliers". They discuss the suitability of these various notions of regret, and discuss when sublinear regret is possible. We remark that their approach is similar to ours in the sense that they assume a form of homogeneous effects across the population, and use an estimate of $\Gamma$. Recently [24] also consider the problem of compliance, however they don't take explicit non-confounding into account and assume an explicit parametric model that determines the non-compliance. This is analogous to the Heckman selection model considered in econometrics [18].

The recent works of [11, 36, 17] considered an online setting where at each time they observe a set $\{(x_t, z_t)\}$ where $x_t$ is the action of interest and $z_t$ is an associated instrument. If action $I_t$ is selected, the reward observed is $y_t = x_{I_t}^\top \theta_t + \varepsilon_t$, where $x_t$ may be endogenous. Similar to the standard linear bandit setting [1, 26], the goal is to minimize regret relative to the best action at each time. We remark that this setting is very different from ours. Effectively, we are choosing which instrument to select at each time to learn the best-performing treatment - in particular we can't choose a particular intervention. In their setting, they are choosing an intervention at each time and using the instrument purely for de-confounding the result. Experimental design for instruments to have more effective estimation has been considered by [8].

In the causal bandit problem, an underlying causal graph between a set of interventions and a reward value is assumed. Actions correspond to intervening (i.e. a "do" operation [31]) at one or more specific nodes in the causal graph and then observing the corresponding value at the reward node. Causal bandits have been studied extensively in the regret setting [25, 28, 6] and the pure exploration setting [32]. Though past works have allowed for unobserved confounders in the graph e.g. [29], their goal is to learn the best performing intervention, which in our setting would be $\arg\max_{z \in \mathcal{Z}} z^\top \Gamma \theta$ instead of $w^*$.

Encouragement designs have been considered in many applications in online and offline settings. One of the earliest works on encouragement designs is [7], which considers the problem of using encouragements to determine the impact of coupons at a grocery store. More recent applications include [4, 30, 13] all in the context of online services and treatments that are required to be served to all users. Most of these works consider a small number of treatments and a heterogeneous treatment effect - hence are interested in LATE estimator. As far as we are aware, we are the only work that considers adaptive encouragement design in the context of the model given in Equation 1 and for multiple treatments.

## 2 Estimators and Inference

We now present estimators for the unknown parameter $\theta$ and prove the associated statistical properties. The estimators discussed in this section are critical to our algorithmic solution outlined in Section 3.

### 2.1 Estimators

Before describing our solution concept, we quickly review potential options for estimating $\theta$ based on a dataset $Z_T = [z_1, \cdots, z_T]^\top \in \mathbb{R}^{T \times d}, X_T = [x_1, \cdots, x_T]^\top \in \mathbb{R}^{T \times d}, Y_T = [y_1, \cdots, y_T]^\top \in \mathbb{R}^T$, assumed to be generated according to the model in Eq. (1). Recall that the *ordinary-least-squares* (OLS) estimator for $\theta$ is given by

$$\widehat{\theta}_{\texttt{LS}} := \arg\min_{\widehat{\theta} \in \mathbb{R}^d} \sum_{t=1}^T (y_t - x_t^\top \widehat{\theta})^2 = (X_T^\top X_T)^{-1} X_T^\top Y_T = \theta + (X_T^\top X_T)^{-1} X_T^\top \varepsilon_T. \quad (3)$$

Observe that $\widehat{\theta}_{\texttt{LS}}$ is potentially a biased and inconsistent estimator for $\theta$ in the presence of endogenous noise since $\mathbb{E}[\varepsilon_t | x_t] \neq 0$. To remediate this problem, we define a general class of estimators that includes several standard estimators. Given an invertible matrix $\Psi \in \mathbb{R}^{d \times d}$, let $\bar{X}_T := Z_T \Psi$, and consider corresponding estimators termed $\Psi$-IV estimators of the form

$$\widehat{\theta}_\Psi := (\bar{X}_T^\top \bar{X}_T)^{-1} \bar{X}_T^\top Y_T = (\Psi^\top Z_T^\top Z_T \Psi)^{-1} \Psi^\top Z_T^\top Y_T = (Z_T^\top Z_T \Psi)^{-1} Z_T^\top Y_T. \quad (4)$$

When $\Psi = I$ we recover the OLS estimator. In the rest of the paper, we will focus on two different potential options for $\Psi$.

**Case 1: Oracle.** $\Psi = \Gamma$. To begin, observe that the structural equation model from Eq. (1) can be combined by substituting the second equation into the first to obtain the reduced form

$$y_t = z_t^\top \Gamma \theta + \eta_t^\top \theta + \varepsilon_t. \quad (5)$$

Since $z_t$ is independent of the i.i.d. process $\eta_t^\top \theta + \varepsilon_t$, the least squares estimator which regresses $y_t$ onto $z_t^\top \Gamma$ is unbiased for estimation of $\theta$ and given by

$$\widehat{\theta}_{\texttt{oracle}} = (\bar{X}_T^\top \bar{X}_T)^{-1} \bar{X}_T^\top Y_T = (Z_T^\top Z_T \Gamma)^{-1} Z_T^\top Y_T. \quad (6)$$

This estimator will be used to design our general solution concept presented in Section 3. Of course in practice we cannot expect to know $\Gamma$, but we may be able to estimate it.

**Case 2: P-2SLS.** $\Psi = \widehat{\Gamma}$. We consider a setting where $\widehat{\Gamma}$ is an (unbiased) estimator of $\Gamma$, learned using least-squares from an *independent* dataset $Z_{T_1} = [z_1', \cdots, z_{T_1}'], X_{T_1} = [z_1', \cdots, z_{T_1}']$ collected non-adaptively.[3] That is, $\widehat{\Gamma} = (Z_{T_1}^\top Z_{T_1})^{-1} Z_{T_1}^\top X_{T_1}$ and:

$$\widehat{\theta}_{\text{P-2SLS}} = (\widehat{\Gamma}^\top Z_T^\top Z_T \widehat{\Gamma})^{-1} \widehat{\Gamma}^\top Z_T^\top Y_T.$$

We refer to the resulting estimator as a pseudo two stage least squares (P-2SLS) estimator. The main advantage of the P-2SLS estimator over standard 2SLS (given in Appendix C) is easier inference since now $\{\varepsilon_t\}_{t \leq T}$ of our dataset is independent of the measurements of the first dataset $Z_{T_1}, X_{T_1}$. In the econometrics literature, such an estimator is referred to as a *two-sample 2SLS estimator* [19].

## 2.2 Confidence Intervals

In the section that follows, we develop a general algorithmic approach that relies on *experimental design* aimed at reducing the uncertainty in our estimates of the optimal treatment. To this end, we first develop finite-time confidence intervals for estimators presented in the previous section given data generated according to the model in Eq. (1) and collected from non-adaptive designs.

We begin by characterizing the properties of the noise structure in the combined model of Eq. (5) with the following set of results.

**Lemma 2.1.** *Under Assumption 1, the noise process $\nu := \eta^\top \theta + \varepsilon$ is $\sigma_\nu^2$-sub-Gaussian where $\sigma_\nu^2 = 2(\sigma_\eta^2 \|\theta\|_2^2 + 1)$, specifically when the instance is compliance, $\sigma_\nu^2 = 2(4\|\theta\|_2^2 + 1)$.*

**Oracle Confidence Interval.** As in the last section, we assume that we have access to a dataset $(Z_T, X_T, Y_T)$ generated according to Eq. 1 and collected non-adaptively. Given Lemma 2.1, it can be shown that $w^\top \widehat{\theta}_{\text{oracle}}$ is a sub-Gaussian random variable satisfying the following.

**Lemma 2.2.** *With probability at least $1 - \delta$ for $\delta \in (0, 1)$ and $w \in \mathbb{R}^d$,*

$$|w^\top (\widehat{\theta}_{\text{oracle}} - \theta)| \leq \sqrt{2\sigma_\nu^2 \|w\|_{\bar{A}(Z_T, \Gamma)^{-1}}^2 \log(2/\delta)},$$

*where $\sigma_\nu^2$ is the sub-Gaussian parameter of the noise $\nu := \eta^\top \theta + \varepsilon$ characterized in Lemma 2.1.*

The proof of this result is in Appendix G.2.

**P-2SLS Confidence Interval.** We now present a novel finite-time confidence interval for the P-2SLS estimator. As discussed in the previous section with respect to this estimator, we assume access a set of data $(Z_{T_1}, X_{T_1})$ generated according to Eq. (1) and collected non-adaptively for the purpose of estimating $\Gamma$. Moreover, assume access to a separate set of data $(Z_{T_2}, X_{T_2}, Y_{T_2})$ generated according to Eq. (1) and collected non-adaptively for the purpose of estimating $\theta$.

**Theorem 2.3.** *Suppose that $\widehat{\Gamma} = (Z_{T_1}^\top Z_{T_1})^{-1} Z_{T_1}^\top X_{T_1}$ and $\widehat{\theta}_{P\text{-}2SLS} = (\widehat{\Gamma}^\top Z_{T_2}^\top Z_{T_2} \widehat{\Gamma})^{-1} \widehat{\Gamma}^\top Z_{T_2}^\top Y_{T_2}$. Then, for any $w \in \mathcal{W}$, with probability at least $1 - \delta$ for $\delta \in (0, 1)$,*

$$|w^\top (\widehat{\theta}_{P\text{-}2SLS} - \theta)| \leq \|w\|_{\bar{A}(Z_{T_2}, \widehat{\Gamma})^{-1}} \sqrt{2\sigma_\nu^2 \log\left(\frac{4}{\delta}\right)} + \|w\|_{\bar{A}(Z_{T_1}, \widehat{\Gamma})^{-1}} \|\theta\|_2 \sqrt{\sigma_\eta^2 \overline{\log}(Z_{T_1}, \delta/4)},$$

*where*

$$\overline{\log}(Z_T, \delta) := 8d \ln\left(1 + \frac{2TL_z^2}{d(2 \wedge \sigma_{\min}(Z_{T_1}^\top Z_{T_1}))}\right) + 16 \ln\left(\frac{2 \cdot 6^d}{\delta} \cdot \log_2^2\left(\frac{4}{2 \wedge \sigma_{\min}(Z_{T_1}^\top Z_{T_1})}\right)\right).$$

The proof is presented in Appendix G.3. Observe that the first term in the P-2SLS estimator confidence interval given by $\sqrt{2\sigma_\nu^2 \|w\|_{\bar{A}(Z_{T_2}, \widehat{\Gamma})^{-1}}^2 \log(4/\delta)}$ matches the Oracle estimator confidence interval in Lemma 2.2 when $\widehat{\Gamma} = \Gamma$. The second term scaling like $\mathcal{O}(\|w\|_{\bar{A}(Z_{T_2}, \widehat{\Gamma})^{-1}} \|\theta\|_2 \sigma_\eta \sqrt{d + \log(1/\delta)})$, is an upper bound on the approximation error $w^\top (\widehat{\Gamma}^{-1} \Gamma - I)\theta$ for any $w \in \mathbb{R}^d$, assuming that $\widehat{\Gamma}$ is learned from an OLS estimator (see Theorem G.3 for details).

---

[3]Formally, we say that a set of data $(Z_T, X_T, Y_T)$ generated via the model in Eq. (1) is collected non-adaptively from an experimental design if $z_t$ is $\mathcal{H}_0$ measurable for all $1 \leq t \leq T$.

We will see that the form of this confidence interval is particularly convenient for our algorithmic approach given in Section 3. In particular, the form of the variance $\|w\|^2_{A(Z_{T_2}, \Gamma)^{-1}}$ on the first term only depends on a design over instruments. Thus, we can choose an experimental design over $Z$'s which reduces this variance optimally.

*Remark* 2.4. In practice we expect the first stage of samples, $(Z_{T_1}, X_{T_1})$ to be collected from either a burn-in period or from existing historical data. We remark that assuming two stages of samples is common in the orthogonal and double machine learning for estimating nuisance parameters in the data generating process (e.g. $\Gamma$) [9]. Our result matches the existing literature on the asymptotic variance of two sample 2SLS estimators (e.g., Theorem 1 of [19]).

*Remark* 2.5. The asymptotic variance of standard 2SLS is known to involve a factor $\sigma_\varepsilon^2$, instead of $\sigma_\nu^2$ as we have [18]. Recent work by [11] shows a variance involving $d\sigma_\varepsilon^2$. However, it's unclear how to use the form of their confidence interval directly for experimental design. In addition, their work is not sufficiently general to handle the general forms of noise that we consider in Lemma 2.1.

## 3 Adaptive Experimental Design Algorithms

We now present adaptive experimental design algorithms for the CPET-LB problem. Our main insight utilizes Eq. 1 by plugging the model for $x$ into the top equation resulting in the relationship

$$y = z^\top \Gamma \theta + \theta^\top \eta + \varepsilon.$$

When $\Gamma$ is known, by Eq. 5, we see that CPET-LB reduces to a standard pure exploration transductive linear bandit problem where the measurement set is given by $\{\Gamma^\top z\}_{z \in \mathcal{Z}} \subset \mathbb{R}^d$, the evaluation set is $\mathcal{W} \subset \mathbb{R}^d$, and the feedback model is given by $y = v^\top \theta + \nu$ where the noise $\nu = \theta^\top \eta + \varepsilon$ is sub-Gaussian and as before the goal is to identify $\arg\max_{w \in \mathcal{W}} w^\top \theta$. An existing approach to this problem is given by the RAGE algorithm [15], which we use as the basis of our approach. Addressing the case of unknown $\Gamma$ is our major algorithmic contribution, where we develop solutions to improve our estimate of $\Gamma$ and learn $w^*$ simultaneously. As a warm-up to this approach, we first consider the setting when $\Gamma$ is known.

### 3.1 Warm-Up: Known Structural Model

Algorithm 1 assumes a parameter $L_\nu$, which acts as an upper bound on the sub-Gaussian constant of the noise $\nu = \theta^\top \eta + \varepsilon$. In each round $k$, an active set of potentially optimal vectors $\widehat{\mathcal{W}}_k \subset \mathcal{W}$ is maintained. CPEG aims to sample in such a way that reduces the uncertainty of the estimates on the *gaps* $(w - w')^\top \theta$ for each pair $w, w' \in \widehat{\mathcal{W}}_k$ maximally each round. In any given round the algorithm takes $N_k$ samples $Z_{N_k}$, the confidence interval of Lemma 2.2 shows that the error in estimating $(w - w')^\top \theta$ scales with $\|w - w'\|^2_{(\Gamma^\top Z_{N_k}^\top Z_{N_k} \Gamma)^{-1}}$. This motivates utilizing an *experimental design* approach where we choose a distribution $\lambda_k \in \Delta(\mathcal{Z})$ to minimize $\max_{w, w' \in \widehat{\mathcal{W}}_k} \|w - w'\|_{(\sum_{z \in \mathcal{Z}} \lambda_z \Gamma^\top z z^\top \Gamma)^{-1}}$. The number of resulting samples taken from this design $N_k$ is chosen to guarantee that the confidence interval of Lemma 2.2 is less than $2^{-k}$. Then, the elimination step in Line 8 guarantees that all $w \in \mathcal{W}$ such that $(w^* - w)^\top \theta > 2 \cdot 2^{-k}$ are then eliminated from the active set by round $k + 1$ of the procedure. To actually choose our samples, as is common in this literature [15], we use an efficient rounding procedure, ROUND that requires a minimum number of samples $r(\omega)$.

**Sample Complexity Guarantee.** The sample complexity of Algorithm 1 depends on the following problem-dependent quantity $\rho^*(\gamma)$ that captures the underlying hardness of a problem instance in terms of $(\mathcal{W}, \mathcal{Z}, \Gamma, \theta)$, when $\gamma = 0$, we abbreviate $\rho^*(0) = \rho^*$,

$$\rho^*(\gamma) = \min_{\lambda \in \Delta(\mathcal{Z})} \max_{w \in \mathcal{W} \setminus \{w^*\}} \frac{\|w^* - w\|^2_{(\sum_{z \in \mathcal{Z}} \lambda_z \Gamma^\top z z^\top \Gamma)^{-1}}}{\langle w^* - w, \theta \rangle^2 \vee \gamma^2}. \tag{7}$$

**Theorem 3.1.** *Algorithm 1 is $\delta$-PAC and terminates in at most $c(1 + \omega)L_\nu \rho^* \log(1/\delta) + cr(\omega)$ samples, where $c$ hides logarithmic factors of $\Delta := \min_w \langle w^* - w, \theta \rangle$ and $|\mathcal{W}|$, as well as constants.*

The proof of this result is in Appendix H.1. In the unconfounded case when $\Gamma = I$ and $\eta = 0, \mathbb{E}_{t-1}[\varepsilon_t|x_t] = 0$ this matches the sample complexity of [15]. In particular, for the case where $\mathcal{Z} = \mathcal{X} = \mathcal{W}$, the problem further reduces to a standard multi-armed bandit, and if $\varepsilon$ is 1-sub-Gaussian noise, [33] shows that $\rho^* = O(\sum_{i=2}^d (\theta_1 - \theta_i)^{-2}))$, which is the optimal sample complexity

of best-arm identification for multi-armed bandits. The following lemma shows that the conditioning of $\Gamma$ can have a strong impact on the resulting sample complexity.

**Lemma 3.2.** *For the compliance setting, we have* $\min_{\lambda \in \Delta^d} \max_{j,j'} \|e_j - e_{j'}\|^2_{(\sum_{i=1}^d \lambda_i \Gamma^\top e_i e_i^\top \Gamma)^{-1}} \leq$ $d \max_{j,j'} \|\Gamma^{-1}(e_j - e_{j'})\|^2_2$. *Furthermore,* $\rho^* \leq \frac{d\sigma^2_{\min}(\Gamma)^{-1}}{\Delta^2_{\min}}$.

To further illustrate the impact of $\Gamma$, imagine an extreme setting where $\Gamma = (1-\varepsilon)/d\mathbf{1}\mathbf{1}^\top + \varepsilon I$ and $\varepsilon \approx 0$, i.e. $\Gamma$ is a perturbation of $1/d\mathbf{1}\mathbf{1}^\top$. It's straightforward to show that the upper bound in the first display of Lemma 3.2 is of the order $O(d\varepsilon^{-2})$ (this is also a lower bound - see Appendix K.1). In particular, the upper bound on the sample complexity is of the form $d\varepsilon^{-2}/\Delta^2_{\min}$. This is in sharp contrast to the linear bandit case, when $\Gamma = I$ and we are guaranteed a sample complexity of no more than $d/\Delta^2_{\min}$ samples. To gain some intuition, regardless of the choice of $\lambda$, $\sum_{i \leq d} \lambda_i \Gamma^\top e_i e_i^\top \Gamma \approx \Gamma$. As a result,

---

**Algorithm 1** CPEG:Confounded pure exploration with $\Gamma$

1: **Input** $\mathcal{Z}, \mathcal{W}, \Psi = \Gamma, \delta, L_\nu \geq \sigma^2_\nu, \omega,$
2: **Initialize:** $k = 1, \mathcal{W}_1 = \mathcal{W}, \zeta_1 = 1$
3: **Set** $f(w, w', \Gamma, \lambda) := \|w - w'\|^2_{(\sum_{z \in \mathcal{Z}} \lambda_z \Gamma^\top zz^\top \Gamma)^{-1}}$
4: **while** $|\mathcal{W}_k| > 1$ **do**
5: $\quad \lambda_k = \arg\min_{\lambda \in \Delta(\mathcal{Z})} \max_{w,w' \in \mathcal{W}_k} f(w, w', \Gamma, \lambda)$.
6: $\quad \rho(\mathcal{W}_k) = \min_{\lambda \in \Delta(\mathcal{Z})} \max_{w,w' \in \mathcal{W}_k} f(w, w', \Gamma, \lambda)$
7: $\quad N_k := \lceil 2(1+\omega)2^{2k}\rho(\mathcal{W}_k)L_\nu \log\left(4k^2|\mathcal{W}|/\delta\right)\rceil \vee r(\omega)$
8: $\quad$ Pull arms in $Z_{N_k} = \text{ROUND}(\lambda_k, N_k)$ and observe $Y_{N_k}$.
9: $\quad$ Compute $\widehat{\theta}^k_\Gamma = \left(Z_{N_k}^\top Z_{N_k}\Gamma\right)^{-1} Z_{N_k}^\top Y_{N_k}$
10: $\quad \mathcal{W}_{k+1} = \mathcal{W}_k \backslash \{w \in \mathcal{W}_k | \exists w' \in \mathcal{W}_k, \langle w' - w, \widehat{\theta}^k_\Gamma \rangle > 2^{-k}\}, k \leftarrow k+1$
11: **end while**
12: **Output:** $w \in \mathcal{W}_k$

---

$\rho^* \to \infty$ as $\varepsilon \to 0$. Intuitively in the limit, regardless of which instrument $i \leq d$ is being pulled, the resulting distribution on the treatments is uniform (the instruments are *weak*). Thus, it is impossible to deconfound the measurement noise, and recover an estimate of $\theta$. This is a phenomenon which does not arise in the standard multi-armed bandit case with unconfounding.

*Remark* 3.3. We also consider a setting where instead of given $\Gamma$ directly, we are given an estimate $\widehat{\Gamma}$ of $\Gamma$ based on offline data. We discuss such an adaptation of Algorithm 1 to this setting in Appendix I and provide a sample complexity which reflects the error in $\Gamma$ (scaling with $\rho^*(\gamma)$ for $\gamma > 0$). We remark that this result is subsumed by the approach of Section 3.2 and so we omit it in the main text.

**Lower bound.** Due to the noise model from confounding and the dependence of the noise $\theta^\top \eta + \varepsilon$, the instance-dependent lower bounds of [15] do not immediately apply. We develop a lower bound tailored for the confounding setting **that nearly match the upper bounds** of our algorithms. What's more, our lower bound illustrates the additional difficulty that arises from confounding by an additional factor of $d^2$ compared to the standard transductive linear bandit problem in the most general setting where entries of $\eta$ are sub-Gaussian, but not necessarily independent nor bounded. Due to space limit, we defer it to Appendix D.

## 3.2 Fully Unknown Structural Model

We now consider the setting where $\Gamma$ is fully unknown. The difficulty of this setting is that the data collection process needs to support both estimation of $\Gamma$ and $\theta$ simultaneously. Our algorithm, built upon Algorithm 1, is summarized in Algorithm 3. At its core, each phase of the algorithm is divided into two sub-phases, for estimating $\Gamma$ and $\theta$ respectively. Specifically, the second sub-phase is essentially same as Algorithm 1 with $\widehat{\Gamma}_k$ in place of $\Gamma$ where $\widehat{\Gamma}_k$ is estimated from the first sub-phase. The main novelty of our algorithmic design lies in the first sub-phase, which resolves the challenge of performing the optimal design for estimating $\Gamma$. To explain this challenge, the confidence interval for P-2SLS estimators of Theorem 2.3 indicates that one should pull arms so that we control both $D_2 := \max_{w,w'} \|w - w'\|^2_{\widehat{A}(Z_{T_2}, \widehat{\Gamma}_k)}$ (error from $\hat{\theta}_{\text{P−2SLS}}$) and $D_1 := \max_{w,w'} \|w - w'\|^2_{\widehat{A}(Z_{T_1}, \widehat{\Gamma}_k)}$ (error from $\widehat{\Gamma}_k$) to be below the target error $O(\zeta^2_k)$ at each phase (ignoring unimportant factors for discussion). Controlling $D_2$ is trivial, which is done in the second sub-phase as we described above.

However, for $D_1$, a similar strategy cannot be done because the estimate $\widehat{\Gamma}_k$ is computed directly by sampling arms in $Z_{T_1}$. That is, the ideal design, based on which we *will* collect data points $z_1, \ldots, z_n$, requires access to the random matrix $\widehat{\Gamma}_k$ that can only be computed *after* sampling $z_1, \ldots, z_{n,}$. This

---

**Algorithm 3** CPEUG: Confounded pure exploration with with **unknown** $\Gamma$

---

**Input** $\mathcal{Z}, \mathcal{W}, \delta, L_\nu \geq \sigma_\nu^2, L_\eta \geq \sigma_\eta^2, \omega, \gamma_{\min} \leq \lambda_{\min}(\Gamma), \lambda_E, \kappa_0$

**Initialize:** $k = 1, \mathcal{W}_1 = \mathcal{W}, \widehat{\Gamma}_0 = \perp, \zeta_1 = 1$

**Define** $f(w, w', \Gamma, \lambda) := \|w - w'\|_{(\sum_{z \in \mathcal{Z}} \Gamma^\top \lambda_z z z^\top \Gamma)^{-1}}^2, M := \frac{32 L_\eta}{\gamma_{\min}^2 \sigma_{\min}(A(\lambda_E, I))} \vee 1, \delta_\ell := \frac{\delta}{4\ell^2}$

**while** $|\mathcal{W}_k| > 1$ **do**

$\quad \widehat{\Gamma}_k = \Gamma - \texttt{estimator}\left(\mathcal{W}_k, \widehat{\Gamma}_{k-1}, \zeta_k, \delta/k^2, \omega, \lambda_E, M, L_\eta\right)$ $\qquad\qquad\qquad$ ▷ Step 1: update $\widehat{\Gamma}$

$\quad \widehat{\theta}_{\text{P-2SLS}}^k = \theta - \texttt{estimator}\left(\mathcal{W}_k, \delta/k^2, \zeta_k, \widehat{\Gamma}_k, \omega, L_\nu\right)$ $\qquad\qquad\qquad\qquad$ ▷ Step 2: update $\widehat{\theta}$

$\quad \mathcal{W}_{k+1} = \mathcal{W}_k \setminus \left\{ w \in \mathcal{W}_k \mid \exists w' \in \mathcal{W}_k, \text{s.t.}, \left\langle w' - w, \widehat{\theta}_{\text{P-2SLS}}^k \right\rangle > \zeta_k \right\}$ $\qquad$ ▷ Step 3: elimination

$\quad k \leftarrow k + 1, \zeta_k = 2^{-k}$

**end while**

**Output:** $\mathcal{W}_k$

---

creates a cycle that seems impossible to resolve. Such an issue, to our knowledge, has not been seen in existing work on pure exploration, and thus resolving it is our key technical contribution.

Our solution is to compute the design based on $\widehat{\Gamma}_k$ from the previous phase. We then perform a doubling trick where we double the sample size (while following the computed design) until $D_1$ becomes smaller than the target error $O(\zeta_k^2)$. The intuition is that in later phases the estimate $\widehat{\Gamma}_k$ from the previous phase will be accurate enough to ensure that the design is efficient. Note that this novel algorithm induces extra randomness in how many samples we end up collecting in the first sub-phase, which remains random even after conditioning on the history, unlike the second sub-phase. This makes the analysis challenging, which we describe after the main result.

Our algorithm additionally employs the so-called E-optimal design to ensure that the covariance matrix of the collected data used to estimate $\Gamma$ is well-conditioned. This conditioning is required to ensure that $\widehat{\Gamma}_k$ concentrates fast enough to $\Gamma$ as shown in the analysis. The E-optimal design is a well-known design objective in experimental design that aims to maximize the smallest singular value: $\lambda_E^* := \arg\min_{\lambda \in \Delta(\mathcal{Z})} \sigma_{\max}(V^{-1}(\lambda))$, where $V = \sum_{z \in \mathcal{Z}} \lambda_z z z^\top$. We denote $\kappa_0^{-1} := \sigma_{\max}(V^{-1}(\lambda_E^*)) = \sigma_{\min}^{-1}(V(\lambda_E^*))$ as the smallest singular value achieved by the E-optimal design.

---

**Algorithm 2** $\theta - \texttt{estimator}$

---

**Input** $\mathcal{W}, \delta, \zeta, \widehat{\Gamma}, \omega, L_\nu$

$\widehat{\lambda} = \arg\min_{\lambda \in \Delta(\mathcal{Z})} \max_{w, w' \in \mathcal{W}} f(w, w', \widehat{\Gamma}, \lambda)$

$\rho(\mathcal{W}) = \min_{\lambda \in \Delta(\mathcal{Z})} \max_{w, w' \in \mathcal{W}} f(w, w', \widehat{\Gamma}, \lambda)$

$N_2 = \left\lceil 2(1 + \omega) \zeta^{-2} \rho(\mathcal{W}) L_\nu \log\left(\frac{4|\mathcal{W}|}{\delta}\right) \right\rceil \vee r(\omega)$

get $N_2$ samples per design $\widehat{\lambda}$ denoted as $\{Z_2, X_2, Y_2\}$ $\quad$ ▷ via ROUND

update $\widehat{\theta}_{\text{P-2SLS}} = (\widehat{\Gamma}^\top Z_2^\top Z_2 \widehat{\Gamma})^{-1} \widehat{\Gamma}^\top Z_2^\top Y_2$

**Output:** $\widehat{\theta}_{\text{P-2SLS}}$

---

We present our analysis result Theorem 3.4 where we show that, even without knowledge of $\Gamma$, the sample complexity scales with the key problem difficulty $\rho^*$ almost matching the sample complexity of Algorithm 1 which relies on knowledge of $\Gamma$.

**Theorem 3.4.** *Algorithm 3 is $\delta$-PAC and terminates in at most*

$$(1 + \omega)((L_\nu \log(1/\delta) + L_\eta \|\theta\|_2^2 (d + \log(1/\delta)))\rho^* + (d + \log(1/\delta))(L_\eta \|\theta\|_2^2 \rho_0 + M))$$

*pulls, ignoring both of the additive and multiplicative logarithms of $\Delta, |\mathcal{W}|, \rho^*, \rho_0, M$, where*

$$\rho_0 = \max_{w \in \mathcal{W} \setminus \{w^*\}} \|w^* - w\|_{(\sum_{z \in \mathcal{Z}} \lambda_{E,z} \Gamma^\top z z^\top \Gamma)^{-1}}^2, \text{ and } M = \frac{32 L_\eta}{\gamma_{\min}^2 \sigma_{\min}(A(\lambda_E, I))} \vee 1.$$

*Note that $\rho_0$ does not get hurt by $\langle w^* - w, \theta \rangle$, ($\rho^*$ does). It comes from the fact that in the first phase, we initialize that algorithm with E-optimal design.*

The challenge of the analysis can be summarized in two-fold. First, since the concentration result in Theorem 2.3 is w.r.t. $\widehat{\Gamma}_k$, we need to analyze how the random matrix $\widehat{\Gamma}_k$ concentrates around $\Gamma$ and how this impacts the sample complexity. For this, we develop a novel concentration inequality

---

**Algorithm 4** $\Gamma - \texttt{estimator}$

---

**Input** $\mathcal{W}, \widehat{\Gamma}, \zeta, \delta, \omega, \lambda_E, M, L_\eta$

**Define** $\texttt{Stop}(\mathcal{W}, Z, \Gamma, \delta) := \max_{w,w' \in \mathcal{W}} \|w - w'\|_{\bar{A}(Z,\Gamma)^{-1}} \|\theta\|_2 \sqrt{L_\eta \overline{\log}(Z, \delta)}$

**Initialize** $\ell = 1$, $N_{0,0} = 0$ $\qquad\qquad\qquad\qquad\qquad\qquad\qquad\qquad\qquad$ ▷ doubling trick initialization

**if** $\widehat{\Gamma} = \perp$ **then**

$\quad$ **while** $\ell = 1$ or $\texttt{Stop}\left(\mathcal{W}, Z_{0,\ell}, \widehat{\Gamma}', \delta_\ell\right) > 1$ **do**

$\quad\quad$ get $2^{\ell-1}\left(r(\omega) \vee \frac{2}{\kappa_0}\right)$ samples denoted as $\{Z_{0,\ell}, X_{0,\ell}, Y_{0,\ell}\}$ per design $\lambda_E$ $\qquad$ ▷ via ROUND

$\quad\quad$ Update $\widehat{\Gamma}'$ by $\texttt{OLS}$ on $\{Z_{0,\ell}, X_{0,\ell}\}$, $\ell \leftarrow \ell + 1$

$\quad$ **end while**

**else**

$\quad$ $\tilde{\lambda} = \arg\min_{\lambda \in \Delta(\mathcal{Z})} \max_{w,w' \in \mathcal{W}} f(w, w', \widehat{\Gamma}, \lambda)$

$\quad$ $N' = \left\lfloor 4gdM \ln\left(1 + 2M\left(d + L_z^2\right) + 2M 2gdM\right) + 8M \ln\left(\frac{2 \cdot 6^d}{\delta}\right) \vee r(\omega) \right\rfloor$

$\quad$ **while** $\ell = 1$ or $\texttt{Stop}\left(\mathcal{W}, Z_{0,\ell} \cup Z_{1,\ell}, \widehat{\Gamma}, \delta_\ell\right) > \zeta$ **do**

$\quad\quad$ $N_{1,\ell} = 2^\ell N'$ $\qquad\qquad\qquad\qquad\qquad\qquad\qquad\qquad\qquad\qquad\qquad$ ▷ doubling trick update

$\quad\quad$ get $N_{1,\ell}$ samples per $\tilde{\lambda}$ denoted as $\{Z_{1,\ell}, X_{1,\ell}, Y_{1,\ell}\}$ $\qquad\qquad\qquad$ ▷ via ROUND

$\quad\quad$ $N_{0,\ell} = \left\lceil 2gdM \ln\left(M\left(d + N_{1,\ell} + L_z^2\right)\right) + 4M \ln\left(\frac{2 \cdot 6^d}{\delta_\ell}\right) \vee r(\omega) \vee \frac{2}{\kappa_0} \right\rceil$

$\quad\quad$ get $(N_{0,\ell} - N_{0,\ell-1})$ samples per $\lambda_E$ augmented to $\{Z_{0,\ell-1}, X_{0,\ell-1}\}$ and get $\{Z_{0,\ell}, X_{0,\ell}\}$

$\quad\quad$ Update $\widehat{\Gamma}'$ by $\texttt{OLS}$ on $\{Z_{0,\ell} \cup Z_{1,\ell}, X_{0,\ell} \cup X_{1,\ell}\}$, $\ell \leftarrow \ell + 1$

$\quad$ **end while**

**end if**

**Output:** $\widehat{\Gamma}'$

---

that relates the confidence width involving $\hat{\Gamma}$ from Theorem 2.3 with the same quantity involving $\Gamma$ in place of $\hat{\Gamma}$. Second, our algorithm creates a long-range error propagation, which is highly nontrivial to analyze. To see this, the quality of $\hat{\Gamma}_k$ is affected by the design objective function $\max_{w,w'} f(w, w', \hat{\Gamma}_{k-1}, \lambda)$, which depends on the error of the estimate $\hat{\Gamma}_{k-1}$ from the previous phase. This error is, in turn, affected by the error of $\hat{\Gamma}_{k-2}$ by the same mechanism. This is repeated all the way back to the first phase. Thus, any abnormal behavior from the first iteration will have a cumulative impact to even the end. In our analysis, we successfully analyze how the error is propagated from the previous iterations, which forms a complicated recursion. Resolving this recursion is our key novelty in the analysis.

*Remark* 3.5. Our algorithm requires knowledge of a lower bound $\gamma_{\min}$ of $\lambda_{\min}(\Gamma)$. The knowledge of $\gamma_{\min}$ is for simplicity only as one can obtain such a lower bound that is at least half of the true value $\lambda_{\min}(\Gamma)$ via an efficient sampling procedure that we describe in Appendix K.

**Experiments.** We provide experiments for the instance of Section 1.2 in the Appendix E. The experiments show that our approach is more sample efficient than natural passive baselines (e.g. A/B testing), or naively applying existing Pure-Exploration linear bandit methods and performs similarly to the oracle complexity.

## 4 Conclusion

This work introduces the CPET-LB problem in which the learning protocol is characterized by a linear structural equation model governed by parameters $\Gamma$ and $\theta$. We provide a general solution that simultaneously estimates the structural model while optimally designing to learn the best-arm. The key ideas behind our approach are based on linear experimental design techniques, an instrumental variable estimator whose variance can be controlled by the design, and novel finite-time confidence intervals on this estimator. This paper presents a number of directions for future work including considering situations where the $d_z \neq d_x$, analysis to improve the dependence on the underlying noise variance, and the pursuit of a tight information-theoretic instance-dependent lower-bound. We hope that this line of work motivates increased discussion of the real impact of confounding on applicability of adaptive experimentation.

## Acknowledgements

Kwang-Sung Jun and Yao Zhao were supported in part by the National Science Foundation under grant CCF-2327013.

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

# Appendix

## Table of Contents

## A  Broader Impacts

This work is algorithmic and not tied to a particular application that would have immediate negative impact.

## B  Illustrative Example

We now present an illustrative experiment that highlights the challenges of endogenous noise and the insufficiency of standard experimentation approaches used in the absence of confounding.

**Instance Definition.** Toward connecting back to membership example in Section 1, consider that a service has $d$ membership options given by the set $\mathcal{A} = \{1, \ldots, d\}$. Let the set $\mathcal{Z} = \{e_1, \cdots, e_d\}$ represent encouragements (incentives or advertisements) for the corresponding membership options given by $\mathcal{W} = \mathcal{X} = \{e_1, \cdots, e_d\}$. We consider a location model that assumes each user $t \in \mathbb{N}$ arriving online has an underlying unobserved one-dimensional preference $u_t \sim \mathcal{N}(0, \sigma_u^2)$. If an algorithm presents the user with encouragement $z_{I_t} = e_{I_t}$ for $I_t \in \mathcal{A}$, then the user selects into the

membership level given by $J_t = \min_{j \in \mathcal{A}} |I_t + u_t - j|$ so that $x_t = e_{J_t}$. For a visual depiction, see Figure 3a. This process captures a user being more likely to opt-in to membership levels that are closer to the encouragement that they were presented. The outcome is then given by $y_t = x_t^\top \theta + u_t$. This problem instance is a specific compliance instance. For this experiment, we take $d = 6$, let $\theta = \begin{bmatrix} 1 & -0.95 & 0 & 0.45 & 0.95 & 0.99 \end{bmatrix}$ and $\sigma_u^2 = 0.35$. Observe that the optimal evaluation vector is $w^* = e_1 = \arg\max_{w \in \mathcal{W}} w^\top \theta$.

We simulate a UCB strategy which maintains estimates of the average reward of each of the possible $d$ incentives, namely $\widehat{\mu}_{i,t} = \sum_{s=1}^t \mathbf{1}\{z_t = e_i\} y_t$ and then pulls the one with the highest upper confidence bound. This models current practice of using a bandit algorithm to select which incentive to show a user. Our results averaged over 100 simulations are in Figure 3d. At each round we estimate the average reward of each level using an OLS estimator, i.e. $\widehat{\theta}_{i,t}^{OLS} = \sum_{s=1}^t \mathbf{1}\{x_t = e_i\} y_t / \sum_{s=1}^t \mathbf{1}\{x_t = e_i\}$, and check whether it matches the true value (denoted as UCB-OLS). We also consider an instrumental variable-estimator (see the next Section) which incorporates knowledge of $\Gamma$ similar to 2SLS to deconfound our estimate (UCB-IV). As the plot demonstrates, UCB-OLS completely fails to identify $\theta_1 = \arg\max_{i \in d} \theta_i$ (this line is hard to see it is at 0) due to a biased estimate, whereas UCB-IV does better. However, UCB-IV methods seem to have a constant probability of error. To see why, note that the expected reward from pulling $z = e_i$ is $e_i^\top \Gamma \theta$. These values are plotted in orange in Figure 3c. In particular, with some constant probability, UCB runs on the empirical rewards from pulling $z$'s zeroes in on arm 6, and as a result fails to give enough samples to learn that arm 1 is indeed the best. In contrast, our proposed method CPEG, Algorithm 1 manages to find the best arm with significantly higher probability (the algorithm was run with $\delta = .1$) in the given time horizon.

## C    Standard 2SLS estimator

Consider $\Psi = \widehat{\Gamma}_{2SLS} = (Z_T^\top Z_T)^{-1} Z_T^\top X_T$. In this setting, we recover the standard two-stage-least-squares (2SLS) estimator,

$$\widehat{\theta}_{\text{2SLS}} = (X_T^\top Z_T (Z_T^\top Z_T)^{-1} Z_T^\top X_T)^{-1} X_T^\top (Z_T^\top Z_T)^{-1} Z_T^\top Y_T = (Z_T^\top X_T)^{-1} Z_T^\top Y_T.$$

Note that the 2SLS estimator is a biased, but consistent estimator of the parameter $\theta$ [3, 18].

Note that in particular, the asymptotic variance of 2SLS is known to be $\sigma_\varepsilon^2 \|w\|_{(\widehat{\Gamma}_{\text{2SLS}}^\top Z_T^\top Z_T \widehat{\Gamma}_{\text{2SLS}})^{-1}}$ [18]. Recent work by [11] provides a confidence interval of the form $|w^\top (\widehat{\theta}_{\text{2SLS}} - \theta)| \leq O(d\sigma_\varepsilon^2 \|w\|_{(\widehat{\Gamma}_{\text{2SLS}}^\top Z_T^\top Z_T \widehat{\Gamma}_{\text{2SLS}})^{-1}} \sqrt{\log(T/\delta)})$. However, it's unclear how to use the form of their confidence interval directly for experimental design due to the dependence of $\widehat{\Gamma}_{2SLS}$ on the random quantity $X$. In addition, their work is not sufficiently general to handle the general forms of noise that we consider in Lemma 2.1.

## D    A non-interactive lower bound

Due to the noise model from confounding and the dependence of the noise $\theta^\top \eta + \varepsilon$, the instance-dependent lower bounds of [15] do not immediately apply. In this section, we develop a lower bound tailored for the confounding setting.

Toward characterizing the optimal sample complexity, we develop a lower bound for a specific non-adaptive algorithm $\mathcal{A}$ that has access to the matrix $\Gamma$ governing the structural equation model. In particular, suppose that the non-adaptive algorithm $\mathcal{A}$ is allowed to select a sequence of $T$ measurements $\{z_{I_1}, \ldots z_{I_t} \ldots, z_{I_T}\}$ to query prior to collecting any observations, where $I_t$ represents the index of the vector $z \in \mathcal{Z}$ chosen at time $t \in \{1, \ldots, T\}$. Then, given the observations $\{y_1, \ldots, \ldots y_t, \ldots, y_T\}$ generated by the environment, a candidate optimal vector $\widehat{w} \in \mathcal{W}$ is returned by the algorithm. We are interested in the necessary number of observations $T$ that must be collected in order to ensure $\mathbb{P}(\widehat{w} \neq w^*) \leq \delta$ for some $\delta \in (0, 1)$. Thus, it is natural that the optimal non-adaptive algorithm $\mathcal{A}$ using the estimator $\widehat{\theta}_{\texttt{oracle}}$ forms a recommendation rule such that $\widehat{w} = \arg\max_{w \in \mathcal{W}} w^\top \widehat{\theta}_{\texttt{oracle}}$. We now state our lower bound result with respect to the non-adaptive oracle algorithm.

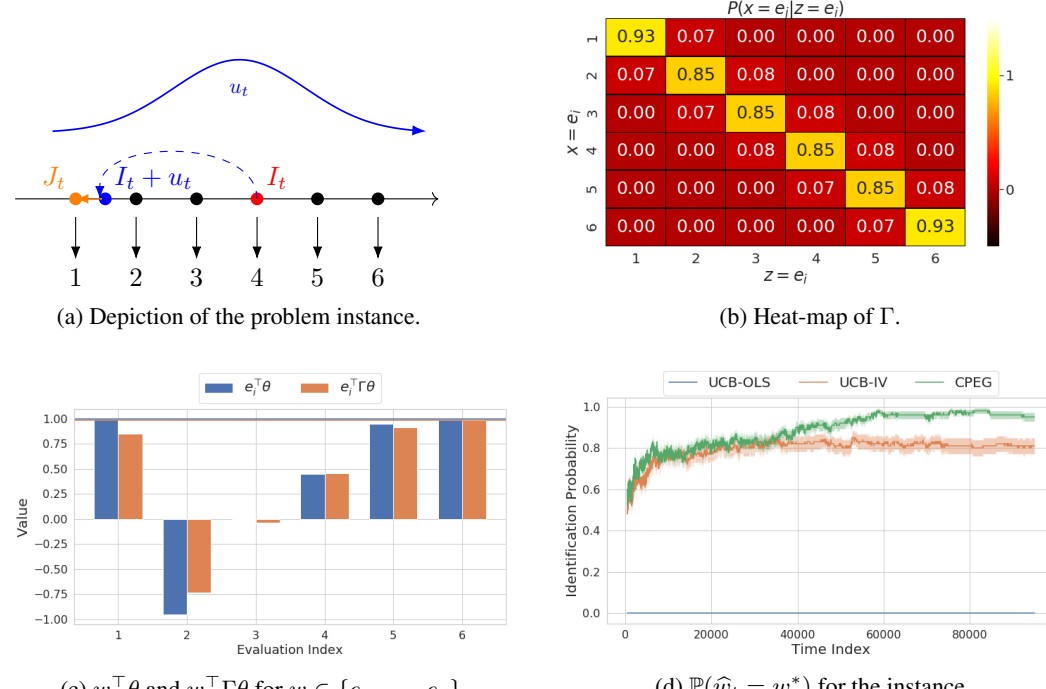

(a) Depiction of the problem instance.

(b) Heat-map of $\Gamma$.

(c) $w^\top\theta$ and $w^\top\Gamma\theta$ for $w \in \{e_1,\ldots,e_d\}$

(d) $\mathbb{P}(\widehat{w}_t = w^*)$ for the instance

Figure 3: (a) A visual depiction of the problem instance from Section B. The user is presented with encouragement $I_t \in \mathcal{A}$ and the user choice is given by $J_t$ where $J_t = \min_{j\in\mathcal{A}}|I_t + u_t - j|$ and $u_t \sim \mathcal{N}(0,\sigma_u^2)$. b) A heat-map showing the structural parameter $\Gamma$ for the problem instance from Section B. (c) A bar chart showing $\mathbb{E}[y|x=w] = w^\top\theta$ and $\mathbb{E}[y|z=w] = z^\top\Gamma\theta$ for all $w \in \mathcal{W}$. This chart shows that the optimal evaluation vector is $w^* = e_1 = \arg\max_{w\in\mathcal{W}}\mathbb{E}[y|x=w]$, while $e_6 = \arg\max_{w\in\mathcal{W}}\mathbb{E}[y|z=w]$ and consequently estimation based on this quantity is problematic. (d) The probability of identifying $w^* = e_1$ for a collection of algorithms on the CPET-LB instance described in Section B. Standard optimistic sampling approaches in combination with an ordinary least squares estimator leads to faulty inferences. Given an instrumental variable estimator, these experimental designs eventually give high probability identification but do so inefficiently compared to our proposed approach (see Section 3).

**Theorem D.1** (Non-Adaptive Oracle Lower Bound). *Consider a problem instance characterized by $\mathcal{W} \subset \mathbb{R}^d$, $\mathcal{Z} \subset \mathbb{R}^d$, $\Gamma \in \mathbb{R}^{d\times d}$, and $\theta \in \mathbb{R}^d$. Assume $\Gamma$ is known, $\theta$ is unknown, and the noise process is jointly Gaussian and defined by $\gamma := \begin{bmatrix} \eta & \varepsilon \end{bmatrix} \sim \mathcal{N}(0,\Sigma)$ where $\Sigma \in \mathbb{R}^{(d+1)\times(d+1)}$ is an arbitrary correlation matrix. For $\delta \in (0,0.05]$, if the non-adaptive oracle algorithm acquires $T \leq \sigma^2\rho^* \log(1/\delta)/2$ samples on the problem instance where $\sigma^2 := v^\top\Sigma v$ and $v := \begin{bmatrix} \theta & 1 \end{bmatrix} \in \mathbb{R}^{d+1}$, then $\mathbb{P}(\widehat{w} \neq w^*) \geq \delta$.*

**Corollary D.2.** *There exists a problem instance characterized by $\mathcal{W} \subset \mathbb{R}^d$, $\mathcal{Z} \subset \mathbb{R}^d$, $\Gamma \in \mathbb{R}^{d\times d}$, and $\theta \in \mathbb{R}^d$ with a noise process satisfying Assumption 1 such that if the non-adaptive oracle algorithm acquires $T \leq \max\{d\|\theta\|_2^2, \sqrt{d}\|\theta\|_2\}\rho^* \log(1/\delta)/2$ samples, then $\mathbb{P}(\widehat{w} \neq w^*) \geq \delta$ for $\delta \in (0,0.05]$.*

The proof of Theorem D.1 is in Appendix F.1. Notably, the result is reminiscent of lower bounds for the standard pure exploration transductive linear bandit problem without confounding [15, 22] when given the measurement set $\{\Gamma^\top z\}_{z\in\mathcal{W}}$, evaluation set $\mathcal{Z}$, and parameter $\theta$.

Notably, the upper bounds for our algorithms nearly match the lower bound of Theorem D.1. However, it is interesting to observe that the sample complexity incurs an additional factor of $d^2$ relative to the standard transductive linear bandit problem in the most general setting where entries of $\eta$ are sub-Gaussian, but not necessarily independent nor bounded. This illustrates the additional difficulty that arises from confounding. We point out that this is not likely to be a tight lower bound. In

particular, it is a lower bound with respect to a non-adaptive algorithm that uses the particular choice of estimator. We leave improved lower bounds to future work.

# E  Experiments

We now present experiments a collection of experiments on `CPET-LB` problem instances. The experiments demonstrate that our approach produces efficient designs for inference and estimation.

## E.1  Comparison Algorithms

The baselines that our approaches are compared with are discussed below. We run experiments both when $\Gamma$ is known and when $\Gamma$ is fully unknown.

### E.1.1  Known $\Gamma$.

To standardize the experiments, the baselines considered run in rounds mirroring the structure of Algorithm 1. Specifically, in round $k \in \mathbb{N}$ a sampling algorithm selects a design $\lambda_k \in \Delta(\mathcal{Z})$, collects $N_k$ samples from the design, and forms a $\Psi - $ IV estimate of $\theta$ with $\Psi = \Gamma$ that is combined with a confidence interval (Lemma 2.2 to either eliminate evaluation vectors or validate a stopping condition. The number of samples $N_k$ taken in round $k \in \mathbb{N}$ by any of the algorithms is given by $N_k = \lceil 2(1+\omega)\zeta_k^{-2}\rho(\mathcal{W}_k)L_\nu \log(4k^2|\mathcal{W}|/\delta)\rceil \vee r(\omega)$ where $\rho(\mathcal{W}_k) = \max_{w,w' \in \mathcal{W}_k} f(w, w', \Gamma, \lambda)$ for design $\lambda_k \in \Delta(\mathcal{Z})$ and an active set of evaluation vectors $\mathcal{W}_k$. The round sample count guarantees that given any experimental design, all vectors $w \in \mathcal{W}$ such that $(w^* - w)^\top \theta > 2 \cdot 2^{-k}$ can be determined to be suboptimal by the end of round $k$. The sampling methods we consider are now described.

- *Static Oracle.* This design selects $\lambda_k = \arg\min_{\lambda \in \Delta(\mathcal{Z})} \max_{w \in \mathcal{W} \setminus \{w^*\}} f(w^*, w', \Gamma, \lambda)$.

- *Static XY-Optimal.* This design selects $\lambda_k = \arg\min_{\lambda \in \Delta(\mathcal{Z})} \max_{w,w' \in \mathcal{W}} f(w, w', \Gamma, \lambda)$.

- *Static Uniform.* This design selects $\lambda_{k,z} = 1/|\mathcal{Z}| \; \forall \; z \in \mathcal{Z}$.

- *Adaptive Uniform (SE).* This design selects $\lambda_{k,w} = 1/|\mathcal{W}_k| \; \forall \; w \in \mathcal{W}_k$. Note that this algorithm is effectively an adaption of action-elimination [14] .

The static designs are independent of the round and simply terminate when all evaluation vectors can be eliminated except for a recommended optimal vector $\widehat{w}$.

### E.1.2  Unknown $\Gamma$.

For this set of experiments, we compare Algorithm 3 against a collection of variations of the sampling procedures. Specifically, we compare against methods that either replace only the experimental design for estimating $\Gamma$, or only the experimental design for estimating $\theta$, or both with uniform sampling. We label the approaches as $N - N$, where $N$ represents the sampling approaches (XY or uniform) for $\Gamma$ and $\theta$ respectively. Moreover, to make our approach more practical, we modify the algorithm so that $\overline{\log}(Z_T, \delta) = 4d + \log(1/\delta)$. The step of incrementally adding more E-optimal design samples is also removed, so we collect E-optimal design samples only once in the beginning of Algorithm 4. We find that even with these modifications to the algorithm, correctness is maintained empirically.

## E.2  Experiment 1: Jump-Around Instance

We first return back to the location model of Section B. Recall that $\mathcal{Z} = \mathcal{W} = \mathcal{X} = \{e_1, \cdots, e_d\}$. For this experiment, we take $d = 6$, let $\theta = \begin{bmatrix} 1 & -0.95 & 0.45 & 0.45 & 0.95 & 0.45 \end{bmatrix}$ and $\sigma_u^2 = 0.275$. The results of the experiment are shown in Figure 4a for the case of known $\Gamma$. We see that Algorithm 1 performs much better than the baselines and nearly matches the oracle design. Delving into the approach, it is able to quickly eliminate all but $w_1$ and $w_{d-1}$ and then puts more mass on $z_1$ and $z_{d-1}$ to reduce the uncertainty on $w_1$ and $w_{d-1}$. For the case of unknown $\Gamma$, the results are shown in Figure 4d, where $\theta_5$ is reduced to $0.9$ so that all approaches could finish.

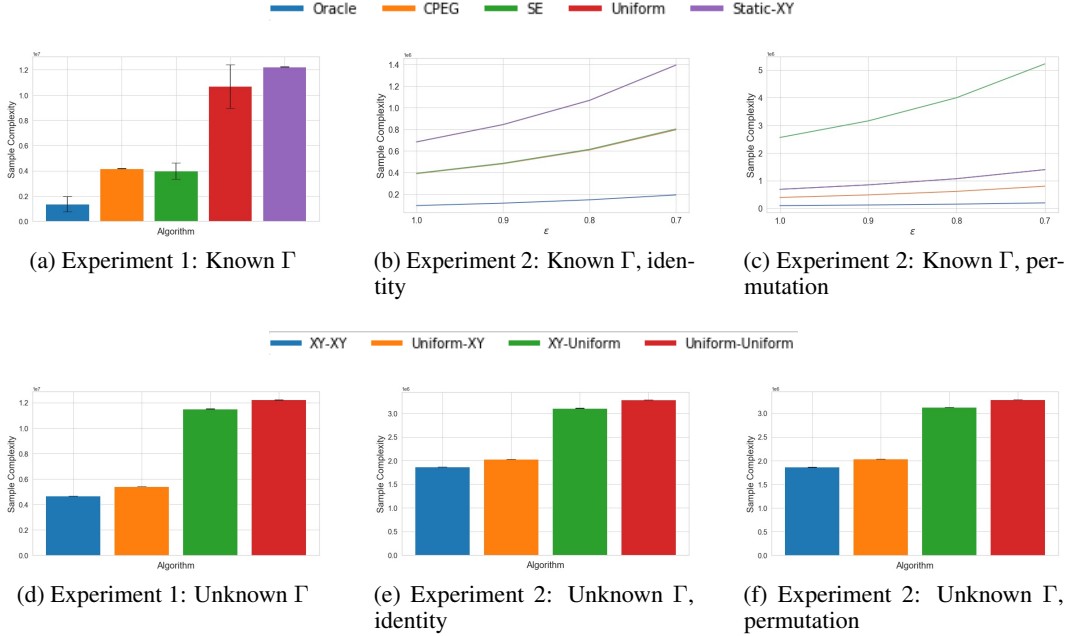

(a) Experiment 1: Known $\Gamma$

(b) Experiment 2: Known $\Gamma$, identity

(c) Experiment 2: Known $\Gamma$, permutation

(d) Experiment 1: Unknown $\Gamma$

(e) Experiment 2: Unknown $\Gamma$, identity

(f) Experiment 2: Unknown $\Gamma$, permutation

Figure 4: Sample complexity for algorithms on `CPET-LB` problems. Our approach is consistently competitive across the experiments.)

### E.3 Experiment 2: Interpolation Instance

Let $\mathcal{Z} = \mathcal{W} = \mathcal{X} = \{e_1, \cdots, e_d\}$ define the measurement, evaluation, and observation sets. We first consider that $\Gamma := \frac{(1-\varepsilon)}{d} 1_d 1_d^\top + \varepsilon I_d$ for a parameter $\varepsilon \in (0,1)$ where $1_d$ is a $d$-dimensional vector of 1's and $I_d$ is the $d$-dimensional identity matrix. For this experiment, we take $d = 4$ and let $\theta = \begin{bmatrix} 0.5 & 0.583 & 0.67 & 0.75 \end{bmatrix}$. As in all compliance instances, $\eta_t = x_t - \Gamma^\top z_t$, and in this simulation $\eta_t = 0.4 \eta_t^\top v_t$, where $v_t = \bar{v}_t / \|\bar{v}_t\|_2$ and $\bar{v}_t \sim \mathcal{N}(0, I_d)$. The results of the experiment are shown in Figure 4b for $\varepsilon \in \{1, 0.9, 0.8, 0.7\}$ with $\Gamma$ known. Note that Static-XY and Uniform overlap, and SE and CPEG overlap. We see that Algorithm 1 and the adaptive uniform strategy perform similarly and near optimally. This is to be expected since the most efficient way to gather observations for treatments is to encourage that treatment, given that if the encouragement is not followed each of the alternatives is equally likely and provides no additional information of interest. Moreover, as discussed earlier, the problem gets more challenging as $\Gamma \to 1_d 1_d^\top / d$. Note that the identity matrix could be replaced with a permutation matrix, in which case uniform sampling with elimination becomes highly suboptimal. The results for the case of $\Gamma$ unknown are shown in Figure 4e with $\varepsilon = 0.99$. This shows the value that comes from the experimental design for estimating both $\Gamma$ and $\theta$.

To demonstrate the superiority of our algorithm over SE, we also consider that $\Gamma := \frac{(1-\varepsilon)}{d} 1_d 1_d^\top + \varepsilon I_d^p$, where $I_d^p$ is a permutation matrix as follows,

$$I_d^p = \begin{bmatrix} 0 & 1 & 0 & 0 \\ 0 & 0 & 1 & 0 \\ 0 & 0 & 0 & 1 \\ 1 & 0 & 0 & 0 \end{bmatrix}.$$

All other settings remain the same as in the previous interpolation instance. The results for the known $\Gamma$ case, shown in Figure 4c, indicate that SE exhibits significant underperformance due to its sampling rule not accounting for the permutation effect in $\Gamma$. In contrast, CPEG consistently achieves near-optimal performance. Note that Static-XY and Uniform still overlap. Figure 4f presents the results for the unknown $\Gamma$ case, where we can notice that, comparing with Figure 4e, estimating different permutation matrices (with the identity matrix as a special case) does not affect problem difficulty.

# F  Proofs of the lower bound

## F.1  Proof of Theorem D.1

*Theorem D.1.* Consider a problem instance characterized by $\mathcal{W} \subset \mathbb{R}^d$, $\mathcal{Z} \subset \mathbb{R}^d$, $\Gamma \in \mathbb{R}^{d \times d}$, and $\theta \in \mathbb{R}^d$. Assume $\Gamma$ is known, $\theta$ is unknown, and the noise process is jointly Gaussian and defined by $\gamma := \begin{bmatrix} \eta & \varepsilon \end{bmatrix} \sim \mathcal{N}(0, \Sigma)$ where $\Sigma \in \mathbb{R}^{(d+1) \times (d+1)}$ is an arbitrary correlation matrix. For $\delta \in (0, 0.05]$, if the non-adaptive oracle algorithm acquires $T \leq \sigma^2 \rho^* \log(1/\delta)/2$ samples on the problem instance where $\sigma^2 := v^\top \Sigma v$ and $v := \begin{bmatrix} \theta & 1 \end{bmatrix} \in \mathbb{R}^{d+1}$, then $\mathbb{P}(\widehat{w} \neq w^*) \geq \delta$.

*Proof.* We begin by recalling the framework of the non-adaptive oracle algorithm and discussing the properties of its estimator for the noise structure described in the statement of the result.

**Non-Adaptive Oracle and Instance Definition.** The non-adaptive oracle algorithm $\mathcal{A}$ selects $T$ measurements to query prior to collecting any data. Let $I_t$ represent the index of the vector $z \in \mathcal{Z}$ chosen at time $t \in \{1, \ldots, T\}$. The noise process for the instance under consideration is assumed to be jointly Gaussian and defined by $\gamma_t := \begin{bmatrix} \eta_t & \varepsilon_t \end{bmatrix} \sim \mathcal{N}(0, \Sigma)$ where $\Sigma \in \mathbb{R}^{(d+1) \times (d+1)}$ is an arbitrary positive semidefinite matrix. Defining $\bar{x}_{I_t} := \Gamma^\top z_{I_t}$, $v := \begin{bmatrix} \theta & 1 \end{bmatrix} \in \mathbb{R}^{d+1}$ and $\nu_t := v^\top \gamma_t$, the feedback model can be described as follows:

$$
\begin{aligned}
y_t &= x_t \theta + \varepsilon_t \\
&= (\Gamma^\top z_{I_t})^\top \theta + \eta_t^\top \theta + \varepsilon_t \\
&=: (\Gamma^\top z_{I_t})^\top \theta + v^\top \gamma_t \\
&=: \bar{x}_{I_t}^\top \theta + \nu_t.
\end{aligned}
$$

Observe that the noise is independent and identically distributed as $\nu_t \sim \mathcal{N}(0, \sigma^2)$ where $\sigma^2 := v^\top \Sigma v$ since $\gamma_t \sim \mathcal{N}(0, \Sigma)$. Moreover, the noise process is exogenous with $\mathbb{E}[\nu_t | \bar{x}_{I_t}] = 0$ since $\bar{x}_{I_t}$ is deterministic given the index choice $I_t$.

Let $\{z_{I_t}\}_{t=1}^T$, $\{\bar{x}_{I_t}\}_{t=1}^T$, and $\{y_t\}_{t=1}^T$ denote the observations collected by the non-adaptive oracle algorithm $\mathcal{A}$ and define $Z_T \in \mathbb{R}^{T \times d}$, $\bar{X}_T \in \mathbb{R}^{T \times d}$, and $Y_T \in \mathbb{R}^T$ to contain the respective stacked observations. Algorithm $\mathcal{A}$ obtains an estimate $\widehat{\theta}_{\texttt{oracle}}$ by minimizing the sum of squares as follows:

$$
\widehat{\theta}_{\texttt{oracle}} := \arg\min_{\widehat{\theta} \in \mathbb{R}^d} \sum_{t=1}^T (y_t - \bar{x}_{I_t} \widehat{\theta})^2 = (\bar{X}_T^\top \bar{X}_T)^{-1} \bar{X}_T^\top Y_T = (Z_T^\top Z_T \Gamma)^{-1} Z_T^\top Y_T.
$$

Given $\widehat{\theta}_{\texttt{oracle}}$, the non-adaptive oracle algorithms $\mathcal{A}$ returns a recommendation defined by $\widehat{w} = \arg\max_{w \in \mathcal{W}} w^\top \widehat{\theta}_{\texttt{oracle}}$. Note that since $\widehat{\theta}_{\texttt{oracle}}$ is obtained by least squares with exogenous, independent and identically distributed mean-zero Gaussian noise, it is straightforward to verify the estimator is distributed as

$$
\widehat{\theta}_{\texttt{Oracle}} - \theta \sim \mathcal{N}\left(0, \sigma^2 \cdot \bar{A}(Z_T, \Gamma)^{-1}\right), \tag{8}
$$

where

$$
\bar{A}(Z_T, \Gamma) := \left( \sum_{t=1}^T \Gamma^\top z_{I_t} z_{I_t} \Gamma \right) = \bar{X}_T^\top \bar{X}_T = \Gamma^\top Z_T^\top Z_T \Gamma.
$$

**Proof by Contradiction.** To begin, recall that

$$
\rho^* := \min_{\lambda \in \Delta(\mathcal{Z})} \max_{w \in \mathcal{W} \setminus \{w^*\}} \frac{\|w^* - w\|_{A(\lambda, \Gamma)^{-1}}^2}{\langle w^* - w, \theta \rangle^2}.
$$

Suppose for the sake of contradiction that the number of samples collected by the non-adaptive oracle algorithm $\mathcal{A}$ is $T \leq \sigma^2 \rho^* \log(1/\delta)/2$ and $\mathbb{P}(\widehat{w} \neq w^*) < \delta$ for $\delta \in (0, 0.05]$. To reach a contradiction, we analyze the distribution of $(w - w^*)^\top \widehat{\theta}$ for some $w \neq w^*$ and show that with probability at least $\delta$ it is positive. We remark that this proof follows similar techniques to that of the proof of Theorem 3 of [22].

Let $\overline{\lambda} \in \Delta(\mathcal{Z})$ represent the empirical sampling distribution of the algorithm $\mathcal{A}$, which is defined such that $\overline{\lambda}_z = \frac{1}{T} \sum_{t=1}^{T} \mathbb{1}\{z_{I_t} = z\}$ for each $z \in \mathcal{Z}$. Moreover, define

$$\rho^*(\overline{\lambda}) := \max_{w \in \mathcal{W} \setminus \{w^*\}} \frac{\|w^* - w\|_{A(\overline{\lambda}, \Gamma)^{-1}}^2}{\langle w^* - w, \theta \rangle^2} \quad \text{and} \quad \widetilde{w} \in \arg\max_{w \in \mathcal{W} \setminus \{w^*\}} \frac{\|w^* - w\|_{A(\overline{\lambda}, \Gamma)^{-1}}^2}{\langle w^* - w, \theta \rangle^2}.$$

Note that $\rho^*(\overline{\lambda}) \geq \rho^*$ and observe by definition,

$$\frac{A(\overline{\lambda}, \Gamma)^{-1}}{T} := \frac{(\sum_{z \in \mathcal{Z}} \overline{\lambda}_z \Gamma^\top z z^\top \Gamma)^{-1}}{T} = \left( \sum_{t=1}^{T} \Gamma^\top z_{I_t} z_{I_t} \Gamma \right)^{-1} := \bar{A}(Z_T, \Gamma)^{-1}.$$

Thus, by Eq. (8),

$$\widehat{\theta}_{\texttt{Oracle}} - \theta \sim \mathcal{N}\left(0, \sigma^2 \cdot \frac{A(\overline{\lambda}, \Gamma)^{-1}}{T}\right),$$

and

$$\frac{(\widetilde{w} - w^*)^\top (\widehat{\theta}_{\texttt{Oracle}} - \theta)}{\langle w^* - \widetilde{w}, \theta \rangle} \sim \mathcal{N}\left(0, \sigma^2 \cdot \frac{\|w^* - \widetilde{w}\|_{A(\overline{\lambda}, \Gamma)^{-1}}^2}{T \cdot \langle w^* - \widetilde{w}, \theta \rangle^2}\right).$$

Furthermore, by the definition of $\rho^*(\overline{\lambda})$, the assumption $T \leq \sigma^2 \rho^* \log(1/\delta)/2$, and the fact $\rho^*(\overline{\lambda}) \geq \rho^*$, we obtain

$$\mathbb{V}\left(\frac{(\widetilde{w} - w^*)^\top (\widehat{\theta}_{\texttt{Oracle}} - \theta)}{\langle w^* - \widetilde{w}, \theta \rangle}\right) = \sigma^2 \cdot \frac{\|w^* - \widetilde{w}\|_{A(\overline{\lambda}, \Gamma)^{-1}}^2}{T \cdot \langle w^* - \widetilde{w}, \theta \rangle^2} := \sigma^2 \cdot \frac{\rho^*(\overline{\lambda})}{T} \geq \frac{2}{\log(1/\delta)}. \tag{9}$$

Now, consider a random variable $W \sim \mathcal{N}(0, 1)$. Proposition 2.1.2 of Vershynin [34] gives an anti-concentration result showing that for all $\zeta > 0$,

$$\mathbb{P}(W \geq \zeta) \geq \left(\frac{1}{\zeta} - \frac{1}{\zeta^3}\right) \frac{1}{\sqrt{2\pi}} e^{-\zeta^2/2}. \tag{10}$$

We apply this result to the quantity $(\widetilde{w} - w^*)^\top (\widehat{\theta}_{\texttt{Oracle}} - \theta)/\langle w^* - \widetilde{w}, \theta \rangle$ to conclude that $(\widetilde{w} - w^*)^\top \widehat{\theta}_{\texttt{Oracle}} > 0$ with probability at least $\delta$. Toward doing so, let $c \in (1, 1.15]$ be a constant and define

$$\widetilde{W} \sim \mathcal{N}\left(0, \frac{2}{\log(1/\delta)}\right) \quad \text{and} \quad W := \frac{\widetilde{W}}{\sqrt{2/\log(1/\delta)}} \quad \text{and} \quad \gamma := \frac{c}{\sqrt{2/\log(1/\delta)}}.$$

Observe that $W \sim \mathcal{N}(0, 1)$. The following analysis holds for $\delta \in (0, 0.05]$ given that $c \in (1, 1.15]$ as assumed:

$$\mathbb{P}\left(\frac{(\widetilde{w} - w^*)^\top (\widehat{\theta}_{\texttt{Oracle}} - \theta)}{\langle w^* - \widetilde{w}, \theta \rangle} \geq c\right) \geq \mathbb{P}(\widetilde{W} \geq c) \qquad \text{(By Eq. 9)}$$

$$= \mathbb{P}\left(\frac{\widetilde{W}}{\sqrt{2/\log(1/\delta)}} \geq \frac{c}{\sqrt{2/\log(1/\delta)}}\right)$$

$$:= \mathbb{P}(W \geq \gamma)$$

$$\geq \left(\frac{1}{\gamma} - \frac{1}{\gamma^3}\right) \frac{1}{\sqrt{2\pi}} e^{-\gamma^2/2} \quad \text{(Proposition 2.1.2 Vershynin 34)}$$

$$= \left(\frac{\sqrt{2}}{c\sqrt{\log(1/\delta)}} - \frac{\sqrt{2}^3}{c^3 \sqrt{\log(1/\delta)}^3}\right) \frac{1}{\sqrt{2\pi}} \delta^{c^2/4}$$

$$\geq \delta.$$

The final inequality can be verified computationally. Thus, with probability at least $\delta$ for $\delta \in (0, 0.05]$, we obtain

$$\begin{aligned} (\widetilde{w} - w^*)^\top \widehat{\theta}_{\texttt{Oracle}} &\geq c(w^* - \widetilde{w})^\top \theta + (\widetilde{w} - w^*)^\top \theta \\ &= c(w^* - \widetilde{w})^\top \theta - (w^* - \widetilde{w})^\top \theta \\ &= (c - 1)(w^* - \widetilde{w})^\top \theta \\ &> 0. \end{aligned}$$

Observe that the final inequality holds since $c > 1$ and $(w^* - \widetilde{w})^\top \theta > 0$ by definition.

This result directly implies that with probability at least $\delta$ for $\delta \in (0, 0.05]$, the vector $\widehat{w}$ returned by algorithm $\mathcal{A}$ is not $w^*$. This is a contradiction, so we conclude that if the non-adaptive oracle algorithm $\mathcal{A}$ acquires $T \leq \sigma^2 \rho^* \log(1/\delta)/2$ samples, then $\mathbb{P}(\widehat{w} \neq w^*) \geq \delta$ for $\delta \in (0, 0.05]$.

$\square$

## F.2 Proof of Corollary D.2

*Corollary D.2.* There exists a problem instance characterized by $\mathcal{W} \subset \mathbb{R}^d$, $\mathcal{Z} \subset \mathbb{R}^d$, $\Gamma \in \mathbb{R}^{d \times d}$, and $\theta \in \mathbb{R}^d$ with a noise process satisfying Assumption 1 such that if the non-adaptive oracle algorithm acquires $T \leq \max\{d\|\theta\|_2^2, \sqrt{d}\|\theta\|_2\}\rho^* \log(1/\delta)/2$ samples, then $\mathbb{P}(\widehat{w} \neq w^*) \geq \delta$ for $\delta \in (0, 0.05]$.

*Proof.* To begin, consider the specifications of Theorem D.1 and its result. That is, a problem an arbitrary instance characterized by $\mathcal{W} \subset \mathbb{R}^d$, $\mathcal{Z} \subset \mathbb{R}^d$, $\Gamma \in \mathbb{R}^{d \times d}$, and $\theta \in \mathbb{R}^d$ where the noise process is jointly Gaussian and defined by $\gamma := \begin{bmatrix} \eta & \varepsilon \end{bmatrix} \sim \mathcal{N}(0, \Sigma)$ where $\Sigma \in \mathbb{R}^{(d+1) \times (d+1)}$ is an arbitrary correlation matrix. Observe that the noise process defined by $\gamma$ satisfies Assumption 1. The result states that if the non-adaptive oracle algorithm acquires $T \leq \sigma^2 \rho^* \log(1/\delta)/2$ samples on the problem instance where $\sigma^2 := v^\top \Sigma v$ and $v := \begin{bmatrix} \theta & 1 \end{bmatrix} \in \mathbb{R}^{d+1}$, then $\mathbb{P}(\widehat{w} \neq w^*) \geq \delta$ for $\delta \in (0, 0.05]$. From this point, we show that there exists a parameter $\theta$ and correlation matrix $\Sigma$ such that $\sigma^2 := v^\top \Sigma v \geq \max\{d\|\theta\|_2^2, \sqrt{d}\|\theta\|_2\}$ in order to reach the stated conclusion.

**Notation.** Let $\zeta_{\eta_i, \varepsilon} \in [-1, 1]$ denote the correlation between $\eta_i$ and $\varepsilon$ for $i \in \{1, \ldots, d\}$. Similarly, let $\zeta_{\eta_i, \eta_j} = \zeta_{\eta_j, \eta_i} \in [-1, 1]$ denote the correlation between $\eta_i$ and $\eta_j$ for $i \neq j \in \{1, \ldots, d\}$. Note that the correlation of $\eta_i$ with itself for $i \in \{1, \ldots, d\}$ is $\sigma_{\eta_i}^2 = \zeta_{\eta_i, \eta_i} = 1$ and similarly the correlation of $\varepsilon$ with itself is $\sigma_\varepsilon^2 = \zeta_{\varepsilon, \varepsilon} = 1$. The correlation matrix $\Sigma$ is then given by

$$\Sigma = \begin{bmatrix} 1 & \zeta_{\eta_2, \eta_1} & \cdots & \zeta_{\eta_d, \eta_1} & \zeta_{\varepsilon, \eta_1} \\ \zeta_{\eta_1, \eta_2} & 1 & \cdots & \zeta_{\eta_d, \eta_2} & \zeta_{\varepsilon, \eta_2} \\ \vdots & \vdots & \ddots & \vdots & \vdots \\ \zeta_{\eta_1, \eta_d} & \zeta_{\eta_2, \eta_d} & \cdots & 1 & \zeta_{\varepsilon, \eta_d} \\ \zeta_{\eta_1, \varepsilon} & \zeta_{\eta_2, \varepsilon} & \cdots & \zeta_{\eta_d, \varepsilon} & 1 \end{bmatrix} := \begin{bmatrix} \Sigma_\theta & \zeta_{\eta, \varepsilon} \\ \zeta_{\eta, \varepsilon}^\top & 1 \end{bmatrix},$$

where

$$\Sigma_\theta = \begin{bmatrix} 1 & \zeta_{\eta_2, \eta_1} & \cdots & \zeta_{\eta_d, \eta_1} \\ \zeta_{\eta_1, \eta_2} & 1 & \cdots & \zeta_{\eta_d, \eta_2} \\ \vdots & \vdots & \ddots & \vdots \\ \zeta_{\eta_1, \eta_d} & \zeta_{\eta_2, \eta_d} & \cdots & 1 \end{bmatrix} \in [-1, 1]^{d \times d} \quad \text{and} \quad \zeta_{\eta, \varepsilon} = \begin{bmatrix} \zeta_{\eta_1, \varepsilon} \\ \zeta_{\eta_2, \varepsilon} \\ \vdots \\ \zeta_{\eta_d, \varepsilon} \end{bmatrix} \in [-1, 1]^d.$$

Moreover, we use the notation $\Sigma_{\theta, i}^\top \in \mathbb{R}^d$ to denote the $i$–th row of $\Sigma_\theta^\top$, or equivalently the $i$–th column of $\Sigma_\theta$, for any $i \in \{1, \ldots, d\}$.

**Lower Bounding Noise Variance.** Given the above notation, we now work toward lower bounding $\sigma^2 := v^\top \Sigma v$. Observe that by algebraic manipulations,

$$v^\top \Sigma v := \begin{bmatrix} \theta \\ 1 \end{bmatrix}^\top \begin{bmatrix} \Sigma_\theta & \zeta_{\eta, \varepsilon} \\ \zeta_{\eta, \varepsilon}^\top & 1 \end{bmatrix} \begin{bmatrix} \theta \\ 1 \end{bmatrix}$$

$$= \begin{bmatrix} \theta^\top \Sigma_{\theta, 1}^\top + \zeta_{\eta_1, \varepsilon} & \theta^\top \Sigma_{\theta, 2}^\top + \zeta_{\eta_2, \varepsilon} & \cdots & \theta^\top \Sigma_{\theta, d}^\top + \zeta_{\eta_d, \varepsilon} & \theta^\top \zeta_{\eta, \varepsilon} + 1 \end{bmatrix} \begin{bmatrix} \theta \\ 1 \end{bmatrix}$$

$$= \theta^\top \sum_{i=1}^{d} \Sigma_{\theta, i}^\top \theta_i + \sum_{i=1}^{d} \zeta_{\eta_i, \varepsilon} \theta_i + \theta^\top \zeta_{\eta, \varepsilon} + 1$$

$$= \theta^\top \Sigma_\theta \theta + 2\theta^\top \zeta_{\eta, \varepsilon} + 1.$$

Since $\Sigma_\theta$ is a real symmetric matrix, an eigendecomposition exists such that $\Sigma_\theta = Q \Lambda Q^\top$ where $\Lambda = \text{diag}(\lambda_1, \ldots, \lambda_d) \in \mathbb{R}^{d \times d}$ is a diagonal matrix containing the eigenvalues of $\Sigma_\theta$ and $Q \in \mathbb{R}^{d \times d}$

is an orthogonal matrix with columns corresponding to the eigenvectors of $\Sigma_\theta$. Let $q_i := Q_i^\top$ denote column $i$ of the matrix $Q$ for $i = \{1, \ldots, d\}$, which is equivalently eigenvector $i$ of $\Sigma_\theta$ for $i = \{1, \ldots, d\}$. Without loss of generality, assume that the eigenvectors are of unit length so that $\|q_i\|_2 = 1$ for all $i = \{1, \ldots, d\}$.

Given this information, suppose that the parameter $\theta$ in the instance is equal to a scalar multiple of the eigenvector of $\Sigma_\theta$ corresponding to the maximum eigenvalue. Note this is equivalent to the statement that $\theta$ is equal to some scalar multiple of the column $q_* \in \mathbb{R}^d$ of the matrix $Q$ where $q_*$ is the eigenvector of $\Sigma_\theta$ corresponding to the maximum eigenvalue $\lambda^*$. Thus, we take $\theta = c \cdot q_*$ for some $c \in \mathbb{R}$ and observe that $\|\theta\|_2 = c$. Toward quantifying the value of $v^\top \Sigma v$ for the problem instance, we begin by characterizing $\theta^\top \Sigma_\theta \theta$ for the choice of $\theta$. Consider the following analysis:

$$
\begin{aligned}
\theta^\top \Sigma_\theta \theta &= \theta^\top Q \Lambda Q^\top \theta \\
&:= (cq_*)^\top Q \Lambda Q^\top (cq_*) && (\theta := cq_*) \\
&= \|\theta\|_2^2 q_*^\top \begin{bmatrix} q_1 & \cdots & q_* & \cdots & q_d \end{bmatrix} \Lambda \begin{bmatrix} q_1 & \cdots & q_* & \cdots & q_d \end{bmatrix}^\top q_* \\
&= \|\theta\|_2^2 \begin{bmatrix} 0 & \cdots & \|q_*\|_2^2 & \cdots & 0 \end{bmatrix} \operatorname{diag}(\lambda_1, \ldots, \lambda^*, \ldots, \lambda_1) \begin{bmatrix} 0 & \cdots & \|q_*\|_2^2 & \cdots & 0 \end{bmatrix}^\top \\
&&& (q_i^\top q_j = 0 \; \forall i \neq j) \\
&= \|\theta\|_2 \lambda^*. && (\|q_*\|_2 = 1)
\end{aligned}
$$

Thus, in general for this choice of $\theta$,

$$
v^\top \Sigma_\theta v = \|\theta\|_2^2 \lambda^* + 2\theta^\top \zeta_{\eta,\varepsilon} + 1.
$$

To conclude, take $\Sigma_\theta := \mathbf{1}_d \mathbf{1}_d^\top$ where $\mathbf{1}_d$ represents the $d$-dimensional vector of all ones. Since this is a rank-1 matrix, the maximum eigenvalue is $\lambda^* = \mathbf{1}_d^\top \mathbf{1}_d = d$ and the remainder of the eigenvalues are zero. Observe that $q_* = \mathbf{1}_d / \sqrt{d}$ is an eigenvector corresponding to the maximum eigenvalue since $\mathbf{1}_d \mathbf{1}_d^\top q_* = dq_*$. Thus,

$$
\begin{aligned}
v^\top \Sigma_\theta v &= \|\theta\|_2^2 \lambda^* + 2\theta^\top \zeta_{\eta,\varepsilon} + 1 \\
&:= \|\theta\|_2^2 \lambda^* + 2\|\theta\|_2 \mathbf{1}_d^\top \mathbf{1}_d / \sqrt{d} + 1 && (\theta := cq_* := \|\theta\|_2 \mathbf{1}_d / \sqrt{d} = \text{ and } \zeta_{\eta,\varepsilon} := \mathbf{1}_d) \\
&= d\|\theta\|_2^2 + 2\sqrt{d}\|\theta\|_2 + 1 \\
&\geq \max\{d\|\theta\|_2^2, \sqrt{d}\|\theta\|_2\}.
\end{aligned}
$$

This completes the proof since we have shown that there exists a parameter $\theta$ and correlation matrix $\Sigma$ such that $\sigma^2 := v^\top \Sigma v \geq \max\{d\|\theta\|_2^2, \sqrt{d}\|\theta\|_2\}$, which by Theorem D.1 allows us to make the stated conclusion. $\qquad \square$

# G   Proofs of the confidence interval

## G.1   Proof of Lemma 2.1

The first statement is an immediate consequence of Lemma G.2 and the second statement is proven in Lemma G.1.

**Lemma G.1.** *In the compliance model, the noise $\eta^\top \theta + \varepsilon$ follows a $\left(8\|\theta\|_2^2 + 2\right)$-sub-Gaussian distribution.*

*Proof.* In compliance, we have $z, x \in \{e_1, \cdots, e_d\}$, and

$$
\eta = x - \left(\mathbb{P}(e_1 \mid z), \cdots, \mathbb{P}(e_d \mid z)\right)^\top.
$$

Let us figure out the sub-Gaussian parameter of the random vector $\eta$. Fix any unit vector $a$. First, we have $\mathbb{E}[\langle \eta, a \rangle] = 0$. Second, we have

$$\left| \eta^\top a \right| \leq \|\eta\|_2 \qquad \qquad \text{(Cauchy–Schwarz inequality)}$$

$$\leq \left\| x - \left( \mathbb{P}(e_1 \mid z), \cdots, \mathbb{P}(e_d \mid z) \right)^\top \right\|_2$$

$$\leq \left( \|x\|_2 + \left\| \left( \mathbb{P}(e_1 \mid z), \cdots, \mathbb{P}(e_d \mid z) \right)^\top \right\|_2 \right)$$

$$\leq \left( 1 + \left\| \left( \mathbb{P}(e_1 \mid z), \cdots, \mathbb{P}(e_d \mid z) \right)^\top \right\|_1 \right) \qquad (x \in \{e_1, \cdots, e_d\} \text{ and } \|x\|_2 \leq \|x\|_1, \forall x)$$

$$= 2.$$

Thus, $\eta^\top a$ is bounded and zero-mean and thus $2^2$-sub-Gaussian. This implies that

$$\forall \beta, \max_{a: \|a\| \leq 1} \mathbb{E}[\exp(\beta \langle \eta, a \rangle)] \leq \exp\left( \frac{\beta^2 2^2}{2} \right).$$

and thus $\eta$ is a $2^2$-sub-Gaussan random vector. Then, $\eta^\top \theta$ is $(2\|\theta\|)^2$-sub-Gaussian.

Using Lemma G.2, we have that $\eta^\top \theta + \varepsilon$ is $2(4\|\theta\|^2 + 1)$-sub-Gaussian.

$\square$

**Lemma G.2.** *Let $A$ and $B$ random variables that are each $\sigma_A^2$- and $\sigma_B^2$-sub-Gaussian but are correlated. Then, $A + B$ is $2(\sigma_A^2 + \sigma_B^2)$-sub-Gaussian.*

*Proof.* By definition of sub-Gaussian, we have for any $\gamma \in \mathbb{R}$,

$$\mathbb{E}\left[ \exp(\gamma(A+B)) \right] = \mathbb{E}\left[ \exp(\gamma A) \exp(\gamma B) \right]$$

$$\leq \sqrt{\mathbb{E}\left[ \exp(2\gamma A) \right]} \sqrt{\mathbb{E}\left[ \exp(2\gamma B) \right]} \qquad \text{(Cauchy-Schwarz)}$$

$$\leq \sqrt{\exp(2\gamma^2 \sigma_A^2)} \sqrt{\exp(2\gamma^2 \sigma_B^2)}$$

$$\leq \exp\left( 2\gamma^2 (\sigma_A^2 + \sigma_B^2) \right).$$

$\square$

### G.2 Proof of Lemma 2.2

*Lemma 2.2.* Suppose that $T$ observations are collected non-adaptively from the structural equation model in Eqs. (1) and $\Gamma \in \mathbb{R}^{d \times d}$ is known. Then, with probability at least $1 - \delta$ for $\delta \in (0, 1)$ and $w \in \mathbb{R}^d$,

$$|w^\top (\widehat{\theta}_{\texttt{oracle}} - \theta)| \leq \sqrt{2\sigma_\nu^2 \|w\|_{\bar{A}(Z_T, \Gamma)^{-1}} \log(2/\delta)}.$$

where $\sigma_\nu^2$ is the sub-Gaussian parameter of the noise process $\nu := \eta^\top \theta + \varepsilon$ as characterized in Lemma 2.1.

*Proof.* Given the knowledge of $\Gamma$, we have the oracle 2SLS estimator

$$\widehat{\theta}_{\texttt{oracle}} = \left( \sum_{t=1}^{T} z_s \left( \Gamma^\top z_s \right)^\top \right)^{-1} \sum_{t=1}^{T} z_s y_t = \left( \sum_{t=1}^{T} z_s z_s^\top \Gamma \right)^{-1} \sum_{t=1}^{T} z_s y_t.$$

Note that

$$y_t = x_t^\top \theta + \varepsilon_t = \left( \Gamma^\top z_t \right)^\top \theta + \eta_t^\top \theta + \varepsilon_t.$$

Denote $\nu_t := \eta_t^\top \theta + \varepsilon_t$. For any $w \in \mathcal{W}$, we have

$$\left\langle \widehat{\theta}_{\texttt{oracle}} - \theta, w \right\rangle = \left\langle \left( \sum_{t=1}^T z_s z_s^\top \Gamma \right)^{-1} \sum_{t=1}^T z_s y_t - \theta, w \right\rangle$$

$$= \left\langle \left( \sum_{t=1}^T z_s z_s^\top \Gamma \right)^{-1} \sum_{t=1}^T z_s \left( z_s^\top \Gamma \theta + \nu_t \right) - \theta, w \right\rangle$$

$$= \left\langle \left( \sum_{t=1}^T z_s z_s^\top \Gamma \right)^{-1} \left( \sum_{t=1}^T z_s z_s^\top \Gamma \theta + \sum_{t=1}^T z_s \nu_t \right) - \theta, w \right\rangle$$

$$= \left\langle \left( \sum_{t=1}^T z_s z_s^\top \Gamma \right)^{-1} \sum_{t=1}^T z_s \nu_t, w \right\rangle$$

$$= \sum_{q=1}^t \left\langle \left( \sum_{t=1}^T z_s z_s^\top \Gamma \right)^{-1} z_q, w \right\rangle \nu_q.$$

By Lemma G.1, the noise $\nu_t$ is $\sigma_\nu^2$-sub-Gaussian, we have

$$\left\langle \left( \sum_{t=1}^T z_s z_s^\top \Gamma \right)^{-1} z_q, w \right\rangle \nu_q$$

is $\left\langle \left( \sum_{t=1}^T z_s z_s^\top \Gamma \right)^{-1} z_q, w \right\rangle^2 \sigma_\nu^2$-sub-Gaussian. Thus

$$\mathbb{P}\left( \left\langle \widehat{\theta}_{\texttt{oracle}} - \theta, w \right\rangle \geq \sqrt{2 \sum_{q=1}^t \left\langle \left( \sum_{t=1}^T z_s z_s^\top \Gamma \right)^{-1} z_q, w \right\rangle^2 \sigma_\nu^2 \log\left(\frac{1}{\delta}\right)} \right) \leq \delta.$$

Concisely,

$$\sum_{q=1}^t \left\langle \left( \sum_{t=1}^T z_s z_s^\top \Gamma \right)^{-1} z_q, w \right\rangle^2 = w^\top \left( \sum_{t=1}^T z_s z_s^\top \Gamma \right)^{-1} \left( \sum_{q=1}^t z_q z_q^\top \right) \left( \left( \sum_{t=1}^T z_s z_s^\top \Gamma \right)^{-1} \right)^\top w$$

$$= w^\top \Gamma^{-1} \left( \left( \sum_{t=1}^T z_s z_s^\top \Gamma \right)^{-1} \right)^\top w$$

$$= w^\top \left( \Gamma^\top \left( \sum_{t=1}^T z_s z_s^\top \right) \Gamma \right)^{-1} w. \tag{11}$$

Thus,

$$\mathbb{P}\left( \left\langle \widehat{\theta}_{\texttt{oracle}} - \theta, w \right\rangle \geq \sqrt{2 w^\top \left( \Gamma^\top \left( \sum_{t=1}^T z_s z_s^\top \right) \Gamma \right)^{-1} w \sigma_\nu^2 \log\left(\frac{1}{\delta}\right)} \right) \leq \delta.$$

We can further write it as

$$\mathbb{P}\left( \left\langle \widehat{\theta}_{\texttt{oracle}} - \theta, w \right\rangle \geq \sqrt{2 \|w\|^2_{\left( \Gamma^\top \left( \sum_{t=1}^T z_s z_s^\top \right) \Gamma \right)^{-1}} \sigma_\nu^2 \log\left(\frac{1}{\delta}\right)} \right) \leq \delta.$$

By taking a union bound over, we have the confidence interval for the absolute value as in the statement of the lemma. $\qquad\square$

### G.3 Proof of Theorem 2.3

*Theorem 2.3.* Suppose that $\widehat{\Gamma}$ is estimated through a design matrix $Z_{T_1} \in \mathbb{R}^{T_1 \times d}$ and $\widehat{\theta}_{\text{P-2SLS}}$ is estimated through a design matrix $Z_{T_2} \in \mathbb{R}^{T_2 \times d}$. Then, for any $w \in \mathcal{W}$, with probability at least $1 - \delta$,

$$|w^\top (\widehat{\theta}_{\text{P-2SLS}} - \theta)| \leq \|w\|_{\bar{A}(Z_{T_2}, \widehat{\Gamma})^{-1}} \sqrt{2\sigma_\nu^2 \log\left(\frac{4}{\delta}\right)} + \|w\|_{\bar{A}(Z_{T_1}, \widehat{\Gamma})^{-1}} \|\theta\|_2 \sqrt{\sigma_\eta^2 \overline{\log}(Z_{T_1}, \delta/4)}.$$

where $\sigma_\nu^2$ is the sub-Gaussian parameter of the noise $\nu := \eta^\top \theta + \varepsilon$, $\sigma_\eta^2$ is the sub-Gaussian parameter of the noise $\eta$, and

$$\overline{\log}(Z_T, \delta) := 8d \ln\left(1 + \frac{2TL_z^2}{d(2 \wedge \sigma_{\min}(Z_{T_1}^\top Z_{T_1}))}\right) + 16 \ln\left(\frac{2 \cdot 6^d}{\delta} \cdot \log_2^2\left(\frac{4}{2 \wedge \sigma_{\min}(Z_{T_1}^\top Z_{T_1})}\right)\right)$$

*Proof.* For the pseudo 2SLS estimator, we have

$$\widehat{\theta}_{\text{P-2SLS}} - \theta = \left(\sum_{t=1}^{T_2} z_{I_t} z_{I_t}^\top \widehat{\Gamma}\right)^{-1} \sum_{t=1}^{T_2} z_{I_t} y_t - \theta$$

$$= \left(\sum_{t=1}^{T_2} z_{I_t} z_{I_t}^\top \widehat{\Gamma}\right)^{-1} \sum_{t=1}^{T_2} z_{I_t}\left(z_{I_t}^\top \Gamma\theta + \nu_t\right) - \theta$$

$$= \left(\sum_{t=1}^{T_2} z_{I_t} z_{I_t}^\top \widehat{\Gamma}\right)^{-1} \left(\sum_{t=1}^{T_2} z_{I_t} z_{I_t}^\top \Gamma\theta + \sum_{t=1}^{T_2} z_{I_t}\nu_t\right) - \theta$$

$$= \left(\sum_{t=1}^{T_2} z_{I_t} z_{I_t}^\top \widehat{\Gamma}\right)^{-1} \sum_{t=1}^{T_2} z_{I_t}\nu_t + \left(\widehat{\Gamma}^{-1}\Gamma - I\right)\theta.$$

For any $w \in \mathcal{W}$, we have

$$\left\langle\widehat{\theta}_{\text{P-2SLS}} - \theta, w\right\rangle = \left\langle\left(\sum_{t=1}^{T_2} z_{I_t} z_{I_t}^\top \widehat{\Gamma}\right)^{-1} \sum_{t=1}^{T_2} z_{I_t}\nu_t + \left(\widehat{\Gamma}^{-1}\Gamma - I\right)\theta, w\right\rangle$$

$$= \left\langle\left(\sum_{t=1}^{T_2} z_{I_t} z_{I_t}^\top \widehat{\Gamma}\right)^{-1} \sum_{t=1}^{T_2} z_{I_t}\nu_t, w\right\rangle + \left\langle\left(\widehat{\Gamma}^{-1}\Gamma - I\right)\theta, w\right\rangle$$

$$= \sum_{q=1}^{T_2}\left\langle\left(\sum_{t=1}^{T_2} z_{I_t} z_{I_t}^\top \widehat{\Gamma}\right)^{-1} z_{I_q}, w\right\rangle\nu_q + \left\langle\left(\widehat{\Gamma}^{-1}\Gamma - I\right)\theta, w\right\rangle$$

$$= \sum_{q=1}^{T_2}\left\langle\left(\sum_{t=1}^{T_2} z_{I_t} z_{I_t}^\top \widehat{\Gamma}\right)^{-1} z_{I_q}, w\right\rangle\nu_q + \left\langle\left(\widehat{\Gamma}^{-1} - \Gamma^{-1}\right)\Gamma\theta, w\right\rangle. \quad (12)$$

We upper bound the first term and the second term separately. For the first term, by Lemma 2.1, we know $\nu_t$ is $\sigma_\nu^2$-subGaussian, we have $\sum_{q=1}^{T_2}\left\langle\left(\sum_{t=1}^{T_2} z_{I_t} z_{I_t}^\top \widehat{\Gamma}\right)^{-1} z_{I_q}, w\right\rangle\nu_q$ is $\sum_{q=1}^{T_2}\left\langle\left(\sum_{t=1}^{T_2} z_{I_t} z_{I_t}^\top \widehat{\Gamma}\right)^{-1} z_{I_q}, w\right\rangle^2 \sigma_\nu^2$-subGaussian. Thus by the concentration inequality of sub-

Gaussian random variables, we have

$$\mathbb{P}\left(\sum_{q=1}^{T_2}\left\langle\left(\sum_{t=1}^{T_2}z_{I_t}z_{I_t}^\top\widehat{\Gamma}\right)^{-1}z_{I_q},w\right\rangle\nu_q\geq\sqrt{2\sum_{q=1}^{T_2}\left\langle\left(\sum_{t=1}^{T_2}z_{I_t}z_{I_t}^\top\widehat{\Gamma}\right)^{-1}z_{I_q},w\right\rangle^2\sigma_\nu^2\log\left(\frac{2}{\delta}\right)}\right)\leq\frac{\delta}{2}.$$

By similar calculation as (11), we have with probability at least $1-\frac{\delta}{2}$,

$$\sum_{q=1}^{T_2}\left\langle\left(\sum_{t=1}^{T_2}z_{I_t}z_{I_t}^\top\widehat{\Gamma}\right)^{-1}z_{I_q},w\right\rangle\nu_q\leq\sqrt{2w^\top\left(\widehat{\Gamma}^\top\left(\sum_{t=1}^{T_2}z_{I_t}z_{I_t}^\top\right)\widehat{\Gamma}\right)^{-1}w\sigma_\nu^2\log\left(\frac{2}{\delta}\right)}$$

$$\leq\sqrt{2\|w\|^2_{\bar{A}(Z_{T_2},\widehat{\Gamma})^{-1}}\sigma_\nu^2\log\left(\frac{2}{\delta}\right)}. \tag{13}$$

Thus with probability at least $1-\frac{\delta}{2}$,

$$\left\langle\widehat{\theta}_{\text{P-2SLS}}-\theta,w\right\rangle\leq\|w\|_{\bar{A}(Z_{T_2},\widehat{\Gamma})^{-1}}\sqrt{2\sigma_\nu^2\log\left(\frac{2}{\delta}\right)}+\left\langle\left(\widehat{\Gamma}^{-1}-\Gamma^{-1}\right)\Gamma\theta,w\right\rangle$$

By Theorem G.3, we have with probability at least $1-\frac{\delta}{2}$,

$$\left\langle\left(\widehat{\Gamma}^{-1}-\Gamma^{-1}\right)\Gamma\theta,w\right\rangle\leq\|w\|_{\bar{A}(Z_{T_1},\widehat{\Gamma})^{-1}}\|\theta\|\sqrt{\sigma_\eta^2\overline{\log}(Z_{T_1},\delta/2)} \tag{14}$$

Combining (13) and (14), we have with probability at least $1-\delta$, for any $w\in\mathcal{W}$,

$$\left\langle\widehat{\theta}_{\text{P-2SLS}}-\theta,w\right\rangle\leq\|w\|_{\bar{A}(Z_{T_2},\widehat{\Gamma})^{-1}}\sqrt{2\sigma_\nu^2\log\left(\frac{2}{\delta}\right)}+\|w\|_{\bar{A}(Z_{T_1},\widehat{\Gamma})^{-1}}\|\theta\|\sqrt{\sigma_\eta^2\overline{\log}(Z_{T_1},\delta/2)}.$$

By a union bound, we have the confidence interval for the absolute value of the inner product,

$$|w^\top(\widehat{\theta}_{\text{P-2SLS}}-\theta)|\leq\|w\|_{\bar{A}(Z_{T_2},\widehat{\Gamma})^{-1}}\sqrt{2\sigma_\nu^2\log\left(\frac{4}{\delta}\right)}+\|w\|_{\bar{A}(Z_{T_1},\widehat{\Gamma})^{-1}}\|\theta\|\sqrt{\sigma_\eta^2\overline{\log}(Z_{T_1},\delta/4)}.$$

$\square$

**Theorem G.3.** *Suppose that the least square estimator $\widehat{\Gamma}$ is estimated through a design matrix $Z_{T_1}\in\mathbb{R}^{T_1\times d}$, then it satisfies, with probability at least $1-\delta$,*

$$\left\langle\left(\widehat{\Gamma}^{-1}-\Gamma^{-1}\right)\Gamma\theta,w\right\rangle\leq\|w\|_{\bar{A}(Z_{T_1},\widehat{\Gamma})^{-1}}\|\theta\|\sqrt{\sigma_\eta^2\overline{\log}(Z_{T_1},\delta)}$$

*for any $w\in\mathcal{W}$.*

*Proof.* Define $V=Z_{T_1}^\top Z_{T_1}$, $S\in\mathbb{R}^{T_1\times d}$ as the matrix with $i$-th row being $\eta_i^\top$, the stacked noise, i.e., the data collection process of the design matrix $Z_{T_1}$ is $X=Z_{T_1}\Gamma+S$. Then we have

$$\left\langle\left(\widehat{\Gamma}^{-1}-\Gamma^{-1}\right)\Gamma\theta,w\right\rangle=w^\top\left(\widehat{\Gamma}^{-1}-\Gamma^{-1}\right)\Gamma\theta$$

$$=w^\top\left(\Gamma+V^{-1}Z_{T_1}^\top S\right)^{-1}V^{-1}Z_{T_1}^\top S\Gamma^{-1}\Gamma\theta \quad\text{(Lemma J.12)}$$

$$=w^\top\widehat{\Gamma}^{-1}V^{-1/2}V^{-1/2}Z_{T_1}^\top S\theta$$

$$\leq\left\|w^\top\widehat{\Gamma}^{-1}V^{-1/2}\right\|\left\|V^{-1/2}Z_{T_1}^\top S\theta\right\|$$

$$\leq\|w\|_{\bar{A}(Z_{T_1},\widehat{\Gamma})^{-1}}\left\|V^{-1/2}Z_{T_1}^\top S\right\|_{\text{op}}\|\theta\|$$

$$\leq\|w\|_{\bar{A}(Z_{T_1},\widehat{\Gamma})^{-1}}\|\theta\|\sqrt{\sigma_\eta^2\overline{\log}(Z_{T_1},\delta)},$$

where the last inequality is due to Lemma G.4. $\square$

**Lemma G.4.** *Suppose we have $z_1, \ldots, z_T \in \mathbb{R}^d$ and $\eta_1, \ldots, \eta_T \in \mathbb{R}^d$ such that $\eta_T \mid z_1, \eta_1, \ldots, z_{T-1}, \eta_{T-1}, z_T$ is $\sigma_\eta^2$-sub-Gaussian vector (defined in Assumption 1). Let $Z, S \in \mathbb{R}^{T \times d}$ be matrices whose t-th row is $z_t^\top$ and $\eta_t^\top$ respectively. Suppose $\|z_t\| \leq L_z, \forall t$. Let $V = Z^\top Z$. Then, $\forall \delta \in (0, 1)$, we have, with probability at least $1 - \delta$,*

$$\left\| V^{-1/2} Z_T^\top S \right\|_{op} \leq \sigma_\eta \sqrt{8d \ln\left(1 + \frac{2TL_z^2}{d(2 \wedge \sigma_{\min}(V))}\right) + 16 \ln\left(\frac{2 \cdot 6^d}{\delta} \cdot \log_2^2\left(\frac{4}{2 \wedge \sigma_{\min}(V)}\right)\right)}.$$

*We abbreviate $\overline{\log}(Z_T, \delta) := 8d \ln\left(1 + \frac{2TL_z^2}{d(2 \wedge \sigma_{\min}(V))}\right) + 16 \ln\left(\frac{2 \cdot 6^d}{\delta} \cdot \log_2^2\left(\frac{4}{2 \wedge \sigma_{\min}(V)}\right)\right).$*

*Proof.* By the definition of operator norm, we have

$$\left\| V^{-1/2} Z_T^\top S \right\|_{op} = \sup_{\{x \mid \|x\|_2 = 1\}} \left\| V^{-1/2} Z_T^\top S x \right\|_2$$

$$= \sup_{\{x \mid \|x\|_2 = 1\}} \sqrt{x^\top S^\top Z_T V^{-1} Z_T^\top S x}$$

$$= \sup_{\{x \mid \|x\|_2 = 1\}} \left\| Z_T^\top S x \right\|_{V^{-1}}.$$

Considering a fixed $w \in \mathbb{R}^d$, by Lemma G.5, we have with probability at least $1 - \delta$,

$$\left\| Z_T^\top S x \right\|_{V^{-1}} \leq \sqrt{2} \sigma_\eta \sqrt{d \ln\left(1 + \frac{2TL_z^2}{d(2 \wedge \sigma_{\min}(V))}\right) + 2 \ln\left(\frac{2}{\delta} \cdot \log_2^2\left(\frac{4}{2 \wedge \sigma_{\min}(V)}\right)\right)}.$$

By Lemma G.6 and a union bound, for the $\varepsilon$-covering $\mathcal{C}_\varepsilon$, we have the following event happens with probability no more than $\delta$:

$$\mathcal{E} := \left\{ \exists x \in \mathcal{C}_\varepsilon, \left\| Z_T^\top S x \right\|_{V^{-1}} \geq \sqrt{2} \sigma_\eta \sqrt{d \ln\left(1 + \frac{2TL_z^2}{d(2 \wedge \sigma_{\min}(V))}\right) + 2 \ln\left(\frac{2|\mathcal{C}_\varepsilon|}{\delta} \cdot \log_2^2\left(\frac{4}{2 \wedge \sigma_{\min}(V)}\right)\right)} \right\}.$$

We abbreviate $\widehat{\log}(Z_T, \delta) := \sqrt{2} \sqrt{d \ln\left(1 + \frac{2TL_z^2}{d(2 \wedge \sigma_{\min}(V))}\right) + 2 \ln\left(\frac{2|\mathcal{C}_\varepsilon|}{\delta} \cdot \log_2^2\left(\frac{4}{2 \wedge \sigma_{\min}(V)}\right)\right)}$

When $\mathcal{E}$ does not happen, we have

$$\left\| V^{-1/2} Z^\top S \right\|_{op} = \sup_{\{x \mid \|x\| = 1\}} \left\| Z_T^\top S x \right\|_{V^{-1}}$$

$$= \sup_{\{x \mid \|x\| = 1\}} \min_{\{y \mid y \in C_\varepsilon\}} \left\| Z_T^\top S(x - y + y) \right\|_{V^{-1}}$$

$$\leq \sup_{\{x \mid \|x\| = 1\}} \min_{\{y \mid y \in C_\varepsilon\}} \left( \left\| Z_T^\top S(x - y) \right\|_{V^{-1}} + \left\| Z_T^\top S y \right\|_{V^{-1}} \right)$$

$$\leq \sup_{\{x \mid \|x\| = 1\}} \min_{\{y \mid y \in C_\varepsilon\}} \left( \left\| V^{-1/2} Z^\top S \right\|_{op} \|x - y\|_2 + \sigma_\eta \widehat{\log}(Z_T, \delta) \right)$$

$$\leq \varepsilon \left\| V^{-1/2} Z_T^\top S \right\|_{op} + \sigma_\eta \widehat{\log}(Z_T, \delta).$$

Thus,

$$\left\| V^{-1/2} Z_T^\top S \right\|_{op} \leq \frac{\sigma_\eta}{1 - \varepsilon} \widehat{\log}(Z_T, \delta).$$

By choosing $\varepsilon = \frac{1}{2}$ and Lemma G.6, we have

$$\left\| V^{-1/2} Z_T^\top S \right\|_{op} \leq \sigma_\eta \sqrt{8d \ln\left(1 + \frac{2TL_z^2}{d(2 \wedge \sigma_{\min}(V))}\right) + 16 \ln\left(\frac{2 \cdot 6^d}{\delta} \cdot \log_2^2\left(\frac{4}{2 \wedge \sigma_{\min}(V)}\right)\right)}.$$

$\square$

**Lemma G.5.** *[Self-Normalized Bound for Vector-Valued Martingales] Suppose we have $z_1, \ldots, z_t \in \mathbb{R}^d$ and $\eta_1, \ldots, \eta_t \in \mathbb{R}^d$ such that $\eta_t \mid z_1, \eta_1, \ldots, z_{t-1}, \eta_{t-1}, z_t$ is $\sigma_\eta^2$-sub-Gaussian vector (defined in Assumption 1). Let $Z, S \in \mathbb{R}^{t \times d}$ be matrices whose $s$-th row is $z_s^\top$ and $\eta_s^\top$ respectively. Suppose $\|z_s\| \le L_z, \forall s$. Let $V_t = Z^\top Z$. Then, $\forall b \in \mathbb{R}^t, \delta \in (0,1)$, we have, with probability at least $1 - \delta$,*

$$\forall t \ge 1, \left\| Z^\top S b \right\|_{V_t^{-1}} \le \sqrt{2} \|b\|_2 \sigma_\eta \sqrt{d \ln \left( 1 + \frac{2t L_z^2}{d(2 \wedge \sigma_{\min}(V_t))} \right) + 2 \ln \left( \frac{2}{\delta} \cdot \log_2^2 \left( \frac{4}{2 \wedge \sigma_{\min}(V_t)} \right) \right)}.$$

*Proof.* Since each row of $S$ is a $\sigma_\eta^2$-subGaussian vector, we have that $(Sb)_i / \|b\|$ is $\sigma_\eta^2$-subGaussian. Using Lemma G.7 with $\varepsilon_s = (Sb)_i / \|b\|$ completes the proof. □

**Lemma G.6.** *[26][Lemma 20.1] There exists a set $\mathcal{C}_\varepsilon \subset \mathbb{R}^d$ with $|\mathcal{C}_\varepsilon| \le \left( \frac{3}{\varepsilon} \right)^d$ such that for any $x \in \{ x \mid x \in \mathbb{R}^d, \|x\|_2 = 1 \}$, there exists a $y \in \mathcal{C}_\varepsilon$ such that $\|x - y\|_2 \le \varepsilon$.*

**Lemma G.7.** *Let $z_1, z_2, \ldots \in \{ z \in \mathbb{R}^d : \|z\|_2 \le L_z \}$ and $\varepsilon_1, \varepsilon_2, \ldots \in \mathbb{R}$ be random variables such that $\varepsilon_k \mid z_1, \varepsilon_1, \ldots, z_{t-1}, \varepsilon_{t-1}, z_t$ is $\sigma_\varepsilon^2$-sub-Gaussian. Let $V_t = \sum_{s=1}^t z_s z_s^\top$. Then,*

$$1 - \delta \le \mathbb{P} \left( \forall t \ge 1, \left\| \sum_{s=1}^t z_s \varepsilon_s \right\|_{V_t^{-1}} \le \sqrt{2} \sigma_\varepsilon \sqrt{d \ln \left( 1 + \frac{2t L_z^2}{d(2 \wedge \sigma_{\min}(V_t))} \right) + 2 \ln \left( \frac{2}{\delta} \cdot \log_2^2 \left( \frac{4}{2 \wedge \sigma_{\min}(V_t)} \right) \right)} \right)$$

*Proof.* Let $Z_t \in \mathbb{R}^{t \times d}$ be the design matrix and define $\varepsilon_t := (\varepsilon_1, \ldots, \varepsilon_t)^\top \in \mathbb{R}^t$. Let us omit the subscript $t$ from $Z_t, \varepsilon_t$ and $V_t$. Note that

$$\|Z^\top \varepsilon\|_{V^{-1}} = \|Z^\top \varepsilon\|_{(\frac{1}{2}V + \frac{1}{2}V)^{-1}} \le \|Z^\top \varepsilon\|_{(\frac{1}{2}V + \frac{1}{2}\sigma_{\min}(V)I)^{-1}} \le \sqrt{2} \|Z^\top \varepsilon\|_{(V + \sigma_{\min}(V)I)^{-1}}$$

It remains to bound $\|Z^\top \varepsilon\|_{(V + \sigma_{\min}(V)I)^{-1}}$. We use union bound with the standard self-normalized inequality of Lattimore and Szepesvári [26][Theorem 20.4 and Note 20.2]. Specifically, let $\lambda_k = 2^{-k+1}$ for $k \ge 1$. Then,

$$1 - \delta \le \mathbb{P} \left( \mathcal{E} := \left\{ \forall k \in \mathbb{N}_+, \forall t \ge 1, \|Z^\top \varepsilon\|_{(V + \lambda_k I)^{-1}} \le \sigma_\varepsilon \sqrt{d \ln \left( 1 + \frac{tL^2}{d\lambda_k} \right) + 2 \ln \left( \frac{(\pi^2/6)k^2}{\delta} \right)} \right\} \right)$$

Under the event $\mathcal{E}$, we have two cases:

- $\sigma_{\min}(V) \ge 1$: choose $k = 1$. Then,
$$\|Z^\top \varepsilon\|_{(V + \sigma_{\min}(V)I)^{-1}} \le \|Z^\top \varepsilon\|_{(V + I)^{-1}}$$
$$\le \sigma_\varepsilon \sqrt{d \ln \left( 1 + \frac{tL^2}{d} \right) + 2 \ln \left( \frac{(\pi^2/6)}{\delta} \right)}$$

- $\sigma_{\min}(V) < 1$: choose $k = \lceil \log_2(2\sigma_{\min}(V)^{-1}) \rceil \ge 2$, which is the $k$ that satisfies $\lambda_k \le \sigma_{\min}(V) < \lambda_{k-1}$. Then,
$$\|Z^\top \varepsilon\|_{(V + \sigma_{\min}(V)I)^{-1}} \le \|Z^\top \varepsilon\|_{(V + \lambda_k I)^{-1}}$$
$$\le \sigma_\varepsilon \sqrt{d \ln \left( 1 + \frac{tL^2}{d\lambda_k} \right) + 2 \ln \left( \frac{(\pi^2/6)k^2}{\delta} \right)}$$
$$\le \sigma_\varepsilon \sqrt{d \ln \left( 1 + \frac{tL^2}{d\sigma_{\min}(V)/2} \right) + 2 \ln \left( \frac{(\pi^2/6)\lceil \log_2(2\sigma_{\min}(V)^{-1}) \rceil^2}{\delta} \right)}.$$
$$(\lambda_k = \tfrac{1}{2}\lambda_{k-1} > \tfrac{1}{2}\sigma_{\min}(V); \text{ def'n of } k)$$

Altogether, we have

$$\|Z^\top \varepsilon\|_{(V + \sigma_{\min}(V)I)^{-1}} \le \sigma_\varepsilon \sqrt{d \ln \left( 1 + \frac{tL^2(1 \vee 2\sigma_{\min}^{-1}(V))}{d} \right) + 2 \ln \left( \frac{(\pi^2/6)(1 \vee \lceil \log_2(2\sigma_{\min}^{-1}(V)) \rceil^2)}{\delta} \right)}.$$

We conclude the proof by simplifying the RHS. □

# H Proofs of sample complexity when given $\Gamma$

## H.1 Proof of Theorem 3.1

---

**Algorithm 5** Confounded pure exploration with known $\Gamma$

---

**Input** $\mathcal{Z}, \mathcal{W}, \Gamma, \delta, \varepsilon, L_\nu \geq \sigma_\nu^2$

**Initialize:** $k = 1, \hat{\mathcal{W}}_k = \mathcal{W}$

**Define** $f(w, w', \Gamma, \lambda) := \|w - w'\|^2_{(\sum_{z \in \mathcal{Z}} \Gamma^\top \lambda_z zz^\top \Gamma)^{-1}}$

**while** $\left|\hat{\mathcal{W}}_k\right| > 1$ **do**

$\quad \lambda_k = \arg\min_{\lambda \in \Delta(\mathcal{Z})} \max_{w,w' \in \mathcal{W}_k} f(w, w', \Gamma, \lambda)$

$\quad \rho(\mathcal{W}_k) = \min_{\lambda \in \Delta(\mathcal{Z})} \max_{w,w' \in \mathcal{W}_k} f(w, w', \Gamma, \lambda)$

$\quad \zeta_k = 2^{-k}$

$\quad N_k = \left\lceil 2(1+\omega)\zeta_k^{-2}\rho(\mathcal{W}_k)L_\nu \log\left(\frac{4k^2|\mathcal{W}|}{\delta}\right) \right\rceil \vee r(\omega)$

$\quad Z_{N_k} = \text{ROUND}(\lambda_k, N_k)$

$\quad$ Pull arms in $Z_{N_k}$ and observe $Y_{N_k}$

$\quad$ Compute $\widehat{\theta}^k_{\texttt{oracle}} = \left(Z_{N_k}^\top Z_{N_k} \Gamma\right)^{-1} Z_{N_k}^\top Y_{N_k}$

$\quad \mathcal{W}_{k+1} = \mathcal{W}_k \backslash \left\{ w \in \mathcal{W}_k \mid \exists w' \in \mathcal{W}_k, \text{s.t.}, \left\langle w' - w, \widehat{\theta}^k_{\texttt{oracle}} \right\rangle > \zeta_k \right\}$

$\quad k = k + 1$

**end while**

**Output:** $\mathcal{W}_k$

---

*Theorem 3.1.* Algorithm 1 is $\delta$-PAC for the CPET-LB problem and terminates in at most $c(1 + \omega)L_\nu \rho^* \log(1/\delta) + cr(\omega)$ samples, where $c$ hides logarithmic factors of $\Delta$ and $|\mathcal{W}|$.

*Proof.* **Part 1 $\delta$-PAC**

By the confidence interval in Lemma 2.2, we have, with probability at least $1 - \frac{\delta}{2k^2|\mathcal{W}|}$,

$$
\begin{aligned}
\left|\left\langle \widehat{\theta}^k_{\texttt{oracle}} - \theta, w - w^* \right\rangle\right| &\leq \|w - w^*\|_{\left(\Gamma^\top\left(\sum_{s=1}^{N_k} z_{I_t} z_{I_t}^\top\right)\Gamma\right)^{-1}} \sqrt{2\sigma_\nu^2 \log\left(\frac{4k^2|\mathcal{W}|}{\delta}\right)} \\
&\leq \frac{\sqrt{1+\omega}\|w - w^*\|_{(\Gamma^\top \lambda_z zz^\top \Gamma)^{-1}}}{\sqrt{N_k}} \sqrt{2\sigma_\nu^2 \log\left(\frac{4k^2|\mathcal{W}|}{\delta}\right)} \\
&\leq \frac{\sqrt{1+\omega}\|w - w^*\|_{(\Gamma^\top \lambda_z zz^\top \Gamma)^{-1}}}{\sqrt{\left\lceil 2(1+\omega)\zeta_k^{-2}\rho(\mathcal{W}_k)L_\nu \log\left(\frac{4k^2|\mathcal{W}|}{\delta}\right)\right\rceil \vee r(\omega)}} \sqrt{2\sigma_\nu^2 \log\left(\frac{4k^2|\mathcal{W}|}{\delta}\right)} \\
&\leq \zeta_k.
\end{aligned}
$$

Define a good event $\mathcal{E}_{k,w}$ for each $k$ and $\mathcal{W}_k$ as

$$
\mathcal{E}_{k,w} = \left\{ \left|\left\langle \widehat{\theta}^k_{\texttt{oracle}} - \theta, w - w^* \right\rangle\right| \leq \zeta_k \right\}.
$$

We claim that with probability at least $1 - \delta$, the event $\bigcap_{k=1}^{\infty} \bigcap_{w \in \mathcal{W}_k} \mathcal{E}_{k,w}$ holds. It can be proved by a union bound.

$$
\begin{aligned}
\mathbb{P}\left(\left(\bigcap_{k=1}^{\infty} \bigcap_{w \in \mathcal{W}_k} \mathcal{E}_{w,k}\right)^c\right) &\leq \sum_{k=1}^{\infty} \sum_{w \in \mathcal{W}_k} \mathbb{P}\left(\mathcal{E}_{k,w}^c\right) \\
&\leq \sum_{k=1}^{\infty} \sum_{w \in \mathcal{W}_k} \frac{\delta}{2k^2|\mathcal{W}|} \\
&\leq \frac{\delta}{2} \sum_{k=1}^{\infty} \frac{1}{k^2} \\
&\leq \delta,
\end{aligned}
\tag{15}
$$

where the last step is by the fact that $\sum_{k=1}^{\infty} \frac{1}{k^2} = \frac{\pi^2}{6} < 2$. Under the event $\bigcap_{k=1}^{\infty} \bigcap_{w \in \mathcal{W}_k} \mathcal{E}_{w,k}$, to show that the best arm is never eliminated, it suffices to show that for any sub-optimal arm $w \in \mathcal{W}_k$,

$$
\begin{aligned}
\left\langle w - w^*, \widehat{\theta}_{\texttt{oracle}}^k \right\rangle &= \left\langle w - w^*, \widehat{\theta}_{\texttt{oracle}}^k - \theta \right\rangle + \langle w - w^*, \theta \rangle \\
&\leq \left\langle w - w^*, \widehat{\theta}_{\texttt{oracle}}^k - \theta \right\rangle \\
&\leq \zeta_k.
\end{aligned}
$$

Thus the best arm $w^*$ never satisfies the elimination condition. Next we show that at the end of stage $k$, any suboptimal arm $w$ that satisfies

$$
\langle \theta, w^* - w \rangle \geq 2\zeta_k
$$

is eliminated. To show this, we need to show that $w$ satisfies the elimination condition,

$$
\begin{aligned}
\max_{w' \in \mathcal{W}_k} \left\langle \widehat{\theta}_{\texttt{oracle}}^k, w' - w \right\rangle &\geq \left\langle \widehat{\theta}_{\texttt{oracle}}^k, w^* - w \right\rangle \\
&= \left\langle \widehat{\theta}_{\texttt{oracle}}^k - \theta, w^* - w \right\rangle + \langle \theta, w^* - w \rangle \\
&\geq -\zeta_k + 2\zeta_k \\
&= \zeta_k.
\end{aligned}
$$

This implies that with probability at least $1 - \delta$, $w^*$ always survives.

**Part 2 sample complexity**

Define $S_k = \{w \in \mathcal{W} \mid \langle w^* - w, \theta \rangle \leq 4\zeta_k\}$. Thus with probability at least $1 - \delta$, we have $\bigcap_k \{\mathcal{W}_k \subseteq S_k\}$. This implies the following is true with probability at least $1 - \delta$ for all $k$

$$
\begin{aligned}
\rho(\mathcal{W}_k) &= \min_{\lambda \in \Delta(\mathcal{Z})} \max_{w,w' \in \mathcal{W}_k} \|w - w'\|_{(\sum_{z \in \mathcal{Z}} \Gamma^\top \lambda_z z z^\top \Gamma)^{-1}}^2 \\
&\leq \min_{\lambda \in \Delta(\mathcal{Z})} \max_{w,w' \in S_k} \|w - w'\|_{(\sum_{z \in \mathcal{Z}} \Gamma^\top \lambda_z z z^\top \Gamma)^{-1}}^2 \\
&= \rho(S_k).
\end{aligned}
$$

Define $\Delta$ to be the minimum gap between $w^*$ and any other $w \in \mathcal{W}$, i.e., $\Delta := \min_{w \in \mathcal{W} \setminus \{w^*\}} \langle w^* - w, \theta \rangle$. Then for $k \geq \lceil \log(4\Delta^{-1}) \rceil$, we have $S_k = \{w^*\}$ with probability

at least $1 - \delta$. The total sample complexity is the summation of the number of samples in each round,

$$\sum_{k=1}^{\lceil \log(4\Delta^{-1}) \rceil} N_k$$

$$= \sum_{k=1}^{\lceil \log(4\Delta^{-1}) \rceil} \left( \left\lceil 4(1+\omega)\zeta_k^{-2}\rho(\mathcal{W}_k)L_\nu \log\left(\frac{4k^2|\mathcal{W}|}{\delta}\right) \right\rceil \vee r(\omega) \right)$$

$$\leq \sum_{k=1}^{\lceil \log(4\Delta^{-1}) \rceil} \left( 8(1+\omega)2^{2k}\rho(\mathcal{W}_k)L_\nu \log\left(\frac{4k^2|\mathcal{W}|}{\delta}\right) \vee r(\omega) \right)$$

$$\leq \sum_{k=1}^{\lceil \log(4\Delta^{-1}) \rceil} \left( 8(1+\omega)2^{2k}\rho(\mathcal{S}_k)L_\nu \log\left(\frac{4k^2|\mathcal{W}|}{\delta}\right) \vee r(\omega) \right). \tag{16}$$

On the other hand, we have

$$\rho^* = \min_{\lambda \in \Delta(\mathcal{Z})} \max_{w \in \mathcal{W}\backslash\{w^*\}} \frac{\|w^* - w\|^2_{(\sum_{z \in \mathcal{Z}} \Gamma^\top \lambda_z zz^\top \Gamma)^{-1}}}{\langle w^* - w, \theta \rangle^2}$$

$$= \min_{\lambda \in \Delta(\mathcal{Z})} \max_k \max_{w \in S_k} \frac{\|w^* - w\|^2_{(\sum_{z \in \mathcal{Z}} \Gamma^\top \lambda_z zz^\top \Gamma)^{-1}}}{\langle w^* - w, \theta \rangle^2}$$

$$\geq \frac{1}{\lceil \log(4\Delta^{-1}) \rceil} \min_{\lambda \in \Delta(\mathcal{Z})} \sum_{k=1}^{\lceil \log(4\Delta^{-1}) \rceil} \max_{w \in S_k} \frac{\|w^* - w\|^2_{(\sum_{z \in \mathcal{Z}} \Gamma^\top \lambda_z zz^\top \Gamma)^{-1}}}{\langle w^* - w, \theta \rangle^2}$$

$$\geq \frac{1}{16\lceil \log(4\Delta^{-1}) \rceil} \sum_{k=1}^{\lceil \log(4\Delta^{-1}) \rceil} 2^{2k} \min_{\lambda \in \Delta(\mathcal{Z})} \max_{w \in S_k} \|w^* - w\|^2_{(\sum_{z \in \mathcal{Z}} \Gamma^\top \lambda_z zz^\top \Gamma)^{-1}}$$

$$\geq \frac{1}{64\lceil \log(4\Delta^{-1}) \rceil} \sum_{k=1}^{\lceil \log(4\Delta^{-1}) \rceil} 2^{2k} \min_{\lambda \in \Delta(\mathcal{Z})} \max_{w,w' \in S_k} \|w - w'\|^2_{(\sum_{z \in \mathcal{Z}} \Gamma^\top \lambda_z zz^\top \Gamma)^{-1}}$$

$$= \frac{1}{64\lceil \log(4\Delta^{-1}) \rceil} \sum_{k=1}^{\lceil \log(4\Delta^{-1}) \rceil} 2^{2k}\rho(S_k), \tag{17}$$

where the last inequality is by the triangle inequality, i.e.,

$$\max_{w,w' \in S_k} \|w - w'\|^2_{(\sum_{z \in \mathcal{Z}} \Gamma^\top \lambda_z zz^\top \Gamma)^{-1}} \leq \max_{w,w' \in S_k} \left( \|w - w^*\|_{(\sum_{z \in \mathcal{Z}} \Gamma^\top \lambda_z zz^\top \Gamma)^{-1}} + \|w^* - w'\|_{(\sum_{z \in \mathcal{Z}} \Gamma^\top \lambda_z zz^\top \Gamma)^{-1}} \right)^2$$

$$\leq 4 \max_{w \in S_k} \|w - w^*\|^2_{(\sum_{z \in \mathcal{Z}} \Gamma^\top \lambda_z zz^\top \Gamma)^{-1}}.$$

Combining (16) and (17), we have

$$\sum_{k=1}^{\lceil \log(4\Delta^{-1}) \rceil} N_k \leq c(1+\omega)L_\nu \log\left(4\Delta^{-1}\right) \log\left(\frac{\log\left(4\Delta^{-1}\right)^2|\mathcal{W}|}{\delta}\right)\rho^* + \log\left(4\Delta^{-1}\right)r(\omega).$$

$$\square$$

# I Proofs of sample complexity when given $\widehat{\Gamma}$

---

**Algorithm 6** Confounded pure exploration with known $\widehat{\Gamma}$

---

**Input** $\mathcal{Z}, \mathcal{W}, \widehat{\Gamma}, \delta, \varepsilon, L_\nu \geq \sigma_\nu^2, L_\eta \geq \sigma_\eta^2$

**Initialize:** $k = 1, \hat{\mathcal{W}}_k = \mathcal{W}$, calculate $\gamma$ using (18)

**Define** $f(w, w', \Gamma, \lambda) := \|w - w'\|_{(\sum_{z \in \mathcal{Z}} \Gamma^\top \lambda_z zz^\top \Gamma)^{-1}}^2$

**while** $\exists w, w' \in \mathcal{W}_k$, s.t., $\left\langle w' - w, \widehat{\theta}_{\mathtt{oracle}}^k \right\rangle > 4\gamma$ or $\zeta_k > \gamma$ **do**

  $\lambda_k = \arg \min_{\lambda \in \Delta(\mathcal{Z})} \max_{w, w' \in \mathcal{W}_k} f(w, w', \widehat{\Gamma}, \lambda)$

  $\rho(\mathcal{W}_k) = \min_{\lambda \in \Delta(\mathcal{Z})} \max_{w, w' \in \mathcal{W}_k} f(w, w', \widehat{\Gamma}, \lambda)$

  $\zeta_k = 2^{-k}$

  $N_k = \left\lceil 2(1 + \omega)\left(\zeta_k^{-2} \wedge \frac{1}{\gamma^2}\right) \rho(\mathcal{W}_k) L_\nu \log\left(\frac{4k^2 |\mathcal{W}|}{\delta}\right) \right\rceil \vee r(\omega)$

  $Z_{N_k} = \mathrm{ROUND}(\lambda_k, N_k)$

  Pull arms in $Z_{N_k}$ and observe $Y_{N_k}$

  Compute $\widehat{\theta}_{\mathtt{P\text{-}2SLS}}^k = \left(Z_{N_k}^\top Z_{N_k} \widehat{\Gamma}\right)^{-1} Z_{N_k}^\top Y_{N_k}$

  $\mathcal{W}_{k+1} = \mathcal{W}_k \backslash \left\{ w \in \mathcal{W}_k \mid \exists w' \in \mathcal{W}_k, \text{s.t.}, \left\langle w' - w, \widehat{\theta}_{\mathtt{P\text{-}2SLS}}^k \right\rangle > \zeta_k + \gamma \right\}$

  $k = k + 1$

**end while**

**Output:** any $w \in \mathcal{W}_k$

---

**Theorem I.1.** *Suppose that we have $\widehat{\Gamma}$ that is an OLS estimate from an offline dataset $\{Z_T, X_T\}$ collected non-adaptively through a fixed design $\xi$ and the efficient rounding procedure ROUND, as well as $T \geq \frac{4\sigma_\eta^2}{\sigma_{\min}(A(\xi, \Gamma))} \overline{\log}(Z_T, \delta/2)$. Algorithm 6 guarantees that with probability at least $1 - \delta$, a $6\gamma$-good arm is returned, where*

$$\gamma := \max_{w, w' \in \mathcal{W}} \|w - w'\|_{\bar{A}(Z_T, \widehat{\Gamma})^{-1}} \|\theta\|_2 \sqrt{L_\eta \overline{\log}(Z_T, \delta)}. \tag{18}$$

*Also, the algorithm terminates in at most*

$$c(1 + \omega) L_\nu \log(1/\delta) \rho^*(\gamma) + c(1 + \omega)^2 L_\nu \log(1/\delta) \frac{\sigma_\eta^2 \overline{\log}(Z_T, \delta)}{\sigma_{\min}(A(\xi, \Gamma))} \frac{\rho(\xi, \gamma)}{T} \vee cr(\omega)$$

*samples, where $c$ hides logarithmic factors of $\Delta$ and $|\mathcal{W}|$ and $\gamma$.*

$$\rho(\lambda, \gamma) := \max_{w \in \mathcal{W} \backslash \{w^*\}} \frac{\|w^* - w\|_{A(\lambda, \Gamma)^{-1}}^2}{\langle w^* - w, \theta \rangle^2 \vee \gamma^2}$$

*and*

$$\rho^*(\gamma) = \min_{\lambda \in \Delta(\mathcal{Z})} \max_{w \in \mathcal{W} \backslash \{w^*\}} \frac{\|w^* - w\|_{A(\lambda, \Gamma)^{-1}}^2}{\langle w^* - w, \theta \rangle^2 \vee \gamma^2}.$$

*Proof.* The proof can be divided into four steps:

- The best arm $w^*$ is never eliminated.

- At the end of stage $k$, any suboptimal arm $w$ that satisfies $\langle \theta, w^* - w \rangle \geq 2\zeta_k + 2\gamma$ is eliminated.

- The stopping condition is met in finite time.

- When the stopping condition is met, there are only $6\gamma$-good arms left.

- The upper bound of the sample complexity.

**Step 1: The best arm $w^*$ is never eliminated.**

By the confidence interval in Theorem 2.3, we have, with probability at least $1 - \frac{\delta}{2k^2|\mathcal{W}|}$, for any $k$ and $w \in \mathcal{W}_k$,

$$
\begin{aligned}
\left| \left\langle \widehat{\theta}^k_{\text{P-2SLS}} - \theta, w - w^* \right\rangle \right| &\leq \|w - w^*\|_{\bar{A}(Z_{N_k}, \widehat{\Gamma})^{-1}} \sqrt{2\sigma_\nu^2 \log\left(\frac{4k^2|\mathcal{W}|}{\delta}\right)} + \gamma \\
&\leq \frac{\sqrt{1+\omega}\|w - w^*\|_{A(\lambda_k, \widehat{\Gamma})^{-1}}}{\sqrt{N_k}} \sqrt{2\sigma_\nu^2 \log\left(\frac{4k^2|\mathcal{W}|}{\delta}\right)} + \gamma \\
&\leq \frac{\sqrt{1+\omega}\|w - w^*\|_{A(\lambda_k, \widehat{\Gamma})^{-1}}}{\sqrt{\left\lceil 2(1+\omega)\zeta_k^{-2}\rho(\mathcal{W}_k)L_\nu \log\left(\frac{4k^2|\mathcal{W}|}{\delta}\right)\right\rceil \vee r(\omega)}} \sqrt{2\sigma_\nu^2 \log\left(\frac{4k^2|\mathcal{W}|}{\delta}\right)} + \gamma \\
&\leq \zeta_k + \gamma.
\end{aligned}
$$

Define a good event $\mathcal{E}_{k,w}$ for each $k$ and $w \in \mathcal{W}_k$ as

$$
\mathcal{E}_{k,w} = \left\{ \left| \left\langle \widehat{\theta}^k_{\text{P-2SLS}} - \theta, w - w^* \right\rangle \right| \leq \zeta_k + \gamma \right\}.
$$

By the same calculation as (15), we claim that with probability at least $1 - \delta$, the event $\bigcap_{k=1}^{\infty} \bigcap_{w \in \mathcal{W}_k} \mathcal{E}_{k,w}$ holds. Under the event $\bigcap_{k=1}^{\infty} \bigcap_{w \in \mathcal{W}_k} \mathcal{E}_{w,k}$, to show that the best arm is never eliminated, it suffices to show that for any sub-optimal arm $w \in \mathcal{W}_k$,

$$
\begin{aligned}
\left\langle w - w^*, \widehat{\theta}^k_{\text{P-2SLS}} \right\rangle &= \left\langle w - w^*, \widehat{\theta}^k_{\text{P-2SLS}} - \theta \right\rangle + \langle w - w^*, \theta \rangle \\
&\leq \left\langle w - w^*, \widehat{\theta}^k_{\text{P-2SLS}} - \theta \right\rangle \\
&\leq \zeta_k + \gamma.
\end{aligned}
$$

Thus the best arm $w^*$ never satisfies the elimination condition.

**Step 2: At the end of stage $k$, any suboptimal arm $w$ that satisfies $\langle \theta, w^* - w \rangle \geq 2\zeta_k + 2\gamma$ is eliminated.**

To prove this, we show that such arm $w$ must satisfy the elimination condition,

$$
\begin{aligned}
\max_{w' \in \mathcal{W}_k} \left\langle \widehat{\theta}^k_{\text{P-2SLS}}, w' - w \right\rangle &\geq \left\langle \widehat{\theta}^k_{\text{P-2SLS}}, w^* - w \right\rangle \\
&= \left\langle \widehat{\theta}^k_{\text{P-2SLS}} - \theta, w^* - w \right\rangle + \langle \theta, w^* - w \rangle \\
&\geq -\zeta_k - \gamma + 2\zeta_k + 2\gamma \\
&= \zeta_k + \gamma.
\end{aligned}
$$

Thus the arm $w$ is eliminated.

**Step 3: The stopping condition is met in finite time.**

Given the result in Step 2 and the fact that $\zeta_k$ is an exponentially decreasing sequence, we know that all of the arms $w$ satisfying $\langle \theta, w^* - w \rangle > 2\gamma$ will be eliminated in finite time. This means that only the arms $w$ satisfying $\langle \theta, w^* - w \rangle \leq 2\gamma$ will remain. We need to show that $\forall w, w' \in \mathcal{W}_k, \left\langle w' - w, \widehat{\theta}^k_{\text{P-2SLS}} \right\rangle \leq 4\gamma$ can be achieved in finite time. When $\zeta_k \leq \gamma$, (which will happen in

finite time),

$$
\begin{aligned}
\left\langle w' - w, \widehat{\theta}^k_{\text{P-2SLS}} \right\rangle =& \left\langle w' - w + w^* - w^*, \widehat{\theta}^k_{\text{P-2SLS}} \right\rangle \\
=& \left\langle w^* - w, \widehat{\theta}^k_{\text{P-2SLS}} \right\rangle + \left\langle w' - w^*, \widehat{\theta}^k_{\text{P-2SLS}} \right\rangle \\
=& \left\langle w^* - w, \widehat{\theta}^k_{\text{P-2SLS}} - \theta + \theta \right\rangle + \left\langle w' - w^*, \widehat{\theta}^k_{\text{P-2SLS}} - \theta + \theta \right\rangle \\
=& \left\langle w^* - w, \widehat{\theta}^k_{\text{P-2SLS}} - \theta \right\rangle + \left\langle w^* - w, \theta \right\rangle + \left\langle w' - w^*, \widehat{\theta}^k_{\text{P-2SLS}} - \theta \right\rangle + \left\langle w' - w^*, \theta \right\rangle \\
\leq& \zeta_k + \gamma + 2\gamma + \zeta_k + \gamma \\
=& 4\gamma.
\end{aligned}
$$

**Step 4: When the stopping condition is met, there are only $6\gamma$-good arms left.**

For any $w \in \mathcal{W}_k$, we have

$$
\begin{aligned}
\langle w^* - w, \theta \rangle =& \left\langle w^* - w, \theta - \widehat{\theta}^k_{\text{P-2SLS}} \right\rangle + \left\langle w^* - w, \widehat{\theta}^k_{\text{P-2SLS}} \right\rangle \\
\leq& \zeta_k + \gamma + 4\gamma \\
=& 6\gamma.
\end{aligned}
$$

**Step 5: The upper bound of the sample complexity.**

Define $\mathcal{W}(2\gamma)$ as the set of $2\gamma$-good arms, i.e., $\mathcal{W}(2\gamma) := \{w \in \mathcal{W} \mid \langle \theta, w^* - w \rangle \leq 2\gamma\}$. Then the best arm in the set $\mathcal{W}\backslash\mathcal{W}(2\gamma)$ has a suboptimality gap $\min_{w\in\mathcal{W}\backslash\mathcal{W}(2\gamma)}\langle \theta, w^* - w \rangle$. We define $\Delta_{\min}(2\gamma) := \min_{w\in\mathcal{W}\backslash\mathcal{W}(2\gamma)}\langle \theta, w^* - w \rangle - 2\gamma$. By the result in Step 2, we know that after $k^* := \left\lceil \log\left(4\Delta_{\min}(2\gamma)^{-1}\right) \right\rceil$ stages, all of the arms in $\mathcal{W}\backslash\mathcal{W}(2\gamma)$ are eliminated. Define $S_k = \{w \in \mathcal{W} \mid \langle w^* - w, \theta \rangle \leq 4\zeta_k + 2\gamma\}$. Thus with probability at least $1 - \delta$, we have $\bigcap_k \{\mathcal{W}_k \subseteq S_k\}$.

The sample complexity of the algorithm is the total number of samples pulled, which is

$$
\begin{aligned}
& \sum_{k=1}^{k^*} N_k \\
=& \sum_{k=1}^{k^*} \left( \left\lceil 2(1+\omega)\left(\zeta_k^{-2} \wedge \frac{1}{\gamma^2}\right)\rho(\mathcal{W}_k)L_\nu \log\left(\frac{4k^2|\mathcal{W}|}{\delta}\right) \right\rceil \vee r(\omega) \right) \\
\leq& \sum_{k=1}^{k^*} \left( 3(1+\omega)\left(2^{2k} \wedge \frac{1}{\gamma^2}\right)\rho(\mathcal{W}_k)L_\nu \log\left(\frac{4k^2|\mathcal{W}|}{\delta}\right) \vee r(\omega) \right) \\
\leq& \sum_{k=1}^{k^*} \left( 3(1+\omega)\left(2^{2k} \wedge \frac{1}{\gamma^2}\right)\rho(S_k)L_\nu \log\left(\frac{4k^2|\mathcal{W}|}{\delta}\right) \vee r(\omega) \right). \quad (19)
\end{aligned}
$$

Note that the factor of $\min_{\lambda\in\Delta(\mathcal{Z})} \max_{w,w'\in S_k} \|w - w'\|^2_{A(\lambda,\Gamma)^{-1}}$ has the underlying true $\Gamma$ in it, while the algorithm uses the plugged-in $\widehat{\Gamma}$. We need to relate it to $\rho(S_k) := \min_{\lambda\in\Delta(\mathcal{Z})} \max_{w,w'\in S_k} \|w - w'\|^2_{A(\lambda,\widehat{\Gamma})^{-1}}$. By Lemma J.9, and defining $\lambda_z^*(S_k)$ as the optimal design for $S_k$, i.e.,

$$
\lambda_k^* = \arg\min_{\lambda\in\Delta(\mathcal{Z})} \max_{w,w'\in S_k} \|w - w'\|^2_{A(\lambda,\Gamma)^{-1}}.
$$

Define $\lambda'_k = \frac{1}{2}\lambda^*_k + \frac{1}{2}\xi$, we have

$$\max_{w,w'\in W_k}\big\|w-w'\big\|^2_{A(\lambda_k,\widehat{\Gamma})^{-1}}$$

$$=\min_{\lambda\in\Delta(\mathcal{Z})}\max_{w,w'\in W_k}\big\|w-w'\big\|^2_{A(\lambda,\widehat{\Gamma})^{-1}}$$

$$\leq\min_{\lambda\in\Delta(\mathcal{Z})}\max_{w,w'\in S_k}\big\|w-w'\big\|^2_{A(\lambda,\widehat{\Gamma})^{-1}}$$

$$\leq\min_{\lambda\in\Delta(\mathcal{Z})}\max_{w,w'\in S_k}3\big\|w-w'\big\|^2_{A(\lambda,\Gamma)^{-1}}+2\max_{w,w'\in S_k}\big\|(\Gamma^{-\top}-\widehat{\Gamma}^{-\top})(w-w')\big\|^2_{A(\lambda,I)^{-1}}$$

$$\leq\max_{w,w'\in S_k}3\big\|w-w'\big\|^2_{A(\lambda'_k,\Gamma)^{-1}}+2\max_{w,w'\in S_k}\big\|(\Gamma^{-\top}-\widehat{\Gamma}^{-\top})(w-w')\big\|^2_{A(\lambda'_k,I)^{-1}}$$

$$\leq\max_{w,w'\in S_k}6\big\|w-w'\big\|^2_{A(\lambda^*_k,\Gamma)^{-1}}+\frac{16\sigma^2_\eta\overline{\log}(Z_T,\delta/2)}{\sigma_{\min}(A(\xi,\Gamma))}\max_{w,w'\in S_k}\big\|w-w'\big\|^2_{A(\xi,\Gamma)^{-1}}\frac{1}{T}, \qquad (20)$$

where the last inequality is due to

$$\big\|w-w'\big\|^2_{A(\lambda'_k,\Gamma)^{-1}}\leq\big\|w-w'\big\|^2_{A\left(\frac{1}{2}\lambda^*_k+\frac{1}{2}\xi,\Gamma\right)^{-1}}$$

$$\leq\big\|w-w'\big\|^2_{A\left(\frac{1}{2}\lambda^*_k,\Gamma\right)^{-1}}$$

$$=2\big\|w-w'\big\|^2_{A(\lambda^*_k,\Gamma)^{-1}},$$

as well as, by Lemma J.11, when $T\geq\frac{4\sigma^2_\eta}{\sigma_{\min}(A(\xi,\Gamma))}\overline{\log}(Z_T,\delta/2)$, we have, with probability at least $1-\delta/2$, we have

$$\max_{w,w'\in S_k}\big\|(\Gamma^{-\top}-\widehat{\Gamma}^{-\top})(w-w')\big\|^2_{A(\lambda'_k,I)^{-1}}$$

$$\leq\frac{4\sigma^2_\eta\overline{\log}(Z_T,\delta/2)}{\sigma_{\min}(A(\lambda'_k,\Gamma))}\max_{w,w'\in S_k}\big\|w-w'\big\|^2_{A(\xi,\Gamma)^{-1}}\frac{1}{T}$$

$$\leq\frac{4\sigma^2_\eta\overline{\log}(Z_T,\delta/2)}{\sigma_{\min}(A(\alpha\lambda^*_k+(1-\alpha)\xi,\Gamma))}\max_{w,w'\in S_k}\big\|w-w'\big\|^2_{A(\xi,\Gamma)^{-1}}\frac{1}{T}$$

$$\leq\frac{8\sigma^2_\eta\overline{\log}(Z_T,\delta/2)}{\sigma_{\min}(A(\xi,\Gamma))}\max_{w,w'\in S_k}\big\|w-w'\big\|^2_{A(\xi,\Gamma)^{-1}}\frac{1}{T}.$$

We can lower bound $\rho^*(\gamma)$ by

$$\rho^*(\gamma)=\min_{\lambda\in\Delta(\mathcal{Z})}\max_{w\in\mathcal{W}\setminus\{w^*\}}\frac{\|w^*-w\|^2_{A(\lambda,\Gamma)^{-1}}}{\langle w^*-w,\theta\rangle^2\vee\gamma^2}$$

$$=\min_{\lambda\in\Delta(\mathcal{Z})}\max_k\max_{w\in S_k\setminus\{w^*\}}\frac{\|w^*-w\|^2_{A(\lambda,\Gamma)^{-1}}}{\langle w^*-w,\theta\rangle^2\vee\gamma^2}$$

$$\geq\frac{1}{k^*}\min_{\lambda\in\Delta(\mathcal{Z})}\sum_{k=1}^{k^*}\max_{w\in S_k\setminus\{w^*\}}\frac{\|w^*-w\|^2_{A(\lambda,\Gamma)^{-1}}}{\langle w^*-w,\theta\rangle^2\vee\gamma^2}$$

$$\geq\frac{1}{16k^*}\sum_{k=1}^{k^*}\left(2^{2k}\wedge\frac{1}{\gamma^2}\right)\min_{\lambda\in\Delta(\mathcal{Z})}\max_{w\in S_k\setminus\{w^*\}}\|w^*-w\|^2_{A(\lambda,\Gamma)^{-1}}$$

$$\geq\frac{1}{64k^*}\sum_{k=1}^{k^*}\left(2^{2k}\wedge\frac{1}{\gamma^2}\right)\min_{\lambda\in\Delta(\mathcal{Z})}\max_{w,w'\in S_k}\|w-w'\|^2_{A(\lambda,\Gamma)^{-1}}. \qquad (21)$$

Given (19), (20) and (21), we have

$$\sum_{k=1}^{k^*}N_k\leq c(1+\omega)L_\nu\log(1/\delta)\rho^*(\gamma)+c(1+\omega)^2L_\nu\log(1/\delta)\frac{\sigma^2_\eta\overline{\log}(Z_T,\delta)}{\sigma_{\min}(A(\xi,\Gamma))}\frac{\rho(\xi,\gamma)}{T}+cr(\omega),$$

where $c$ hides logarithmic factors of $\Delta$ and $|\mathcal{W}|$ and $\gamma$. $\qquad\square$

## J  Proofs of sample complexity with unknown $\Gamma$

---

**Algorithm 7** Optimal design with unknown $\Gamma$

---

**Input** $\mathcal{Z}, \mathcal{W}, \delta, L_\nu \geq \sigma_\nu^2, L_\eta \geq \sigma_\eta^2, \omega, \gamma_{\min} \leq \lambda_{\min}(\Gamma), \lambda_E, \kappa_0$

**Initialize:** $k = 1, \mathcal{W}_1 = \mathcal{W}, \widehat{\Gamma}_0 = \perp, \zeta_1 = 1$

**Define** $f(w, w', \Gamma, \lambda) := \|w - w'\|^2_{\left(\sum_{z \in \mathcal{Z}} \Gamma^\top \lambda_z z z^\top \Gamma\right)^{-1}}, M := \frac{32 L_\eta}{\gamma_{\min}^2 \sigma_{\min}\left(A(\lambda_E, I)\right)} \vee 1, \delta_{k,\ell} :=$
$\frac{\delta}{4k^2 \ell^2}$

**while** $|\mathcal{W}_k| > 1$ **do**

  $\widehat{\Gamma}_k = \Gamma - \mathtt{estimator}\left(\mathcal{W}_k, \widehat{\Gamma}_{k-1}, \zeta_k, \delta, k, \omega, \lambda_E, M, L_\eta\right)$  ▷ Step 1: update $\widehat{\Gamma}$

  $\widehat{\theta}^k_{\mathtt{P\text{-}2SLS}} = \theta - \mathtt{estimator}\left(\mathcal{W}_k, \delta, \zeta_k, \widehat{\Gamma}_k, \omega, L_\nu, k\right)$  ▷ Step 2: update $\widehat{\theta}$

  $\mathcal{W}_{k+1} = \mathcal{W}_k \backslash \left\{w \in \mathcal{W}_k \mid \exists w' \in \mathcal{W}_k, \text{s.t.}, \left\langle w' - w, \widehat{\theta}^k_{\mathtt{P\text{-}2SLS}}\right\rangle > \zeta_k\right\}$  ▷ Step 3: elimination

  $k \leftarrow k + 1, \zeta_k = 2^{-k}$

**end while**

**Output:** $\mathcal{W}_k$

---

The algorithms of $\Gamma - \mathtt{estimator}$ and $\theta - \mathtt{estimator}$ we present below are slightly different from the one in the main text. In the main text, we omit the phase index $k$ in the algorithm for simplicity.

---

**Algorithm 8** $\Gamma - \mathtt{estimator}$

---

**Input** $\mathcal{W}_k, \widehat{\Gamma}_{k-1}, \zeta_k, \delta, k, \omega, \lambda_E, M, L_\eta$

**Define** $\mathtt{Stop}(\mathcal{W}, Z, \Gamma, \delta) := \max_{w, w' \in \mathcal{W}} \|w - w'\|_{\bar{A}(Z, \Gamma)^{-1}} \|\theta\|_2 \sqrt{L_\eta \overline{\log}(Z, \delta)}$

**Initialize** $\ell = 1, N_{k,0,0} = 0$  ▷ doubling trick initialization

**if** $k = 1$ **then**

  **while** $\ell = 1$ or $\mathtt{Stop}\left(\mathcal{W}_k, Z_{k,0,\ell}, \widehat{\Gamma}_k, \delta_{k,\ell}\right) > 1$ **do**

    get $2^{\ell-1}\left(r(\omega) \vee \frac{2}{\kappa_0}\right)$ samples denoted as $\left\{Z_{k,0,\ell}, X_{k,0,\ell}, Y_{k,0,\ell}\right\}$ per design $\lambda_E$ ▷ via ROUND

    Update $\widehat{\Gamma}_k$ by OLS on $\left\{Z_{k,0,\ell}, X_{k,0,\ell}\right\}, \ell \leftarrow \ell + 1$

  **end while**

**else**

  $\tilde{\lambda}_k = \arg\min_{\lambda \in \Delta(\mathcal{Z})} \max_{w, w' \in \mathcal{W}_k} f(w, w', \widehat{\Gamma}_{k-1}, \lambda)$

  $N' = \left\lfloor 4gdM \ln\left(1 + 2M\left(d + L_z^2\right) + 2M2gdM\right) + 8M \ln\left(\frac{2 \cdot 6^d}{\delta_{k,1}}\right) \vee r(\omega) \right\rfloor$

  **while** $\ell = 1$ or $\mathtt{Stop}\left(\mathcal{W}_k, Z_{k,0,\ell} \cup Z_{k,1,\ell}, \widehat{\Gamma}_k, \delta_{k,\ell}\right) > \zeta_k$ **do**

    $N_{k,1,\ell} = 2^\ell N'$  ▷ doubling trick update

    get $N_{k,1,\ell}$ samples per $\tilde{\lambda}_k$ denoted as $\left\{Z_{k,1,\ell}, X_{k,1,\ell}, Y_{k,1,\ell}\right\}$  ▷ via ROUND

    $N_{k,0,\ell} = \left\lceil 2gdM \ln\left(M\left(d + N_{k,1,\ell} + L_z^2\right)\right) + 4M \ln\left(\frac{2 \cdot 6^d}{\delta_{k,\ell}}\right) \vee r(\omega) \vee \frac{2}{\kappa_0} \right\rceil$

    get $(N_{k,0,\ell} - N_{k,0,\ell-1})$ samples per $\lambda_E$ augmented to $\left\{Z_{k,0,\ell-1}, X_{k,0,\ell-1}\right\}$ and get $\left\{Z_{k,0,\ell}, X_{k,0,\ell}\right\}$

    Update $\widehat{\Gamma}_k$ by OLS on $\left\{Z_{k,0,\ell} \cup Z_{k,1,\ell}, X_{k,0,\ell} \cup X_{k,1,\ell}\right\}, \ell \leftarrow \ell + 1$

  **end while**

**end if**

**Output:** $\widehat{\Gamma}_k$

---

---

**Algorithm 9** $\theta - \texttt{estimator}$

---

**Input** $\mathcal{W}_k, \delta, \zeta_k, \widehat{\Gamma}_k, \omega, L_\nu, k$

$\widehat{\lambda}_k = \arg\min_{\lambda \in \Delta(\mathcal{Z})} \max_{w,w' \in \mathcal{W}_k} f(w, w', \widehat{\Gamma}_k, \lambda)$

$\rho(\mathcal{W}_k) = \min_{\lambda \in \Delta(\mathcal{Z})} \max_{w,w' \in \mathcal{W}_k} f(w, w', \widehat{\Gamma}_k, \lambda)$

$N_{k,2} = \left\lceil 2(1+\omega)\zeta_k^{-2}\rho(\mathcal{W}_k)L_\nu \log\left(\frac{4k^2|\mathcal{W}|}{\delta}\right) \right\rceil \vee r(\omega)$

get $N_{k,2}$ samples per design $\widehat{\lambda}_k$ denoted as $\left\{ Z_{k,2}, X_{k,2}, Y_{k,2} \right\}$     $\triangleright$ via ROUND

update $\widehat{\theta}^k_{\texttt{P-2SLS}} = (\widehat{\Gamma}_k^\top Z_{k,2}^\top Z_{k,2} \widehat{\Gamma}_k)^{-1} \widehat{\Gamma}_k^\top Z_{k,2}^\top Y_{k,2}$

**Output:** $\widehat{\theta}^k_{\texttt{P-2SLS}}$

---

**Lemma J.1.** *Algorithm 3 and* $N_{k,1,\ell}, N_{k,0,\ell}$ *guarantees three properties:*

- *Property 1:* $N_{k,1,\ell} \geq N_{k,0,\ell}$.

- *Property 2:* $\frac{1}{2}\left(N_{k,0,\ell} + N_{k,1,\ell}\right) \leq N_{k,0,\ell-1} + N_{k,1,\ell-1}$.

- *Property 3:* $\dfrac{N_{k,1,1}}{8d\ln\left(1 + \frac{N_{k,1,1}L_z^2}{d}\right) + 16\ln\left(\frac{2\cdot 6^d k2}{\delta}\right)} \leq M\ln(dM)$.

*Proof.* For Property 1, recall that $N_{k,0,\ell}$ and $N_{k,1,\ell}$ are defined as

$$N_{k,0,\ell} = \left\lceil 2gdM\ln\left(M\left(d + N_{k,1,\ell} + L_z^2\right)\right) + 4M\ln\left(\frac{2\cdot 6^d}{\delta_{k,\ell}}\right) \vee r(\omega) \right\rceil$$

and

$$N_{k,1,\ell} = 2^\ell \left\lceil 4gdM\ln\left(1 + 2M\left(d + L_z^2\right) + 2M2gdM\right) + 8M\ln\left(\frac{2\cdot 6^d}{\delta_{k,1}}\right) \vee r(\omega) \right\rceil.$$

We prove the result by induction. For the result for $\ell = 1$, note that $N_{k,1,1}$ is of the form $N_{k,1,1} = 2a + 2b\ln(1 + 2c + 2bd)$ and $N_{k,0,1}$ is of the form $N_{k,0,1} = a + b\ln(c + dr)$, where

- $a = 4M\ln\left(\frac{2\cdot 6^d}{\delta_{k,1}}\right)$

- $b = 2gdM$

- $c = M\left(d + L_z^2\right)$

- $d = M$

- $r = N_{k,1,\ell}$.

By the contraposition of Lemma K.5, we have

$$r > 2a + 2b\ln(1 + 2c + 2bd) \implies r > a + b\ln(c + dr).$$

Thus we have $N_{k,1,1} \geq N_{k,0,1}$. Now we assume the result holds for $\ell$, i.e., $N_{k,1,\ell} \geq N_{k,0,\ell}$ and prove that it holds for $\ell + 1$. We have

$$N_{k,1,\ell+1} = 2N_{k,1,\ell} \geq 2N_{k,0,\ell}.$$

It suffices to prove that

$$2N_{k,0,\ell} \geq N_{k,0,\ell+1}.$$

We have

$$2N_{k,0,\ell}$$

$$=2\left[2gdM\ln\left(M\left(d+N_{k,1,\ell}+L_z^2\right)\right)+4M\ln\left(\frac{2\cdot 6^d}{\delta_{k,\ell}}\right)\vee r(\omega)\right]$$

$$\geq 2gdM\ln\left(M^2\left(d+N_{k,1,\ell}+L_z^2\right)^2\right)+8M\ln\left(\frac{2\cdot 6^d\ell^2}{\delta_{k,1}}\right)\vee r(\omega)$$

$$\geq 2gdM\ln\left(M^2\left(d+N_{k,1,\ell}+L_z^2\right)^2\right)+8M\left(\ln\left(\frac{2\cdot 6^d}{\delta_{k,1}}\right)+2\ln(\ell)\right)\vee r(\omega)$$

$$\geq 2gdM\ln\left(M^2\left(d+N_{k,1,\ell}+L_z^2\right)^2\right)+4M\left(\ln\left(\frac{2\cdot 6^d}{\delta_{k,1}}\right)+2\ln(\ell+1)\right)\vee r(\omega)$$

$$(2\ln\ell\geq\ln(\ell+1)\text{ for }\ell\geq 2)$$

$$\geq 2gdM\ln\left(M\left(d+2N_{k,1,\ell}+L_z^2\right)\right)+4M\ln\left(\frac{2\cdot 6^d(\ell+1)^2}{\delta_{k,1}}\right)\vee r(\omega)$$

$$=N_{k,0,\ell+1}.$$

For Property 2, it suffices to prove that

$$\frac{1}{2}N_{k,0,\ell}\leq N_{k,0,\ell-1}.$$

This is equivalent to prove that

$$\frac{1}{2}\big(a+b\ln(c+2dr)\big)\leq a+b\ln(c+dr).$$

We have

$$\frac{1}{2}\big(a+b\ln(c+2dr)\big)\leq a+b\ln(c+dr)$$

$$\iff \frac{1}{2}b\ln(c+2dr)\leq\frac{1}{2}a+b\ln(c+dr)$$

$$\iff b\ln\left(\sqrt{c+2dr}\right)\leq\frac{1}{2}a+b\ln(c+dr)$$

$$\impliedby \sqrt{c+2dr}\leq c+dr$$

$$\iff c+2dr\leq c^2+2cdr+d^2r^2$$

$$\iff dr(dr+2c-2)+c^2-c\geq 0.$$

The last inequality holds because $c=M>1$.

For Property 3, we have,

$$\frac{4dM\ln\left(1+2M\left(d+L_z^2\right)+2MdM\right)+8M\ln\left(\frac{2\cdot 6^dk^2}{\delta}\right)}{8d\ln\left(1+\frac{N_{k,1,1}L_z^2}{d}\right)+16\ln\left(\frac{2\cdot 6^dk^2}{\delta}\right)}$$

$$\leq\frac{4dM\ln\left(1+2M\left(d+L_z^2\right)+2MdM\right)}{8d\ln\left(1+\frac{N_{k,1,1}L_z^2}{d}\right)+16\ln\left(\frac{2\cdot 6^dk^2}{\delta}\right)}+\frac{8M\ln\left(\frac{2\cdot 6^dk^2}{\delta}\right)}{8d\ln\left(1+\frac{N_{k,1,1}L_z^2}{d}\right)+16\ln\left(\frac{2\cdot 6^dk^2}{\delta}\right)}$$

$$\leq\frac{M\ln\left(1+2M\left(d+L_z^2\right)+2MdM\right)}{2\ln\left(1+ML_z^2\right)}+\frac{M}{2}\qquad\text{(loosely apply }N_{k,1,1}\geq dM)$$

$$\leq M\ln(dM).$$

$$\square$$

**Theorem J.2.** *Algorithm 3 guarantees that with probability at least $1 - \delta$, the best arm is returned, and the algorithm terminates in at most*

$$(1 + \omega)\left(\left(L_\nu \log(1/\delta) + L_\eta \|\theta\|_2^2 \big(d + \log(1/\delta)\big)\right)\rho^* + \big(d + \log(1/\delta)\big)\left(L_\eta \|\theta\|_2^2 \rho_0 + M\right)\right)$$

*pulls, ignoring both of the additive and multiplicative logarithms of $\Delta, |\mathcal{W}|, \rho^*, \rho_0, M$, where*

$$\rho^* = \min_{\lambda \in \Delta(\mathcal{Z})} \max_{w \in \mathcal{W} \backslash \{w^*\}} \frac{\|w^* - w\|^2_{(\sum_{z \in \mathcal{Z}} \lambda_z \Gamma^\top z z^\top \Gamma)^{-1}}}{\langle w^* - w, \theta \rangle^2},$$

*and*

$$\rho_0 = \max_{w \in \mathcal{W} \backslash \{w^*\}} \|w^* - w\|^2_{(\sum_{z \in \mathcal{Z}} \lambda_{E,z} \Gamma^\top z z^\top \Gamma)^{-1}},$$

*and*

$$M = \frac{32 L_\eta}{\gamma^2_{\min} \sigma_{\min}\big(A(\lambda_E, I)\big)} \vee 1.$$

*Note that $\rho_0$ does not get hurt by $\langle w^* - w, \theta \rangle$. It comes from the fact that in the first phase, we initialize that algorithm with E-optimal design.*

*Proof.* **Part 1: correctness of the algorithm**

The idea of the proof is similar to the proof of Theorem 3.1.

Recall that the confidence interval of P-2SLS can be break down into two terms (12).

$$\left\langle \widehat{\theta}_{\text{P-2SLS}} - \theta, w \right\rangle = \sum_{q=1}^{T_2} \left\langle \left(\sum_{t=1}^{T_2} z_{I_t} z_{I_t}^\top \widehat{\Gamma}\right)^{-1} z_{I_q}, w \right\rangle \nu_q + \left\langle \left(\widehat{\Gamma}^{-1} - \Gamma^{-1}\right) \Gamma \theta, w \right\rangle.$$

Given $\widehat{\Gamma}$, for a $w \in \mathcal{W}$, with probability at least $1 - \frac{\delta}{2}$ the first term satisfies

$$\sum_{q=1}^{T_2} \left\langle \left(\sum_{t=1}^{T_2} z_{I_t} z_{I_t}^\top \widehat{\Gamma}\right)^{-1} z_{I_q}, w \right\rangle \nu_q \leq \|w\|_{\bar{A}(Z_{T_2}, \widehat{\Gamma})^{-1}} \sqrt{2\sigma_\nu^2 \log\left(\frac{4}{\delta}\right)}.$$

For any $w \in \mathcal{W}$, with probability at least $1 - \frac{\delta}{2}$, the second term satisfies

$$\left\langle \left(\widehat{\Gamma}^{-1} - \Gamma^{-1}\right) \Gamma \theta, w \right\rangle \leq \|w\|_{\bar{A}(Z_{T_1}, \widehat{\Gamma})^{-1}} \|\theta\|_2 \sqrt{\sigma_\eta^2 \overline{\log}(Z_{T_1}, \delta/4)}.$$

Note that by Lemma G.4, the above inequality holds for all $w \in \mathcal{W}$, and the RHS is essentially a result of

$$\left\| V^{-1/2} Z_T^\top S \right\|_{\text{op}} \leq \sqrt{\sigma_\eta^2 \overline{\log}(Z_T, \delta/4)}.$$

In the vanilla form of the confidence of P-2SLS, we can define good events as

- for the first term, for any $w \in \mathcal{W}$,

$$\sum_{q=1}^{T_2} \left\langle \left(\sum_{t=1}^{T_2} z_{I_t} z_{I_t}^\top \widehat{\Gamma}\right)^{-1} z_{I_q}, w \right\rangle \nu_q \leq \|w\|_{\bar{A}(Z_{T_2}, \widehat{\Gamma})^{-1}} \sqrt{2\sigma_\nu^2 \log\big(16k^2 |\mathcal{W}|/\delta\big)}.$$

- for the second term, $\left\| V^{-1/2} Z_T^\top S \right\|_{\text{op}} \leq \sqrt{\sigma_\eta^2 \overline{\log}(Z_T, \delta/4)}.$

For our algorithm design, since we use the doubling trick for the first sub-phase, we need to define the good event for the first sub-phase as the samples from each doubling trick iteration satisfies the self-normalized concentration inequality of Lemma G.4.

We define the good event for $\ell$-th doubling trick iteration in the first sub-phase of phase $k$ as

$$\mathcal{E}_{k,\ell}^1 = \left\{ \left\| V_{k,\ell}^{-1/2} Z_{k,\ell}^\top S_{k,\ell} \right\|_{\text{op}}^2 \leq \sigma_\eta^2 \overline{\log}\left( Z_{k,\ell}, \frac{\delta}{4k^2(\ell+1)^2} \right) \right\},$$

where $Z_{k,\ell} = Z_{k,0,\ell} \cup Z_{k,1,\ell}$, $V_{k,\ell} = Z_{k,\ell}^\top Z_{k,\ell}$ and $S_{k,\ell}$ is stacked noise matrix during collecting samples $Z_{k,\ell}$ per the model $X = Z\Gamma + S$. By a union bound, we have

$$\mathbb{P}\left( \left( \bigcap_{k=1}^\infty \bigcap_{\ell=1}^\infty \mathcal{E}_{k,\ell}^1 \right)^c \right) \leq \sum_{k=1}^\infty \sum_{\ell=1}^\infty \mathbb{P}\left( \mathcal{E}_{k,\ell}^1 \right) \leq \delta/2.$$

For the second sub-phase in phase $k$, we define the good event for the second sub-phase in phase $k$ and $w \in \mathcal{W}$ as

$$\mathcal{E}_{k,w}^2 = \left\{ \sum_{q=1}^{N_{k,2}} \left\langle \left( \sum_{t=1}^{N_{k,2}} z_{I_t} z_{I_t}^\top \widehat{\Gamma} \right)^{-1} z_{I_q}, w \right\rangle \nu_q \leq \|w\|_{\bar{A}(Z_{k,2},\widehat{\Gamma})^{-1}} \sqrt{2\sigma_\nu^2 \log\left( 16k^2 |\mathcal{W}|/\delta \right)} \right\}.$$

By a union bound, we have

$$\mathbb{P}\left( \left( \bigcap_{k=1}^\infty \bigcap_{w \in \mathcal{W}_k} \mathcal{E}_{k,w}^2 \right)^c \right) \leq \sum_{k=1}^\infty \sum_{w \in \mathcal{W}} \mathbb{P}\left( \mathcal{E}_{k,w}^2 \right) \leq \delta/2.$$

Under the good event $\left( \bigcap_{k=1}^\infty \bigcap_{\ell=1}^\infty \mathcal{E}_{k,\ell}^1 \right) \cap \left( \bigcap_{k=1}^\infty \bigcap_{w \in \mathcal{W}_k} \mathcal{E}_{k,w}^2 \right)$, we have with probability at least $1 - \delta$, for all $k$ and $w \in \mathcal{W}$,

$$\left| \left\langle \widehat{\theta}_{\text{P-2SLS}}^k - \theta, w - w^* \right\rangle \right| \leq \zeta_k.$$

The rest of proof is same as the proof of Theorem 3.1.

**Part 2: sample complexity of algorithm**

**Sample complexity for first sub-phase**

Recall that $\lambda_E = \arg\max_{\lambda \in \Delta(\mathcal{Z})} \sigma_{\min}\left( \sum_{z \in \mathcal{Z}} \lambda_z zz^\top \right)$ is the E-optimal design to maximize the minimum singular value of $\sum_{z \in \mathcal{Z}} \lambda_z zz^\top$ and $\kappa_0 = \max_\lambda \sigma_{\min}(\sum_{z \in \mathcal{Z}} \lambda_z zz^\top)$ is the maximum minimum singular value of $\sum_{z \in \mathcal{Z}} \lambda_z zz^\top$. At the beginning of first sub-phase in phase $k$, the algorithm first samples $N_{k,0,0}$ arms according to $\lambda_E$.

Before we proceed to the main proof of the sample complexity, we first address a minor technique issue to avoid cumbersomeness. For the logarithmic term that appears in the algorithm and confidence interval,

$$\overline{\log}(Z_T, \delta) := 8d \ln\left( 1 + \frac{2TL_z^2}{d(2 \wedge \sigma_{\min}(Z_T^\top Z_T))} \right) + 16 \ln\left( \frac{2 \cdot 6^d}{\delta} \cdot \log_2^2\left( \frac{4}{2 \wedge \sigma_{\min}(Z_T^\top Z_T)} \right) \right),$$

when $2 \wedge \sigma_{\min}(V) = 2$, it is equivalent to

$$\widetilde{\log}(T, \delta) := 8d \ln\left( 1 + \frac{TL_z^2}{d} \right) + 16 \ln\left( \frac{2 \cdot 6^d}{\delta} \right).$$

Our algorithm design guarantees that $2 \wedge \sigma_{\min}(V) = 2$ is always true whenever we need to use the logarithmic term $\overline{\log}(Z_T, \delta)$, given that the samples of our interest always includes the E-optimal

design samples $Z_{k,0,\ell}$ and the number of samples from the E-optimal design $\left|Z_{k,0,\ell}\right|$ is always larger than $\frac{2}{\kappa_0}$. So for the remaining part of the proof, we will use $\widetilde{\log}(N,\delta)$ instead of $\overline{\log}(Z_T,\delta)$.

Denote the samples of E-optimal design that mixed into the samples from $\ell$-th doubling trick iteration in the first sub-phase of phase $k$ as $Z_{k,0,\ell}$ and $\left|Z_{k,0,\ell}\right| = N_{k,0,\ell}$. By Lemma J.3, our choice of $N_{k,0,\ell}$

$$N_{k,0,\ell} \geq \frac{64gd\sigma_\eta^2}{\sigma_{\min}\big(A(\lambda_E,\Gamma)\big)}\ln\left(\frac{32g\sigma_\eta^2}{\sigma_{\min}\big(A(\lambda_E,\Gamma)\big)}\Big(d+N_{k,1,\ell}L_z^2+L_z^2\Big)\right)+\frac{128g\sigma_\eta^2}{\sigma_{\min}\big(A(\lambda_E,\Gamma)\big)}\ln\left(\frac{2\cdot 6^d}{\delta}\right),$$

is a sufficient condition to guarantee

$$N_{k,0,\ell} \geq \frac{4g\sigma_\eta^2\widetilde{\log}\big(N_{k,0,\ell}+N_{k,1,\ell},\delta_{k,\ell}\big)}{\sigma_{\min}\big(A(\lambda_E,\Gamma)\big)}\vee r(\omega). \tag{22}$$

Multiply both sides of (22) by $\frac{N_{k,0,\ell}+N_{k,1,\ell}}{N_{k,0,\ell}}$ and we have

$$N_{k,0,\ell}+N_{k,1,\ell} \geq \left(\frac{4g\sigma_\eta^2\widetilde{\log}\big(N_{k,0,\ell}+N_{k,1,\ell},\delta_{k,\ell}\big)}{\sigma_{\min}\big(A(\lambda_E,\Gamma)\big)}\vee r(\omega)\right)\frac{N_{k,0,\ell}+N_{k,1,\ell}}{N_{k,0,\ell}}.$$

By Property 1 of Lemma J.1, we have $\alpha_{k,\ell} := \frac{N_{k,0,\ell}}{N_{k,0,\ell}+N_{k,1,\ell}} < 1/2$, then

$$\begin{aligned}
N_{k,0,\ell}+N_{k,1,\ell} &\geq \frac{1}{\alpha_{k,\ell}}\left(\frac{4g\sigma_\eta^2\widetilde{\log}\big(N_{k,0,\ell}+N_{k,1,\ell},\delta_{k,\ell}\big)}{\sigma_{\min}\big(A(\lambda_E,\Gamma)\big)}\vee r(\omega)\right)\\
&= \frac{4g\sigma_\eta^2\widetilde{\log}\big(N_{k,0,\ell}+N_{k,1,\ell},\delta_{k,\ell}\big)}{\sigma_{\min}\big(A(\alpha_{k,\ell}\lambda_E,\Gamma)\big)}\vee\frac{r(\omega)}{\alpha_{k,\ell}}\\
&\geq \frac{4g\sigma_\eta^2\widetilde{\log}\big(N_{k,0,\ell}+N_{k,1,\ell},\delta_{k,\ell}\big)}{\sigma_{\min}\big(A(\alpha_{k,\ell}\lambda_E+(1-\alpha_{k,\ell})\tilde{\lambda}_k,\Gamma)\big)}\vee r(\omega). \tag{23}
\end{aligned}$$

These condition on $N_{k,0,\ell}$ and $N_{k,0,\ell}+N_{k,1,\ell}$ are needed for the proof.

Denote the total number of doubling trick iterations as $L_k$ for phase $k$. In the case of $L_k = 1$, the samples from the first doubling trick iteration satisfies stopping condition of the first sub-phase already, and the algorithm will not enter the second doubling trick iteration. Thus the total number of samples for the first sub-phase is

$$\begin{aligned}
N_{k,0,1}+N_{k,1,1} &\leq 2N_{k,1,1} && \text{(Property 1 of Lemma J.1)}\\
&\leq 8gdM\ln\left(1+2M\Big(d+L_z^2\Big)+2M2gdM\right)+16M\ln\left(\frac{2\cdot 6^d}{\delta_{k,1}}\right)\vee r(\omega).
\end{aligned}$$

In the case of $L_k > 1$, for $\ell \in \{1,\cdots,L_k\}$, denote $\widehat{\Gamma}^\ell$ as the estimate of $\Gamma$ at the end of the $\ell$-th doubling trick iteration. With these notations, we have

- At the end of $L_k$-th doubling trick iteration, the stopping condition is satisfied, i.e.,

$$\max_{w,w'\in\mathcal{W}_k}\left\|w-w'\right\|^2_{\bar{A}(Z_{k,0,L_k}\cup Z_{k,1,L_k},\widehat{\Gamma}^{L_k})^{-1}}\|\theta\|_2^2 L_\eta\widetilde{\log}\big(N_{k,0,L_k}+N_{k,1,L_k},\delta_{k,L_k}\big) \leq \zeta_k^2. \tag{24}$$

  We short $\widetilde{\log}\big(N_{k,0,\ell}+N_{k,1,\ell},\delta_{k,\ell}\big) = \widetilde{\log}_{k,\ell}$.

- At the end of $(L_k-1)$-th doubling trick iteration, the stopping condition is not satisfied, i.e.,

$$\max_{w,w'\in\mathcal{W}_k}\left\|w-w'\right\|^2_{\bar{A}(Z_{k,0,L_k-1}\cup Z_{k,1,L_k-1},\widehat{\Gamma}^{L_k-1})^{-1}}\|\theta\|_2^2 L_\eta\widetilde{\log}_{k,L_k-1} > \zeta_k^2. \tag{25}$$

Denote $\xi_{L_k}$ as the empirical distribution of $Z_{k,0,L_k} \cup Z_{k,1,L_k}$. Then above two conditions imply that the number of samples for $L_k$-th and $(L_k - 1)$-th doubling trick iterations respectively satisfy

$$N_{k,0,L_k} + N_{k,1,L_k} \geq \frac{\|\theta\|_2^2 L_\eta \widetilde{\log}_{k,L_k}}{\zeta_k^2} \max_{w,w' \in \mathcal{W}_k} \|w - w'\|_{A(\xi_{L_k}, \widehat{\Gamma}^{L_k})^{-1}}^2. \tag{26}$$

and

$$N_{k,0,L_k-1} + N_{k,1,L_k-1} < \frac{\|\theta\|_2^2 L_\eta \widetilde{\log}_{k,L_k-1}}{\zeta_k^2} \max_{w,w' \in \mathcal{W}_k} \|w - w'\|_{A(\xi_{L_k-1}, \widehat{\Gamma}^{L_k-1})^{-1}}^2. \tag{27}$$

Note that by Property 2 of Lemma J.1,

$$\frac{1}{2}\left(N_{k,0,L_k} + N_{k,1,L_k}\right) \leq N_{k,0,L_k-1} + N_{k,1,L_k-1}.$$

Thus

$$N_{k,0,L_k} + N_{k,1,L_k} < \frac{2\|\theta\|_2^2 L_\eta \widetilde{\log}_{k,L_k-1}}{\zeta_k^2} \max_{w,w' \in \mathcal{W}_k} \|w - w'\|_{A(\xi_{L_k-1}, \widehat{\Gamma}^{L_k-1})^{-1}}^2. \tag{28}$$

For any $\ell \in \{1, \ldots, L_k\}$, by Lemma J.7, the factor of $\max_{w,w' \in \mathcal{W}_k} \|w - w'\|_{A(\xi_\ell, \widehat{\Gamma}^\ell)^{-1}}^2$ can be upper bounded by

$$\max_{w,w' \in \mathcal{W}_k} \|w - w'\|_{A(\xi_\ell, \widehat{\Gamma}^\ell)^{-1}}^2$$

$$\leq 3 \max_{w,w' \in \mathcal{W}_k} \|w - w'\|_{A(\xi_\ell, \widehat{\Gamma}^{L_k-1})^{-1}}^2 + 2 \max_{w,w' \in \mathcal{W}_k} \left\|\left((\widehat{\Gamma}^\ell)^{-\top} - (\widehat{\Gamma}^{L_k-1})^{-\top}\right)(w - w')\right\|_{\left(\sum_z \xi_\ell zz^\top\right)^{-1}}^2$$

$$\leq 3 \max_{w,w' \in \mathcal{W}_k} \|w - w'\|_{A(\xi_\ell, \widehat{\Gamma}^{L_k-1})^{-1}}^2 + 2 \max_{w,w' \in \mathcal{W}_k} \left\|\left(\Gamma^{-\top} - (\widehat{\Gamma}^\ell)^{-\top}\right)(w - w')\right\|_{\left(\sum_z \xi_\ell zz^\top\right)^{-1}}^2$$

$$+ 2 \max_{w,w' \in \mathcal{W}_k} \left\|\left(\Gamma^{-\top} - (\widehat{\Gamma}^{L_k-1})^{-\top}\right)(w - w')\right\|_{\left(\sum_z \xi_\ell zz^\top\right)^{-1}}^2. \tag{29}$$

We will upper bound the three terms in the RHS of (29) separately.

**For the first term**, by Lemma J.6,

$$\max_{w,w' \in \mathcal{W}_k} \|w - w'\|_{A(\xi_\ell, \widehat{\Gamma}^{L_k-1})^{-1}}^2 \leq 4 \max_{w,w' \in \mathcal{W}_k} \|w - w'\|_{A(\tilde{\lambda}_k, \widehat{\Gamma}^{L_k-1})^{-1}}^2, \tag{30}$$

where $\tilde{\lambda}_k$ is the optimal design for $\mathcal{W}_k$ in the first sub-phase of phase $k-1$ (based on the last doubling trick iteration), i.e.,

$$\tilde{\lambda}_k = \arg \min_{\lambda \in \Delta(\mathcal{Z})} \max_{w,w' \in \mathcal{W}_k} \|w - w'\|_{A(\lambda, \widehat{\Gamma}^{L_k-1})^{-1}}^2.$$

Then for any $\lambda$,

$$\max_{w,w' \in \mathcal{W}_k} \|w - w'\|_{A(\tilde{\lambda}_k, \widehat{\Gamma}^{L_k-1})^{-1}}^2$$

$$\leq 3 \max_{w,w' \in \mathcal{W}_k} \|w - w'\|_{A(\lambda, \Gamma)^{-1}}^2 + 2 \max_{w,w' \in \mathcal{W}_k} \left\|\left(\Gamma^{-\top} - (\widehat{\Gamma}^{L_k-1})^{-\top}\right)(w - w')\right\|_{A(\lambda, I)^{-1}}^2 \quad \text{(Lemma J.7)}$$

$$\overset{b_1}{\leq} 3 \max_{w,w' \in \mathcal{W}_k} \|w - w'\|_{A(\lambda'_k, \Gamma)^{-1}}^2 + 2 \max_{w,w' \in \mathcal{W}_k} \left\|\left(\Gamma^{-\top} - (\widehat{\Gamma}^{L_k-1})^{-\top}\right)(w - w')\right\|_{\left(\sum_z \lambda'_k zz^\top\right)^{-1}}^2$$

$$\overset{b_2}{\leq} 6 \max_{w,w' \in \mathcal{W}_k} \|w - w'\|_{A(\lambda_k^*, \Gamma)^{-1}}^2 + 2 \max_{w,w' \in \mathcal{W}_k} \left\|\left(\Gamma^{-\top} - (\widehat{\Gamma}^{L_k-1})^{-\top}\right)(w - w')\right\|_{\left(\sum_z \lambda'_k zz^\top\right)^{-1}}^2. \tag{31}$$

where for $(b_1)$, we plug in $\lambda'_k := \alpha_k^* \lambda_E + (1 - \alpha_k^*)\lambda_k^*$, with $\alpha_k^* \leq 1/2$ will be defined later and $\lambda_k^*$ is the optimal design for $\mathcal{W}_k$ given $\Gamma$, i.e.,

$$\lambda_k^* = \arg \min_{\lambda \in \Delta(\mathcal{Z})} \max_{w,w' \in \mathcal{W}_k} \|w - w'\|_{A(\lambda, \Gamma)^{-1}}^2.$$

$(b_2)$ is due to the fact that $\alpha_k^* \le 1/2$. For the second term in the RHS of (31), given the condition (23), i.e.,

$$N_{k-1,0,L_{k-1}} + N_{k-1,1,L_{k-1}} \ge \frac{4gL_\eta \widetilde{\log}_{k-1,L_{k-1}}}{\sigma_{\min}\big(A(\xi_{L_{k-1}},\Gamma)\big)}$$

by Lemma J.11 we have,

$$
\left\|\big(\Gamma^{-\top} - (\widehat{\Gamma}^{L_{k-1}})^{-\top}\big)\big(w - w'\big)\right\|^2_{(\sum_z \lambda_k' zz^\top)^{-1}}
$$

$$
\le \frac{2L_\eta \widetilde{\log}_{k-1,L_{k-1}}}{\sigma_{\min}\big(A(\lambda_k',\Gamma)\big)} \frac{1}{N_{k-1,0,L_{k-1}} + N_{k-1,1,L_{k-1}}} \|w - w'\|^2_{A(\xi_{L_{k-1}},\Gamma)^{-1}}
$$

$$
\le \frac{1}{N_{k-1,0,L_{k-1}}} \frac{2L_\eta \widetilde{\log}_{k-1,L_{k-1}}}{\sigma_{\min}\big(A(\lambda_k',\Gamma)\big)} \frac{N_{k-1,0,L_{k-1}}}{N_{k-1,0,L_{k-1}}+N_{k-1,1,L_{k-1}}} \|w - w'\|^2_{A(\xi_{L_{k-1}},\Gamma)^{-1}}
$$

$$
\le \frac{1}{N_{k-1,0,L_{k-1}}} \frac{2L_\eta \widetilde{\log}_{k-1,L_{k-1}}}{\sigma_{\min}\big(A(\alpha_k^* \lambda_E,\Gamma)\big)} \frac{N_{k-1,0,L_{k-1}}}{N_{k-1,0,L_{k-1}}+N_{k-1,1,L_{k-1}}} \|w - w'\|^2_{A(\xi_{L_{k-1}},\Gamma)^{-1}}
$$

$$
\le \frac{1}{N_{k-1,0,L_{k-1}}} \frac{2L_\eta \widetilde{\log}_{k-1,L_{k-1}}}{\sigma_{\min}\big(A(\lambda_E,\Gamma)\big)} \|w - w'\|^2_{A(\xi_{L_{k-1}},\Gamma)^{-1}} \qquad \text{(set } \alpha_k^* = \alpha_{k-1})
$$

$$
\overset{b_1}{\le} \frac{1}{4g} \|w - w'\|^2_{A(\xi_{L_{k-1}},\Gamma)^{-1}}
$$

$$
\overset{b_2}{\le} \frac{6}{4g} \|w - w'\|^2_{A(\xi_{L_{k-1}},\widehat{\Gamma}^{L_{k-1}})^{-1}}. \tag{32}
$$

where for $(b_1)$, we use the condition (22) on $N_{k-1,0,L_{k-1}}$, and $(b_2)$ is due to Lemma J.5. Plug (32) into (31), we have

$$
\max_{w,w'\in\mathcal{W}_k} \|w - w'\|^2_{A(\tilde\lambda_k,\widehat{\Gamma}^{L_{k-1}})^{-1}} \le 6 \max_{w,w'\in\mathcal{W}_k} \|w - w'\|^2_{A(\lambda_k^*,\Gamma)^{-1}} + \frac{3}{g} \max_{w,w'\in\mathcal{W}_k} \|w - w'\|^2_{A(\xi_{L_{k-1}},\widehat{\Gamma}^{L_{k-1}})^{-1}}. \tag{33}
$$

Plug (33) into (30), we have the first term in the RHS of (29) can be upper bounded by

$$
\max_{w,w'\in\mathcal{W}_k} \|w - w'\|^2_{A(\xi_\ell,\widehat{\Gamma}^{L_{k-1}})^{-1}}
$$

$$
\le 24 \max_{w,w'\in\mathcal{W}_k} \|w - w'\|^2_{A(\lambda_k^*,\Gamma)^{-1}} + \frac{12}{g} \max_{w,w'\in\mathcal{W}_k} \|w - w'\|^2_{A(\xi_{L_{k-1}},\widehat{\Gamma}^{L_{k-1}})^{-1}}
$$

$$
\le 24 \max_{w,w'\in\mathcal{W}_k} \|w - w'\|^2_{A(\lambda_k^*,\Gamma)^{-1}} + \frac{12}{g} \frac{\zeta_{k-1}^2}{\|\theta\|_2^2 L_\eta \widetilde{\log}_{k-1,L_{k-1}}} (N_{k-1,0,L_{k-1}} + N_{k-1,1,L_{k-1}}), \tag{34}
$$

where for the last inequality we use the fact that the stopping condition is satisfied at the end of $(L_{k-1})$-th doubling trick iteration for phase $k-1$ per (26).

**For the second term** in the RHS of (29), by Lemma J.10, when the condition (23) is satisfied, i.e., when

$$
N_{k,0,\ell} + N_{k,1,\ell} \ge \frac{4gL_\eta \widetilde{\log}_{k,\ell}}{\sigma_{\min}\big(A(\xi_\ell,\Gamma)\big)},
$$

we have

$$
\max_{w,w'\in\mathcal{W}_k} \left\|\big(\Gamma^{-\top} - (\widehat{\Gamma}^\ell)^{-\top}\big)\big(w - w'\big)\right\|^2_{(\sum_z \xi_\ell zz^\top)^{-1}} \le \frac{1}{g} \max_{w,w'\in\mathcal{W}_k} \|w - w'\|^2_{A(\xi_\ell,\Gamma)^{-1}}
$$

$$
\le \frac{6}{g} \max_{w,w'\in\mathcal{W}_k} \|w - w'\|^2_{A(\xi_\ell,\widehat{\Gamma}^\ell)^{-1}}, \tag{35}
$$

where the last inequality is due to Lemma J.5.

**For the third term** in the RHS of (29), by Lemma J.11, when the condition (23) is satisfied, i.e., when

$$N_{k-1,0,L_{k-1}} + N_{k-1,1,L_{k-1}} \geq \frac{4gL_\eta \widetilde{\log}_{k-1,L_{k-1}}}{\sigma_{\min}\big(A(\xi_{L_{k-1}},\Gamma)\big)}$$

we have

$$\max_{w,w'\in\mathcal{W}_k}\left\|(\Gamma^{-\top} - (\widehat{\Gamma}^{L_{k-1}})^{-\top})(w-w')\right\|^2_{(\sum_z \xi_\ell zz^\top)^{-1}}$$

$$\leq\frac{2L_\eta\widetilde{\log}_{k-1,L_{k-1}}}{\sigma_{\min}\big(A(\xi_\ell,\Gamma)\big)}\frac{1}{N_{k-1,0,L_{k-1}} + N_{k-1,1,L_{k-1}}} \max_{w,w'\in\mathcal{W}_k}\left\|w-w'\right\|^2_{A(\xi_{L_{k-1}},\Gamma)^{-1}}$$

$$\leq\frac{2L_\eta\widetilde{\log}_{k-1,L_{k-1}}}{\sigma_{\min}\big(A(\alpha_\ell\lambda_E,\Gamma)\big)}\frac{1}{N_{k-1,0,L_{k-1}} + N_{k-1,1,L_{k-1}}} \max_{w,w'\in\mathcal{W}_k}\left\|w-w'\right\|^2_{A(\xi_{L_{k-1}},\widehat{\Gamma}^{L_{k-1}})^{-1}}$$

$$\overset{b_1}{\leq}\frac{N_{k,0,\ell} + N_{k,1,\ell}}{N_{k,0,\ell}}\frac{1}{N_{k-1,0,L_{k-1}} + N_{k-1,1,L_{k-1}}}\frac{2\widetilde{\log}_{k-1,L_{k-1}}}{\sigma_{\min}\big(A(\lambda_E,\Gamma)\big)}(N_{k-1,0,L_{k-1}} + N_{k-1,1,L_{k-1}})\frac{\zeta^2_{k-1}}{\|\theta\|^2_2\widetilde{\log}_{k-1,L_{k-1}}}$$

$$\leq\frac{N_{k,0,\ell} + N_{k,1,\ell}}{N_{k,0,\ell}}\frac{2}{\sigma_{\min}\big(A(\lambda_E,\Gamma)\big)}\frac{\zeta^2_{k-1}}{\|\theta\|^2_2}$$

$$\overset{b_2}{\leq}\frac{N_{k,0,\ell} + N_{k,1,\ell}}{4gL_\eta\widetilde{\log}_{k,\ell}}\frac{\zeta^2_{k-1}}{\|\theta\|^2_2} \tag{36}$$

where $(b_1)$ is due to (26), $(b_2)$ is due to (22). Plug (34), (35) and (36) into (29), we have

$$\max_{w,w'\in\mathcal{W}_k}\left\|w-w'\right\|^2_{A(\xi_\ell,\widehat{\Gamma}^\ell)^{-1}}$$

$$\leq 72\max_{w,w'\in\mathcal{W}_k}\left\|w-w'\right\|^2_{A(\lambda^*_k,\Gamma)^{-1}} + \frac{36}{g}\frac{\zeta^2_{k-1}}{\|\theta\|^2_2L_\eta\widetilde{\log}_{k-1,L_{k-1}}}(N_{k-1,0,L_{k-1}} + N_{k-1,1,L_{k-1}})$$

$$+ \frac{12}{g}\max_{w,w'\in\mathcal{W}_k}\left\|w-w'\right\|^2_{A(\xi_\ell,\widehat{\Gamma}^\ell)^{-1}} + \frac{2}{g}\frac{N_{k,0,\ell+1} + N_{k,1,\ell+1}}{L_\eta\widetilde{\log}_{k,\ell}}\frac{\zeta^2_{k-1}}{\|\theta\|^2_2}. \tag{37}$$

With $\ell = L_k - 1$ and the fact that $g > 24$ whose exact value will be set later, we can rearrange (37) as

$$\max_{w,w'\in\mathcal{W}_k}\left\|w-w'\right\|^2_{A(\xi_{L_{k-1}},\widehat{\Gamma}^{L_k-1})^{-1}}$$

$$\leq 144\max_{w,w'\in\mathcal{W}_k}\left\|w-w'\right\|^2_{A(\lambda^*_k,\Gamma)^{-1}} + \frac{72}{g}\frac{\zeta^2_{k-1}}{\|\theta\|^2_2L_\eta\widetilde{\log}_{k-1,L_{k-1}}}(N_{k-1,0,L_{k-1}} + N_{k-1,1,L_{k-1}})$$

$$+ \frac{1}{g}\frac{N_{k,0,L_k} + N_{k,1,L_k}}{L_\eta\widetilde{\log}_{k,L_k-1}}\frac{\zeta^2_{k-1}}{\|\theta\|^2_2}. \tag{38}$$

Note that the LHS of above can be lower bounded by (28),

$$\frac{\zeta^2_k}{2\|\theta\|^2_2L_\eta\widetilde{\log}_{k,L_k-1}}(N_{k,0,L_k} + N_{k,1,L_k}) < \max_{w,w'\in\mathcal{W}_k}\left\|w-w'\right\|^2_{A(\xi_{L_{k-1}},\widehat{\Gamma}^{L_k-1})^{-1}}. \tag{39}$$

Rearrange the terms in (39) and (38) and setting $g$ to be larger enough and using the fact that $\zeta_k = \zeta_{k-1}/2$, we have

$$N_{k,0,L_k} + N_{k,1,L_k} \leq \frac{576}{\zeta^2_k}\|\theta\|^2_2L_\eta\widetilde{\log}_{k,L_k-1}\max_{w,w'\in\mathcal{W}_k}\left\|w-w'\right\|^2_{A(\lambda^*_k,\Gamma)^{-1}} + \frac{72}{g}\frac{\widetilde{\log}_{k,L_k-1}}{\widetilde{\log}_{k-1,L_{k-1}}}(N_{k-1,0,L_{k-1}} + N_{k-1,1,L_{k-1}}).$$

Note that by definition $\widetilde{\log}_{k,L_k-1} < \widetilde{\log}_{k,L_k}$, thus

$$N_{k,0,L_k} + N_{k,1,L_k} \leq \frac{576}{\zeta^2_k}\|\theta\|^2_2L_\eta\widetilde{\log}_{k,L_k}\max_{w,w'\in\mathcal{W}_k}\left\|w-w'\right\|^2_{A(\lambda^*_k,\Gamma)^{-1}} + \frac{72}{g}\frac{\widetilde{\log}_{k,L_k}}{\widetilde{\log}_{k-1,L_{k-1}}}(N_{k-1,0,L_{k-1}} + N_{k-1,1,L_{k-1}}).$$

Denote $D_k := N_{k,0,L_k} + N_{k,1,L_k}$ and divide both sides of above by $\widetilde{\log}_{k,L_k}$, we have

$$\frac{D_k}{\widetilde{\log}_{k,L_k}} \leq \frac{576}{\zeta_k^2}\|\theta\|_2^2 L_\eta \max_{w,w'\in\mathcal{W}_k}\|w-w'\|_{A(\lambda_k^*,\Gamma)^{-1}}^2 + \frac{72}{g}\frac{D_{k-1}}{\widetilde{\log}_{k-1,L_{k-1}}}.$$

In the case of $L_k = 1$, by Property 3 of Lemma J.1, we have

$$\frac{D_k}{\widetilde{\log}_{k,L_k}} = \frac{N_{k,1,1}}{8d\ln\left(1+\frac{N_{k,1,1}L_z^2}{d}\right) + 16\ln\left(\frac{2\cdot 6^d k^2}{\delta}\right)} \leq M\ln(dM).$$

Thereby we have

$$\frac{D_k}{\widetilde{\log}_{k,L_k}} \leq \max\left\{M\ln(dM), \frac{576}{\zeta_k^2}\|\theta\|_2^2 L_\eta \max_{w,w'\in\mathcal{W}_k}\|w-w'\|_{A(\lambda_k^*,\Gamma)^{-1}}^2 + \frac{72}{g}\frac{D_{k-1}}{\widetilde{\log}_{k-1,L_{k-1}}}\right\}.$$
$$(40)$$

Taking a summation over $k$ on both sides of (40), we have

$$\sum_{k=1}^{K^*}\frac{D_k}{\widetilde{\log}_{k,L_k}} = \frac{D_1}{\widetilde{\log}_{1,L_1}} + \sum_{k=2}^{K^*}\frac{D_k}{\widetilde{\log}_{k,L_k}} \tag{41}$$

$$\leq \frac{D_1}{\widetilde{\log}_{1,L_1}} + \sum_{k=2}^{K^*}\max\left\{M\ln(dM), \frac{576}{\zeta_k^2}\|\theta\|_2^2 L_\eta \max_{w,w'\in\mathcal{W}_k}\|w-w'\|_{A(\lambda_k^*,\Gamma)^{-1}}^2 + \frac{72}{g}\frac{D_{k-1}}{\widetilde{\log}_{k-1,L_{k-1}}}\right\}$$
$$(42)$$

$$\leq \frac{D_1}{\widetilde{\log}_{1,L_1}} + \sum_{k=2}^{K^*}M\ln(dM) + \sum_{k=2}^{K^*}\left(\frac{576}{\zeta_k^2}\|\theta\|_2^2 L_\eta \max_{w,w'\in\mathcal{W}_k}\|w-w'\|_{A(\lambda_k^*,\Gamma)^{-1}}^2 + \frac{72}{g}\frac{D_{k-1}}{\widetilde{\log}_{k-1,L_{k-1}}}\right)$$
$$(43)$$

$$\leq \frac{D_1}{\widetilde{\log}_{1,L_1}} + \sum_{k=2}^{K^*}M\ln(dM) + \sum_{k=2}^{K^*}\frac{576}{\zeta_k^2}\|\theta\|_2^2 L_\eta \max_{w,w'\in\mathcal{W}_k}\|w-w'\|_{A(\lambda_k^*,\Gamma)^{-1}}^2 + \sum_{k=2}^{K^*}\frac{72}{g}\frac{D_{k-1}}{\widetilde{\log}_{k-1,L_{k-1}}}.$$
$$(44)$$

Thus by setting $g = 72 \times 2$ and rearranging the terms, we have

$$\sum_{k=1}^{K^*}\frac{D_k}{\widetilde{\log}_{k,L_k}} \leq \frac{2D_1}{\widetilde{\log}_{1,L_1}} + 2\sum_{k=2}^{K^*}M\ln(dM) + 2\sum_{k=2}^{K^*}\frac{576}{\zeta_k^2}\|\theta\|_2^2 L_\eta \max_{w,w'\in\mathcal{W}_k}\|w-w'\|_{A(\lambda_k^*,\Gamma)^{-1}}^2.$$
$$(45)$$

For $D_1$, which corresponds to the first sub-phase where we use E-optimal design with doubling trick, we have the stopping condition,

$$\max_{w,w'\in\mathcal{W}_1}\|w-w'\|_{\bar{A}(Z_{1,0,L_1},\widehat{\Gamma}^{L_1})^{-1}}^2\|\theta\|_2^2 L_\eta\widetilde{\log}_{1,L_1} \leq \zeta_1^2.$$

This implies that when the stopping condition is met

$$N_{1,0,L_1} \geq \frac{\|\theta\|_2^2 L_\eta\widetilde{\log}_{1,L_1}}{\zeta_1^2}\max_{w,w'\in\mathcal{W}_1}\|w-w'\|_{A(\xi_{L_1},\widehat{\Gamma}^{L_1})^{-1}}^2. \tag{46}$$

and

$$N_{1,0,L_1-1} < \frac{2\|\theta\|_2^2 L_\eta\widetilde{\log}_{1,L_1-1}}{\zeta_1^2}\max_{w,w'\in\mathcal{W}_1}\|w-w'\|_{A(\xi_{L_1-1},\widehat{\Gamma}^{L_1-1})^{-1}}^2. \tag{47}$$

Since we use E-optimal design for the first phase, the factor of $\left\|w - w'\right\|^2_{A(\xi_{L_1-1}, \widehat{\Gamma}^{L_1-1})^{-1}}$ can be upper bounded by

$$\left\|w - w'\right\|^2_{A(\xi_{L_1-1}, \widehat{\Gamma}^{L_1-1})^{-1}}$$

$$\leq 3\left\|w - w'\right\|^2_{A(\xi_{L_1-1}, \Gamma)^{-1}} + 2\left\|(\Gamma^{-\top} - (\widehat{\Gamma}^{L_1-1})^{-\top})(w - w')\right\|^2_{(\sum_z \xi_{L_1-1} zz^\top)^{-1}} \quad \text{(Lemma J.7)}$$

$$\leq 3\left\|w - w'\right\|^2_{A(\lambda_E, \Gamma)^{-1}} + 2\left\|w - w'\right\|^2_{A(\lambda_E, \Gamma)^{-1}} \quad \text{(Lemma J.10)}$$

$$\leq 6\left\|w - w'\right\|^2_{A(\lambda_E, \Gamma)^{-1}}. \quad (48)$$

Plug (48) into (47), and use the fact that $2N_{1,0,L_1-1} = N_{1,0,L_1}, \zeta_1 = 1$, we have

$$N_{1,0,L_1} \leq 24\|\theta\|_2^2 L_\eta \widetilde{\log}_{1,L_1-1} \max_{w,w'\in\mathcal{W}_1} \left\|w - w'\right\|^2_{A(\lambda_E,\Gamma)^{-1}} \leq 24\|\theta\|_2^2 L_\eta \widetilde{\log}_{1,L_1} \max_{w,w'\in\mathcal{W}_1} \left\|w - w'\right\|^2_{A(\lambda_E,\Gamma)^{-1}}. \quad (49)$$

Thus we have,

$$\frac{D_1}{\widetilde{\log}_{1,L_1}} \leq \frac{2N_{1,0,L_1}}{\widetilde{\log}_{1,L_1}} \leq 48\|\theta\|_2^2 L_\eta \max_{w,w'\in\mathcal{W}_1} \left\|w - w'\right\|^2_{A(\lambda_E,\Gamma)^{-1}} =: \rho_1.$$

By the same calculation as (17) and (19), we have

$$\|\theta\|_2^2 L_\eta \sum_{k=2}^{K^*} \frac{1}{\zeta_k^2} \max_{w,w'\in\mathcal{W}_k} \left\|w - w'\right\|^2_{A(\lambda_k^*,\Gamma)^{-1}} \leq c\|\theta\|_2^2 L_\eta K^* \rho^* =: \rho_2,$$

where $c$ is an absolute constant. Next we lower bound the left hand side of (45). To do this, we first upper bound $\widetilde{\log}_{k,L_k}$ as

$$\widetilde{\log}_{k,L_k} = 8d\ln\left(1 + \frac{D_k L_z^2}{d}\right) + 16\ln\left(\frac{2\cdot 6^d k^2 L_k^2}{\delta}\right)$$

$$\leq 8d\ln\left(1 + \frac{D_k L_z^2}{d}\right) + 32\ln(L_k) + 16\ln\left(\frac{2\cdot 6^d k^2}{\delta}\right)$$

$$\leq 8d\ln\left(1 + \frac{D_k L_z^2}{d}\right) + 32\ln\left(\log_2\left(\frac{D_k}{d}\right)\right) + 16\ln\left(\frac{2\cdot 6^d k^2}{\delta}\right)$$

$$\leq 32d\ln\left(1 + \frac{D_k L_z^2}{d}\right) + 16\ln\left(\frac{2\cdot 6^d k^2}{\delta}\right), \quad (50)$$

where the inequality above uses the fact that $L_k$ is the index of the last doubling trick iteration for phase $k$, by the design of the doubling trick, we have $L_k \leq \log_2\left(\frac{D_k}{d}\right)$.

$$\sum_{k=1}^{K^*} \frac{D_k}{\widetilde{\log}_{k,L_k}} = \sum_{k=1}^{K^*} \frac{D_k}{8d\ln\left(1 + \frac{D_k L_z^2}{d}\right) + 16\ln\left(\frac{2\cdot 6^d k^2 L_k^2}{\delta}\right)}$$

$$\geq \sum_{k=1}^{K^*} \frac{D_k}{32d\ln\left(1 + \frac{D_k L_z^2}{d}\right) + 16\ln\left(\frac{2\cdot 6^d K^{*2}}{\delta}\right)} \quad \text{(due to (50))}$$

$$\geq \sum_{k=1}^{K^*} \frac{D_k}{32d\ln\left(1 + \frac{\sum_{k=1}^{K^*} D_k L_z^2}{d}\right) + 16\ln\left(\frac{2\cdot 6^d K^{*2}}{\delta}\right)}$$

$$= \frac{1}{32d\ln\left(1 + \frac{\sum_{k=1}^{K^*} D_k L_z^2}{d}\right) + 16\ln\left(\frac{2\cdot 6^d K^{*2}}{\delta}\right)} \sum_{k=1}^{K^*} D_k.$$

Denote $\rho_3 := K^* M \ln(dM)$, looking back at (44), we have

$$\sum_{k=1}^{K^*} D_k \leq \left( 32d \ln\left( 1 + \frac{\sum_{k=1}^{K^*} D_k L_z^2}{d} \right) + 16 \ln\left( \frac{2 \cdot 6^d K^{*2}}{\delta} \right) \right) (\rho_1 + \rho_2 + \rho_3).$$

By Lemma K.5, we have

$$\sum_{k=1}^{K^*} D_k \leq 32(\rho_1 + \rho_2 + \rho_3) \ln\left( \frac{2 \cdot 6^d K^{*2}}{\delta} \right) + 64d(\rho_1 + \rho_2 + \rho_3) \ln\left( 3 + (\rho_1 + \rho_2 + \rho_3) \frac{2L_z^2}{d} \right).$$
$$(51)$$

### Sample complexity for second sub-phase

The design for the second sub-phase of phase $k$ is based on $\widehat{\Gamma}^{L_k}$, the estimate of $\Gamma$ at the end of the $L_k$-th doubling trick iteration in the first sub-phase of phase $k$,

$$\hat{\lambda}_k = \arg \min_{\lambda \in \Delta(\mathcal{Z})} \max_{w,w' \in \mathcal{W}_k} \left\| w - w' \right\|_{A(\lambda, \widehat{\Gamma}^{L_k})^{-1}}^2.$$

Then for any $\lambda$, by Lemma J.7 we have

$$\max_{w,w' \in \mathcal{W}_k} \left\| w - w' \right\|_{A(\hat{\lambda}_k, \widehat{\Gamma}^{L_k})^{-1}}^2$$

$$\leq 3 \max_{w,w' \in \mathcal{W}_k} \left\| w - w' \right\|_{A(\lambda, \Gamma)^{-1}}^2 + 2 \max_{w,w' \in \mathcal{W}_k} \left\| (\Gamma^{-\top} - (\widehat{\Gamma}^{L_k})^{-\top})(w - w') \right\|_{A(\lambda, I)^{-1}}^2$$

We plug in $\lambda_k'' := \alpha_k^* \lambda_E + (1 - \alpha_k^*)\lambda_k^*$, with $\alpha_k^* \leq 1/2$ will be defined later and $\lambda_k^*$ is the optimal design for $\mathcal{W}_k$,

$$\max_{w,w' \in \mathcal{W}_k} \left\| w - w' \right\|_{A(\hat{\lambda}_k, \widehat{\Gamma}^{L_k})^{-1}}^2$$

$$\leq 3 \max_{w,w' \in \mathcal{W}_k} \left\| w - w' \right\|_{A(\lambda_k'', \Gamma)^{-1}}^2 + 2 \max_{w,w' \in \mathcal{W}_k} \left\| (\Gamma^{-\top} - (\widehat{\Gamma}^{L_k})^{-\top})(w - w') \right\|_{(\sum_z \lambda_k'' zz^\top)^{-1}}^2$$

$$\leq 6 \max_{w,w' \in \mathcal{W}_k} \left\| w - w' \right\|_{A(\lambda_k^*, \Gamma)^{-1}}^2 + 2 \max_{w,w' \in \mathcal{W}_k} \left\| (\Gamma^{-\top} - (\widehat{\Gamma}^{L_k})^{-\top})(w - w') \right\|_{(\sum_z \lambda_k'' zz^\top)^{-1}}^2, \quad (52)$$

where the last inequality is due to the fact that $\alpha_k^* \leq 1/2$. For the second term in the RHS of above, by Lemma J.11 we have,

$$\left\| (\Gamma^{-\top} - (\widehat{\Gamma}^{L_k})^{-\top})(w - w') \right\|_{(\sum_z \lambda_k'' zz^\top)^{-1}}^2$$

$$\leq \frac{4L_\eta \widetilde{\log}_{k,L_k}}{\sigma_{\min}(A(\lambda_k'', \Gamma))} \left\| w - w' \right\|_{\bar{A}(Z_{k,0} \cup Z^{L_k}, \Gamma)^{-1}}^2$$

$$\leq \frac{1}{N_{k,0,L_k}} \frac{4L_\eta \widetilde{\log}_k}{\sigma_{\min}(A(\lambda_k'', \Gamma))} \frac{N_{k,0,L_k}}{N_{k,0,L_k} + N_{k,1L_k}} \left\| w - w' \right\|_{A(\xi_{L_k}, \Gamma)^{-1}}^2$$

$$\leq \frac{1}{N_{k,0,L_k}} \frac{4L_\eta \widetilde{\log}_{k,L_k}}{\sigma_{\min}(A(\alpha_k^* \lambda_E, \Gamma))} \frac{N_{k,0,L_k}}{N_{k,0,L_k} + N_{k,1,L_k}} \left\| w - w' \right\|_{A(\xi_{L_k}, \Gamma)^{-1}}^2$$

$$\leq \frac{1}{N_{k,0,L_k}} \frac{4L_\eta \widetilde{\log}_{k,L_k}}{\sigma_{\min}(A(\lambda_E, \Gamma))} \left\| w - w' \right\|_{A(\xi_{L_k}, \Gamma)^{-1}}^2 \qquad \text{(set } \alpha_k^* = \alpha_k)$$

$$\overset{b_1}{\leq} \frac{1}{g} \left\| w - w' \right\|_{A(\xi_{L_k}, \Gamma)^{-1}}^2$$

$$\overset{b_2}{\leq} \frac{1}{g} \left\| w - w' \right\|_{A(\xi_{L_k}, \widehat{\Gamma}^{L_k})^{-1}}^2, \qquad (53)$$

where for $(b_1)$, we have $N_{k,0,L_k} \geq \frac{4gL_\eta \widetilde{\log}_{k,L_k}}{\sigma_{\min}(A(\lambda_E,\Gamma))}$, and $(b_2)$ is due to Lemma J.5. Plug (53) into (52), we have

$$
\begin{aligned}
\max_{w,w' \in \mathcal{W}_k} \|w - w'\|^2_{A(\hat\lambda_k, \widehat{\Gamma}^{L_k})^{-1}} &\leq 6 \max_{w,w' \in \mathcal{W}_k} \|w - w'\|^2_{A(\lambda_k^*,\Gamma)^{-1}} + \frac{2}{g} \max_{w,w' \in \mathcal{W}_k} \|w - w'\|^2_{A(\xi_{L_k}, \widehat{\Gamma}^{L_k})^{-1}} \\
&\leq 6 \max_{w,w' \in \mathcal{W}_k} \|w - w'\|^2_{A(\lambda_k^*,\Gamma)^{-1}} + \frac{2}{g} \frac{\zeta_k^2}{\|\theta\|_2^2 L_\eta \widetilde{\log}_{k,L_k}} (N_{k,0,L_k} + N_{k,1,L_k}),
\end{aligned}
\tag{54}
$$

where the last inequality is due to (26). According to the algorithm design, the number of samples in the second sub-phase of phase $k$ is defined as

$$
N_{k,2} = \left\lceil 2(1+\omega)\zeta_k^{-2}\rho(\mathcal{W}_k)L_\nu \log\left(\frac{4k^2|\mathcal{W}|}{\delta}\right) \right\rceil \vee r(\omega),
$$

with $\rho(\mathcal{W}_k) = \min_{\lambda \in \Delta(\mathcal{Z})} \max_{w,w' \in \mathcal{W}_k} \|w - w'\|^2_{A(\lambda,\widehat{\Gamma}^{L_k})^{-1}}$. Then we have, by setting $g \geq 4$,

$$
N_{k,2} \lesssim\gtrsim (1+\omega)\zeta_k^{-2}L_\nu \max_{w,w' \in \mathcal{W}_k} \|w - w'\|^2_{A(\lambda_k^*,\Gamma)^{-1}} \log\left(\frac{4k^2|\mathcal{W}|}{\delta}\right) + (1+\omega)(N_{k,0,L_k} + N_{k,1,L_k}),
$$

where '$\lesssim\gtrsim$' hides logarithmic factors of $|\mathcal{W}|$ for the second term and constants for simplicity. Plug (54) into the above inequality. Also note that by Lemma 2.1, we can always set $L_\nu = 2(\|\theta\|_2^2 L_\eta + 1)$. Thus,

$$
\begin{aligned}
\sum_{k=1}^{K^*} N_{k,2} &\lesssim\gtrsim (1+\omega) \sum_{k=1}^{K^*} \zeta_k^{-2}L_\nu \max_{w,w' \in \mathcal{W}_k} \|w - w'\|^2_{A(\lambda_k^*,\Gamma)^{-1}} \log\left(\frac{4k^2|\mathcal{W}|}{\delta}\right) + (1+\omega) \sum_{k=1}^{K^*}(N_{k,0,L_k} + N_{k,1,L_k}) \\
&\lesssim\gtrsim (1+\omega)K^*L_\nu\rho^* \log\left(\frac{4K^{*2}|\mathcal{W}|}{\delta}\right) + (1+\omega) \sum_{k=1}^{K^*}(N_{k,0,L_k} + N_{k,1,L_k}).
\end{aligned}
$$

This essentially means that the sample complexity for the second sub-phase $\sum N_{k,2}$ can be upper bounded by summation of the sample complexity we pay for Algorithm 1 and the sample complexity of the first sub-phase $\sum_{k=1}^{K^*}(N_{k,0,L_k} + N_{k,1,L_k})$. Combine (51) and (55), we conclude the result.

$\square$

**Lemma J.3.** *Denote*

$$
\widetilde{\log}(N_{k,0}, \delta_{k,0}) = 8d\ln\left(1 + \frac{N_{k,0}L_z^2}{d}\right) + 16\ln\left(\frac{2 \cdot 6^d}{\delta}\right).
$$

*A sufficient condition for*

$$
N_{k,0,\ell} \geq \frac{g\sigma_\eta^2 \widetilde{\log}(N_{k,0,\ell} + N_{k,1,\ell}, \delta_{k,\ell})}{\sigma_{\min}(A(\lambda_E,\Gamma))} \vee r(\omega).
\tag{55}
$$

*to hold is*

$$
N_{k,0,\ell} \geq \frac{16gd\sigma_\eta^2}{\sigma_{\min}(A(\lambda_E,\Gamma))} \ln\left(\frac{8g}{\sigma_{\min}(A(\lambda_E,\Gamma))}\left(d + N_{k,1,\ell}L_z^2 + L_z^2\right)\right) + \frac{32g\sigma_\eta^2}{\sigma_{\min}(A(\lambda_E,\Gamma))} \ln\left(\frac{2 \cdot 6^d}{\delta}\right).
$$

*Proof.* Given

$$
\widetilde{\log}(N_{k,0,\ell} + N_{k,1,\ell}, \delta_{k,\ell}) = 8d\ln\left(1 + \frac{(N_{k,0,\ell} + N_{k,1,\ell})L_z^2}{d}\right) + 16\ln\left(\frac{2 \cdot 6^d}{\delta}\right).
$$

By Lemma J.4, for the formula

$$
X \geq A\ln(D + BX) + C
$$

we have

- $A = \frac{8gd\sigma_\eta^2}{\sigma_{\min}\left(A(\lambda_E,\Gamma)\right)}$,

- $B = \frac{L_z^2}{d}$,

- $C = \frac{16g\sigma_\eta^2}{\sigma_{\min}\left(A(\lambda_E,\Gamma)\right)} \ln\left(\frac{2\cdot 6^d}{\delta}\right)$,

- $D = 1 + \frac{N_{k,1,\ell}L_z^2}{d}$.

Thus a sufficient condition for the inequality to hold is

$$
\begin{aligned}
N_{k,0,\ell} \geq & \frac{16gd\sigma_\eta^2}{\sigma_{\min}\left(A(\lambda_E,\Gamma)\right)} \ln\left( \frac{8gd\sigma_\eta^2}{\sigma_{\min}\left(A(\lambda_E,\Gamma)\right)}\left( 1 + \frac{N_{k,1,\ell}L_z^2}{d} + \frac{L_z^2}{d} \right) \right) + \frac{32g\sigma_\eta^2}{\sigma_{\min}\left(A(\lambda_E,\Gamma)\right)} \ln\left( \frac{2\cdot 6^d}{\delta} \right) \\
= & \frac{16gd\sigma_\eta^2}{\sigma_{\min}\left(A(\lambda_E,\Gamma)\right)} \ln\left( \frac{8g\sigma_\eta^2}{\sigma_{\min}\left(A(\lambda_E,\Gamma)\right)}\left( d + N_{k,1,\ell}L_z^2 + L_z^2 \right) \right) + \frac{32g\sigma_\eta^2}{\sigma_{\min}\left(A(\lambda_E,\Gamma)\right)} \ln\left( \frac{2\cdot 6^d}{\delta} \right).
\end{aligned}
$$

$\square$

**Lemma J.4.** *Let $X \geq 1, A, B \geq 0$, then a sufficient condition for $X \geq A\ln\left(D + BX\right) + C$ is*
$$
X \geq 2A\ln(AD + AB) + 2C.
$$

*Proof.* The proof is motivated by Gales et al. [16]. Let $f \in (0,1)$, then

$$
\begin{aligned}
& X \geq A\ln\left(D + BX\right) + C \\
\Leftrightarrow & X \geq A\ln\left(\frac{fX}{A}\right) + A\ln\left(\frac{A}{f}\left(\frac{D}{X} + B\right)\right) + C \\
\Leftarrow & X \geq A\left(\frac{fX}{A} - 1\right) + A\ln\left(\frac{A}{f}\left(\frac{D}{X} + B\right)\right) + C \qquad \text{(since } \ln(x) \leq x - 1) \\
\Leftarrow & X(1 - f) \geq A\ln\left(\frac{A}{2f}\left(\frac{D}{X} + B\right)\right) + C \\
\Leftrightarrow & X \geq \frac{1}{1-f}A\ln\left(\frac{A}{2f}\left(\frac{D}{X} + B\right)\right) + \frac{c}{1-f}.
\end{aligned}
$$

Set $f = 1/2$ and by the fact $X \geq 1$, we have
$$
X \geq 2A\ln(AD + AB) + 2C.
$$

$\square$

**Lemma J.5.** *Suppose that we have a data set $\{Z_T, X_T\}$. Denote the empirical distribution of $Z_T$ as $\xi$. The number of samples satisfies*

$$
T \geq \frac{8\sigma_\eta^2}{\sigma_{\min}\left(A(\xi,\Gamma)\right)} \overline{\log}(Z_T, \delta) \vee r(\omega).
$$

*$\widehat{\Gamma}$ is the OLS estimate of $\Gamma$ based on $\{Z_T, X_T\}$. Then*

$$
\|w\|_{A(\xi,\Gamma)^{-1}}^2 \leq 6\|w\|_{A(\xi,\widehat{\Gamma})^{-1}}^2.
$$

*Proof.* By Lemma J.7, we have

$$
\begin{aligned}
\|w\|_{A(\xi,\Gamma)^{-1}}^2 \leq & 3\|w\|_{A(\xi,\widehat{\Gamma})^{-1}}^2 + 2\left\|(\Gamma^{-\top} - \widehat{\Gamma}^{-\top})w\right\|_{A(\xi,I)^{-1}}^2 \\
\leq & 3\|w\|_{A(\xi,\widehat{\Gamma})^{-1}}^2 + \frac{1}{2}\|w\|_{A(\xi,\Gamma)^{-1}}^2,
\end{aligned}
$$

where the last inequality is due to Lemma J.10 with $g = 2$. Rearranging the terms, we have

$$\|w\|^2_{A(\xi,\Gamma)^{-1}} \leq 6\|w\|^2_{A(\xi,\widehat{\Gamma})^{-1}}.$$

$\square$

**Lemma J.6.** *Suppose that we use ROUND to sample $N_0$ arms according to $\lambda_E$ denoted as $Z_0$, and $N_1$ arms according to $\lambda_1$ denoted as $Z_1$, with $N_1 \geq N_0$. Denote the empirical distribution of all the collected samples as $\xi$, then*

$$\|w\|^2_{A(\xi,\Gamma)} \leq 4\|w\|^2_{A(\lambda_1,\Gamma)^{-1}}.$$

*Proof.* Denote the empirical distribution of $Z_0$ as $\xi_0$, and the empirical distribution of $Z_1$ as $\xi_1$, then we have

$$
\begin{aligned}
\|w\|^2_{\bar{A}(Z_0 \cup Z_1, \Gamma)^{-1}} &= w^\top \left( \sum_{z \in Z_0 \cup Z_1} \Gamma^\top z z^\top \Gamma \right)^{-1} w \\
&= w^\top \left( \sum_{z \in Z_0} \Gamma^\top z z^\top \Gamma + \sum_{z \in Z_1} \Gamma^\top z z^\top \Gamma \right)^{-1} w \\
&= w^\top \left( N_0 \sum_{z \in \mathcal{Z}} \xi_{z,0} \Gamma^\top z z^\top \Gamma + N_1 \sum_{z \in \mathcal{Z}} \xi_{z,1} \Gamma^\top z z^\top \Gamma \right)^{-1} w \\
&= \frac{1}{N_0 + N_1} w^\top \left( \frac{N_0}{N_0 + N_1} \sum_{z \in \mathcal{Z}} \xi_{z,0} \Gamma^\top z z^\top \Gamma + \frac{N_1}{N_0 + N_1} \sum_{z \in \mathcal{Z}} \xi_{z,1} \Gamma^\top z z^\top \Gamma \right)^{-1} w \\
&\leq \frac{1}{N_0 + N_1} w^\top \left( \frac{N_1}{N_0 + N_1} \sum_{z \in \mathcal{Z}} \xi_{z,1} \Gamma^\top z z^\top \Gamma \right)^{-1} w \\
&\leq \frac{2}{N_0 + N_1} w^\top \left( \sum_{z \in \mathcal{Z}} \xi_{z,1} \Gamma^\top z z^\top \Gamma \right)^{-1} w \qquad (N_1 \geq N_0) \\
&\leq \frac{4}{N_0 + N_1} w^\top \left( \sum_{z \in \mathcal{Z}} \lambda_1 \Gamma^\top z z^\top \Gamma \right)^{-1} w \\
&= \frac{4}{N_0 + N_1} \|w\|^2_{A(\lambda_1,\Gamma)^{-1}}.
\end{aligned}
$$

The result follows by noting that

$$\|w\|^2_{\bar{A}(Z_0 \cup Z_1, \Gamma)^{-1}} = \frac{1}{N_0 + N_1} \|w\|^2_{A(\xi,\Gamma)}.$$

$\square$

**Lemma J.7.** *Suppose that we have an estimate $\widehat{\Gamma}$ that is invertible, for any $x \in \mathbb{R}^d$ and covariance matrix $V$, we have*

$$\|x\|^2_{(\Gamma^\top V \Gamma)^{-1}} \leq 3\|x\|^2_{(\widehat{\Gamma}^\top V \widehat{\Gamma})^{-1}} + 2\left\|(\Gamma^{-\top} - \widehat{\Gamma}^{-\top})x\right\|^2_{V^{-1}}. \tag{56}$$

*Proof.* Suppose we have an estimate $\widehat{\Gamma}$. Then,

$$\|x\|^2_{(\Gamma^\top V \Gamma)^{-1}} = \|x\|^2_{(\widehat{\Gamma}^\top V \widehat{\Gamma})^{-1}} + \|x\|^2_{(\Gamma^\top V \Gamma)^{-1}} - \|x\|^2_{(\widehat{\Gamma}^\top V \widehat{\Gamma})^{-1}}.$$

Note that

$$(\Gamma^\top V \Gamma)^{-1} - (\widehat{\Gamma}^\top V \widehat{\Gamma})^{-1} = \Gamma^{-1} V^{-1} \Gamma^{-\top} - \widehat{\Gamma}^{-1} V^{-1} \widehat{\Gamma}^{-\top}$$
$$= \Gamma^{-1} V^{-1} \Gamma^{-\top} - \widehat{\Gamma}^{-1} V^{-1} \widehat{\Gamma}^{-\top} + \Gamma^{-1} V^{-1} \widehat{\Gamma}^{-\top} - \Gamma^{-1} V^{-1} \widehat{\Gamma}^{-\top}$$
$$= \Gamma^{-1} V^{-1} (\Gamma^{-\top} - \widehat{\Gamma}^{-\top}) + (\Gamma^{-1} - \widehat{\Gamma}^{-1}) V^{-1} \widehat{\Gamma}^{-\top}.$$

Thus,

$$\|x\|^2_{(\Gamma^\top V \Gamma)^{-1}} = \|x\|^2_{(\widehat{\Gamma}^\top V \widehat{\Gamma})^{-1}} + x^\top \Gamma^{-1} V^{-1} (\Gamma^{-\top} - \widehat{\Gamma}^{-\top}) x + x^\top (\Gamma^{-1} - \widehat{\Gamma}^{-1}) V^{-1} \widehat{\Gamma}^{-\top} x$$
$$\leq \|x\|^2_{(\widehat{\Gamma}^\top V \widehat{\Gamma})^{-1}} + \|x\|_{(\Gamma^\top V \Gamma)^{-1}} \left\|(\Gamma^{-\top} - \widehat{\Gamma}^{-\top}) x\right\|_{V^{-1}} + \left\|(\Gamma^{-\top} - \widehat{\Gamma}^{-\top}) x\right\|_{V^{-1}} \|x\|_{(\widehat{\Gamma}^\top V \widehat{\Gamma})^{-1}}$$
$$\leq \|x\|^2_{(\widehat{\Gamma}^\top V \widehat{\Gamma})^{-1}} + \frac{1}{2} \|x\|^2_{(\Gamma^\top V \Gamma)^{-1}} + \frac{1}{2} \left\|(\Gamma^{-\top} - \widehat{\Gamma}^{-\top}) x\right\|^2_{V^{-1}} + \frac{1}{2} \left\|(\Gamma^{-\top} - \widehat{\Gamma}^{-\top}) x\right\|^2_{V^{-1}} + \frac{1}{2} \|x\|^2_{(\widehat{\Gamma}^\top V \widehat{\Gamma})^{-1}}.$$
(AM-GM)

This implies that

$$\|x\|^2_{(\Gamma^\top V \Gamma)^{-1}} \leq 3 \|x\|^2_{(\widehat{\Gamma}^\top V \widehat{\Gamma})^{-1}} + 2 \left\|(\Gamma^{-\top} - \widehat{\Gamma}^{-\top}) x\right\|^2_{V^{-1}}.$$

$\square$

**Lemma J.8.** *Suppose $\lambda_f = \arg\min f(\lambda)$ and $\lambda_g = \arg\min g(\lambda)$ and $f(\lambda) \leq g(\lambda) + h(\lambda)$, then*
$$f(\lambda_f) \leq g(\lambda_g) + h(\lambda_g).$$

*Proof.*
$$f(\lambda_f) \leq \min_\lambda \big( g(\lambda) + h(\lambda) \big) \leq g(\lambda_g) + h(\lambda_g).$$

$\square$

**Lemma J.9.** *Define $\lambda_z^*$ and $\tilde{\lambda}_z$*
$$\lambda_z^* := \arg\min_{\lambda \in \Delta(\mathcal{Z})} \max_{w,w' \in \mathcal{W}} \|w - w'\|^2_{A(\lambda_z, \Gamma)^{-1}},$$

*and*

$$\tilde{\lambda}_z := \arg\min_{\lambda \in \Delta(\mathcal{Z})} \max_{w,w' \in \mathcal{W}} \|w - w'\|^2_{A(\lambda_z, \widehat{\Gamma})^{-1}}$$

*as the optimal design regarding $\Gamma$ and that regarding its estimate $\widehat{\Gamma}$ respectively. Then, we have*

$$\max_{w,w' \in \mathcal{W}} \|w - w'\|^2_{A(\tilde{\lambda}_z, \widehat{\Gamma})^{-1}} \leq \max_{w,w' \in \mathcal{W}} 3 \|w - w'\|^2_{A(\lambda_z^*, \Gamma)^{-1}} + \max_{w,w' \in \mathcal{W}} 2 \left\|(\Gamma^{-\top} - \widehat{\Gamma}^{-\top})(w - w')\right\|^2_{\left(\sum_z \lambda_z^* z z^\top\right)^{-1}}.$$

*Proof.* By Lemma J.7, for any $w, w' \in \mathcal{W}$ and $\lambda_z \in \Delta_\mathcal{Z}$,

$$\|w - w'\|^2_{A(\lambda_z, \widehat{\Gamma})^{-1}} \leq 3 \|w - w'\|^2_{A(\lambda_z, \Gamma)^{-1}} + 2 \left\|(\Gamma^{-\top} - \widehat{\Gamma}^{-\top})(w - w')\right\|^2_{\left(\sum_z \lambda_z z z^\top\right)^{-1}}.$$

Thus

$$\max_{w,w' \in \mathcal{W}} \|w - w'\|^2_{A(\lambda_z, \widehat{\Gamma})^{-1}} \leq \max_{w,w' \in \mathcal{W}} 3 \|w - w'\|^2_{A(\lambda_z, \Gamma)^{-1}} + \max_{w,w' \in \mathcal{W}} 2 \left\|(\Gamma^{-\top} - \widehat{\Gamma}_k^{-\top})(w - w')\right\|^2_{\left(\sum_z \lambda_z z z^\top\right)^{-1}}.$$

By Lemma J.8,

$$\max_{w,w' \in \mathcal{W}} \|w - w'\|^2_{A(\tilde{\lambda}_z, \widehat{\Gamma})^{-1}} \leq \max_{w,w' \in \mathcal{W}} 3 \|w - w'\|^2_{A(\lambda_z^*, \Gamma)^{-1}} + \max_{w,w' \in \mathcal{W}} 2 \left\|(\Gamma^{-\top} - \widehat{\Gamma}^{-\top})(w - w')\right\|^2_{\left(\sum_z \lambda_z^* z z^\top\right)^{-1}}.$$

$\square$

**Lemma J.10.** *Suppose that we have $\widehat{\Gamma}$ that is an OLS estimate from an offline dataset $\{Z_{T_1}, X_{T_1}\}$ collected non-adaptively through a fixed design $\lambda$ and the efficient rounding procedure ROUND. Let $V = Z_{T_1}^\top Z_{T_1}$. Then, for any $x \in \mathbb{R}^d$ and $g \geq 1$, we have, with probability $1 - \delta$,*

$$\left\| (\Gamma^{-\top} - \widehat{\Gamma}^{-\top})x \right\|_{V^{-1}}^2 \leq \frac{1}{g} \|x\|_{\bar{A}(Z_{T_1}, \Gamma)^{-1}}^2,$$

*when*

$$T_1 \geq \frac{4g\sigma_\eta^2}{\sigma_{\min}\left(A(\lambda, \Gamma)\right)} \overline{\log}(Z_{T_1}, \delta) \vee 2p, \tag{57}$$

*where $p$ is the cardinality of support of $\lambda$.*

*Proof.* We first show that

$$\left\| (\Gamma^{-\top} - \widehat{\Gamma}^{-\top})x \right\|_{V^{-1}}^2$$

$$= \left\| \Gamma^{-\top} S^\top Z_{T_1} V^{-1} (\Gamma + V^{-1} Z_{T_1}^\top S)^{-\top} x \right\|_{V^{-1}}^2 \qquad \text{(Lemma J.12)}$$

$$\overset{a_1}{=} \left\| V^{-\frac{1}{2}} \Gamma^{-\top} S^\top Z_{T_1} V^{-1} (\Gamma + V^{-1} Z_{T_1}^\top S)^{-\top} x \right\|_2^2 \qquad (\|Ax\|_V^2 = \left\| V^{\frac{1}{2}} Ax \right\|_2^2)$$

$$\leq \left\| V^{-\frac{1}{2}} \Gamma^{-\top} S^\top Z_{T_1} V^{-\frac{1}{2}} \right\|_{\text{op}}^2 \left\| V^{-\frac{1}{2}} (\Gamma + V^{-1} Z_{T_1}^\top S)^{-\top} x \right\|_2^2 \qquad (\|Ax\| \leq \|A\|_{\text{op}} \|x\|)$$

$$\leq \left\| V^{-\frac{1}{2}} \Gamma^{-\top} \right\|_{\text{op}}^2 \left\| S^\top Z_{T_1} V^{-\frac{1}{2}} \right\|_{\text{op}}^2 \left\| V^{-\frac{1}{2}} (\Gamma + V^{-1} Z_{T_1}^\top S)^{-\top} x \right\|_2^2 \qquad (\|AB\|_{\text{op}} \leq \|A\|_{\text{op}} \|B\|_{\text{op}})$$

$$= \left\| \Gamma^{-1} V^{-1} \Gamma^{-\top} \right\|_{\text{op}} \left\| V^{-1/2} Z_{T_1}^\top S \right\|_{\text{op}}^2 \left\| V^{-1/2} \left( \Gamma + V^{-1} Z_{T_1}^\top S \right)^{-\top} x \right\|_2^2 \qquad (\|A^\top A\|_{\text{op}} = \|A\|_{\text{op}}^2)$$

$$= \left\| \Gamma^{-1} V^{-1} \Gamma^{-\top} \right\|_{\text{op}} \left\| V^{-1/2} Z_{T_1}^\top S \right\|_{\text{op}}^2 \left\| V^{-1/2} \Gamma^{-\top} x - V^{-1/2} \Gamma^{-\top} \left( V^{-1} Z_{T_1}^\top S \right)^\top \left( \Gamma + V^{-1} Z_{T_1}^\top S \right)^{-\top} x \right\|_2^2$$

$$\qquad \text{(Lemma J.13: } (A + B)^{-1} = A^{-1} - (A + B)^{-1} B A^{-1})$$

$$\overset{a_2}{\leq} \left\| \Gamma^{-1} V^{-1} \Gamma^{-\top} \right\|_{\text{op}} \left\| V^{-1/2} Z_{T_1}^\top S \right\|_{\text{op}}^2 \left( \left\| V^{-1/2} \Gamma^{-\top} x \right\|_2^2 + \left\| V^{-1/2} \Gamma^{-\top} \left( V^{-1} Z_{T_1}^\top S \right)^\top \left( \Gamma + V^{-1} Z_{T_1}^\top S \right)^{-\top} x \right\|_2^2 \right).$$

We can upper bound the the term $\left\| V^{-1/2} \Gamma^{-\top} \left( V^{-1} Z_{T_1}^\top S \right)^\top \left( \Gamma + V^{-1} Z_{T_1}^\top S \right)^{-\top} x \right\|_2^2 =: \mathfrak{V}$ by noticing that it appears in both of $a_1, a_2$ above. Thus we have the inequality

$$\mathfrak{V} \leq \left\| \Gamma^{-1} V^{-1} \Gamma^{-\top} \right\|_{\text{op}} \left\| V^{-1/2} Z_{T_1}^\top S \right\|_{\text{op}}^2 \left( \left\| V^{-1/2} \Gamma^{-\top} x \right\|_2^2 + \mathfrak{V} \right).$$

By rearranging the terms, we have

$$\mathfrak{V} \leq \frac{1}{1 - \left\| \Gamma^{-1} V^{-1} \Gamma^{-\top} \right\|_{\text{op}} \left\| V^{-1/2} Z_{T_1}^\top S \right\|_{\text{op}}^2} \left\| \Gamma^{-1} V^{-1} \Gamma^{-\top} \right\|_{\text{op}} \left\| V^{-1/2} Z_{T_1}^\top S \right\|_{\text{op}}^2 \left\| V^{-1/2} \Gamma^{-\top} x \right\|_2^2.$$

By Lemma G.4, with probability $1 - \delta$, we have

$$\left\| V^{-1/2} Z_{T_1}^\top S \right\|_{\text{op}}^2 \leq \sigma_\eta^2 \overline{\log}(Z_{T_1}, \delta).$$

Thus, with probability $1 - \delta$, we have

$$\mathfrak{V} \leq \frac{1}{1 - \left\| \Gamma^{-1} V^{-1} \Gamma^{-\top} \right\|_{\text{op}} \sigma_\eta^2 \overline{\log}(Z_{T_1}, \delta)} \left\| \Gamma^{-1} V^{-1} \Gamma^{-\top} \right\|_{\text{op}} \sigma_\eta^2 \overline{\log}(Z_{T_1}, \delta) \left\| V^{-1/2} \Gamma^{-\top} x \right\|_2^2.$$

To further upper bound $\mathfrak{V}$, we first find a sufficient condition on $T_1$ such that $\left\|\Gamma^{-1}V^{-1}\Gamma^{-\top}\right\|_{\text{op}}\sigma_\eta^2\overline{\log}(Z_{T_1},\delta) \leq \frac{1}{2g}$, $g \geq 1$. By Lemma J.14, when $T_1 \geq 2p$,

$$\left\|\Gamma^{-1}V^{-1}\Gamma^{-\top}\right\|_{\text{op}}\sigma_\eta^2\overline{\log}(Z_{T_1},\delta) \leq \frac{2\sigma_\eta^2}{T_1\sigma_{\min}\left(\sum_{z\in\mathcal{Z}}\lambda_z\Gamma^\top zz^\top\Gamma\right)}\overline{\log}(Z_{T_1},\delta).$$

To upper bound the right hand side by $\frac{1}{2g}$, we need

$$T_1 \geq \frac{4g\sigma_\eta^2}{\sigma_{\min}\left(\sum_{z\in\mathcal{Z}}\lambda_z\Gamma^\top zz^\top\Gamma\right)}\overline{\log}(Z_{T_1},\delta). \tag{58}$$

With this condition (58), we have with probability $1-\delta$,

$$\begin{aligned}
\mathfrak{V} \leq& \frac{1}{1-\left\|\Gamma^{-1}V^{-1}\Gamma^{-\top}\right\|_{\text{op}}\sigma_\eta^2\overline{\log}(Z_{T_1},\delta)}\left\|\Gamma^{-1}V^{-1}\Gamma^{-\top}\right\|_{\text{op}}\sigma_\eta^2\overline{\log}(Z_{T_1},\delta)\left\|V^{-1/2}\Gamma^{-\top}x\right\|_2^2\\
\leq& \frac{1}{1-\frac{1}{2g}}\frac{1}{2g}\left\|V^{-1/2}\Gamma^{-\top}x\right\|_2^2\\
\leq& \frac{1}{g}\left\|V^{-1/2}\Gamma^{-\top}x\right\|_2^2. \tag{59}
\end{aligned}$$

$\square$

**Lemma J.11.** *Suppose that we have $\widehat{\Gamma}$ that is an OLS estimate from an offline dataset $\{Z_{T_1}, X_{T_1}\}$ collected non-adaptively through a fixed design $\lambda$ and the efficient rounding procedure ROUND. Let $\dot{V}$ be any positive definite matrix. Then, for any $x \in \mathbb{R}^d$, we have, with probability $1-\delta$,*

$$\left\|(\Gamma^{-\top}-\widehat{\Gamma}^{-\top})x\right\|_{\dot{V}^{-1}}^2 \leq 2\left\|\Gamma^{-1}\dot{V}^{-1}\Gamma^{-\top}\right\|_{\text{op}}\sigma_\eta^2\overline{\log}(Z_{T_1},\delta)\|x\|_{\bar{A}(Z_{T_1},\Gamma)^{-1}}^2,$$

*when*

$$T_1 \geq \frac{4\sigma_\eta^2}{\sigma_{\min}\left(A(\lambda,\Gamma)\right)}\overline{\log}(Z_{T_1},\delta) \vee 2p,$$

*where $p$ is the cardinality of support of $\lambda$.*

*Proof.*

$$\begin{aligned}
&\left\|(\Gamma^{-\top}-\widehat{\Gamma}^{-\top})x\right\|_{\dot{V}^{-1}}^2\\
=&\left\|\Gamma^{-\top}S^\top Z_{T_1}V^{-1}(\Gamma+V^{-1}Z_{T_1}^\top S)^{-\top}x\right\|_{\dot{V}^{-1}}^2 & \text{(Lemma J.12)}\\
\overset{a_1}{=}&\left\|\dot{V}^{-\frac{1}{2}}\Gamma^{-\top}S^\top Z_{T_1}V^{-1}(\Gamma+V^{-1}Z_{T_1}^\top S)^{-\top}x\right\|_2^2 & (\|Ax\|_V^2 = \left\|V^{\frac{1}{2}}Ax\right\|_2^2)\\
\leq&\left\|\dot{V}^{-\frac{1}{2}}\Gamma^{-\top}S^\top Z_{T_1}V^{-\frac{1}{2}}\right\|_{\text{op}}^2\left\|V^{-\frac{1}{2}}(\Gamma+V^{-1}Z_{T_1}^\top S)^{-\top}x\right\|_2^2 & (\|Ax\| \leq \|A\|_{\text{op}}\|x\|)\\
\leq&\left\|\dot{V}^{-\frac{1}{2}}\Gamma^{-\top}\right\|_{\text{op}}^2\left\|S^\top Z_{T_1}V^{-\frac{1}{2}}\right\|_{\text{op}}^2\left\|V^{-\frac{1}{2}}(\Gamma+V^{-1}Z_{T_1}^\top S)^{-\top}x\right\|_2^2 & (\|AB\|_{\text{op}} \leq \|A\|_{\text{op}}\|B\|_{\text{op}})\\
=&\left\|\Gamma^{-1}\dot{V}^{-1}\Gamma^{-\top}\right\|_{\text{op}}\left\|V^{-1/2}Z_{T_1}^\top S\right\|_{\text{op}}^2\left\|V^{-1/2}\left(\Gamma+V^{-1}Z_{T_1}^\top S\right)^{-\top}x\right\|_2^2 & (\|A^\top A\|_{\text{op}} = \|A\|_{\text{op}}^2)\\
=&\left\|\Gamma^{-1}\dot{V}^{-1}\Gamma^{-\top}\right\|_{\text{op}}\left\|V^{-1/2}Z_{T_1}^\top S\right\|_{\text{op}}^2\left\|V^{-1/2}\Gamma^{-\top}x-V^{-1/2}\Gamma^{-\top}\left(V^{-1}Z_{T_1}^\top S\right)^\top\left(\Gamma+V^{-1}Z_{T_1}^\top S\right)^{-\top}x\right\|_2^2\\
&\text{(Lemma J.13: }(A+B)^{-1} = A^{-1}-(A+B)^{-1}BA^{-1})\\
\overset{a_2}{\leq}&\left\|\Gamma^{-1}\dot{V}^{-1}\Gamma^{-\top}\right\|_{\text{op}}\left\|V^{-1/2}Z_{T_1}^\top S\right\|_{\text{op}}^2\left(\left\|V^{-1/2}\Gamma^{-\top}x\right\|_2^2+\left\|V^{-1/2}\Gamma^{-\top}\left(V^{-1}Z_{T_1}^\top S\right)^\top\left(\Gamma+V^{-1}Z_{T_1}^\top S\right)^{-\top}x\right\|_2^2\right).
\end{aligned}$$

Given the condition (58) holds, we have with probability $1 - \delta$,

$$\left\|(\Gamma^{-\top} - \widehat{\Gamma}^{-\top})x\right\|^2_{\dot{V}^{-1}}$$

$$\overset{a_1}{\leq} \left\|\Gamma^{-1}\dot{V}^{-1}\Gamma^{-\top}\right\|_{\text{op}} \left\|V^{-1/2}Z_{T_1}^\top S\right\|^2_{\text{op}} \left(\left\|V^{-1/2}\Gamma^{-\top}x\right\|^2_2 + \left\|V^{-1/2}\Gamma^{-\top}x\right\|^2_2\right)$$

$$= 2\left\|\Gamma^{-1}\dot{V}^{-1}\Gamma^{-\top}\right\|_{\text{op}} \left\|V^{-1/2}Z_{T_1}^\top S\right\|^2_{\text{op}} \left\|V^{-1/2}\Gamma^{-\top}x\right\|^2_2$$

$$\leq 2\left\|\Gamma^{-1}\dot{V}^{-1}\Gamma^{-\top}\right\|_{\text{op}} \sigma_\eta^2 \overline{\log}(Z_{T_1}, \delta)\|x\|^2_{(\Gamma^\top V\Gamma)^{-1}},$$

where $(a_1)$ is due to (59) and setting $g = 1$. $\qquad\square$

**Lemma J.12.** *For a least square estimate $\widehat{\Gamma}$ that is estimated through a design matrix $Z$ and $\widehat{\Gamma}$ is invertible, we have*

$$\widehat{\Gamma}^{-1} - \Gamma^{-1} = -\left(\Gamma + V^{-1}Z^\top S\right)^{-1} V^{-1}Z^\top S\Gamma^{-1}.$$

*Proof.* Since $\widehat{\Gamma}$ is a least square estimator, we have

$$\widehat{\Gamma} = \Gamma + V^{-1}Z^\top S.$$

By Lemma J.13, we have

$$\widehat{\Gamma}^{-1} - \Gamma^{-1} = \left(\Gamma + V^{-1}Z^\top S\right)^{-1} - \Gamma^{-1}$$

$$= -\left(\Gamma + V^{-1}Z^\top S\right)^{-1} V^{-1}Z^\top S\Gamma^{-1}.$$

$\qquad\square$

**Lemma J.13.** *For two invertible matrixes $A, B \in \mathbb{R}^{d\times d}$, we have*
$$(A + B)^{-1} = A^{-1} - (A + B)^{-1}BA^{-1}.$$

*Proof.* We have

$$(A + B)^{-1} = A^{-1} + (A + B)^{-1} - A^{-1}$$

$$= A^{-1} + \left((A + B)^{-1}A - I\right)A^{-1}$$

$$= A^{-1} + (A + B)^{-1}\left(A - (A + B)\right)A^{-1}$$

$$= A^{-1} - (A + B)^{-1}BA^{-1}.$$

$\qquad\square$

**Lemma J.14.** *Suppose that we have a design matrix $Z_T$ that is sampled from a distribution $\lambda \in \Delta_{\mathcal{Z}}$, with the efficient rounding procedure ROUND. Let $p$ represent the cardinality of support of $\lambda$. We have, if $T \geq 2p$,*

$$\left\|\left(\Gamma^\top Z_T^\top Z_T \Gamma\right)^{-1}\right\|_{\text{op}} \leq \frac{2}{T\sigma_{\min}\left(\sum_{z\in\mathcal{Z}}\lambda_z\Gamma^\top zz^\top\Gamma\right)}.$$

*where $\sigma_{\min}(\cdot)$ is the smallest singular value of a matrix.*

*Proof.* Suppose that each arm $z \in \mathcal{Z}$ is sampled $t_z$ times, the empirical distribution of $Z_T$ is $\xi := \left(\frac{t_z}{T}\right)_{z\in\mathcal{Z}}$. Thus, we have

$$\left\|\left(\Gamma^\top Z_T^\top Z_T \Gamma\right)^{-1}\right\|_{\text{op}} = \frac{1}{\sigma_{\min}\left(\Gamma^\top Z_T^\top Z_T \Gamma\right)}$$

$$= \frac{1}{\sigma_{\min}\left(T\sum_{z\in\mathcal{Z}}\xi_z\Gamma^\top zz^\top\Gamma\right)}$$

$$= \frac{1}{T\sigma_{\min}\left(\sum_{z\in\mathcal{Z}}\xi_z\Gamma^\top zz^\top\Gamma\right)}.$$

By Fiez et al. [15, Proposition 2], we have

$$\sum_{z \in \mathcal{Z}} \xi_z \Gamma^\top z z^\top \Gamma \succeq \alpha \sum_{z \in \mathcal{Z}} \lambda_z \Gamma^\top z z^\top \Gamma,$$

where $\alpha \in [\frac{T}{T+2p}, 1]$ when $T \geq 2p$. Given the fact that both of $\sum_{z \in \mathcal{Z}} \xi_z \Gamma^\top z z^\top \Gamma$ and $\sum_{z \in \mathcal{Z}} \lambda_z \Gamma^\top z z^\top \Gamma$ are positive definite, we have $\sigma_{\min}\left(\sum_{z \in \mathcal{Z}} \xi_z \Gamma^\top z z^\top \Gamma\right) \geq \alpha \sigma_{\min}\left(\sum_{z \in \mathcal{Z}} \lambda_z \Gamma^\top z z^\top \Gamma\right)$. Thus, we have

$$\left\| \left( \Gamma^\top Z_T^\top Z_T \Gamma \right)^{-1} \right\|_{\mathrm{op}} \leq \frac{1}{\alpha T \sigma_{\min}\left( \sum_{z \in \mathcal{Z}} \lambda_z \Gamma^\top z z^\top \Gamma \right)}.$$

When $T \geq 2p$, we have $\alpha \geq 1/2$, which implies the result. $\qquad\square$

# K    Estimating $\lambda_{\min}(\Gamma)$

In this section, we introduce a simple adaptive procedure that finds a high probability lower bound on $\gamma^*_{\min} := \lambda_{\min}(\Gamma)$ that is sufficiently accurate (i.e., within a constant factor of $\gamma^*_{\min}$). For simplicity, we assume $\|z_t\| \leq 1, \forall t$ in this section.

Our algorithm leverages confidence bounds to adaptively determine how many samples we like to take. Let $\hat{\Gamma}_t := (Z^\top Z)^{-1} Z^\top X$ be the least square estimate of $\Gamma$ after sampling $t$ times to obtain $\{(z_s, x_s)\}_{s=1}^t$ where $Z \in \mathbb{R}^{t \times d}$ and $X \in \mathbb{R}^{t \times d}$ are the design matrices. Let $V_t := \sum_{s=1}^t z_s z_s^\top$. We define the lower and upper confidence bound for $\gamma^*_{\min}$ as follows:

$$\mathrm{LCB}(t) := \lambda_{\min}(\hat{\Gamma}_t) - \sqrt{\frac{\psi_t}{t}}, \quad \mathrm{UCB}(t) := \lambda_{\min}(\hat{\Gamma}_t) + \sqrt{\frac{\psi_t}{t}}$$

where

$$\psi_t = \sigma_{\min}^{-1}\left(\frac{1}{t} V_t\right) \cdot \left( 8d \ln\left( 1 + \frac{2t}{d(2 \wedge \sigma_{\min}(V_t))} \right) + 16 \ln\left( \frac{2 \cdot 6^d}{\delta} \cdot \log_2^2\left( \frac{4}{2 \wedge \sigma_{\min}(V_t)} \right) \right) \right).$$

and $\mathrm{LCB}(0) := -\infty$ and $\mathrm{UCB}(0) := \infty$.

The following lemma shows that $\mathrm{LCB}(t)$ and $\mathrm{UCB}(t)$ form a valid anytime confidence bound for $\gamma^*_{\min}$.

**Lemma K.1.** *(Correctness of the confidence bounds)*

$$1 - \delta \leq \mathbb{P}(\forall t \geq 1, \mathrm{LCB}(t) \leq \gamma^*_{\min} \leq \mathrm{UCB}(t))$$

Equipped with the confidence bounds, we are now ready to describe our algorithm for learning $\gamma^*_{\min}$ (see Algorithm 10). Since the tightness of the confidence bounds depends on the smallest eigenvalue of $V_t$, it is natural to use the E-optimal design as defined in Section 3.2. Recall that the solution of the E-optimal design is $\lambda_E^*$ and $\kappa_0$ is the smallest singular value achieved by $\lambda_E^*$. We take in a rounding procedure for the E-optimal design $\mathrm{ROUND}_E(\lambda, t)$ that takes in $t$ samples and design $\lambda$ and outputs integer sample count assignments $\{N_z\}_{z \in \mathcal{Z}}$ so that if we sample according to these counts then we have

$$\sigma_{\min}^{-1}(\frac{1}{t} V_t) \leq (1 + \omega)\kappa_0^{-1} \tag{60}$$

After determining the base sample counts $\{m_z\}_{z \in \mathcal{Z}}$ by $\mathrm{ROUND}(\lambda_E^*, \lceil r(\omega) \rceil, \omega)$, we start doubling the sample size until we satisfy the condition $\mathrm{LCB}(t) > \frac{1}{2}\mathrm{UCB}(t)$. Note that the sampling scheme in the while loop is designed such that the total number of samples collected up to (and including) $j$-th iteration is $2^{j-1}\lceil r(\omega) \rceil$. Once the loop stops, we return $\mathrm{LCB}(t)$ as the claimed lower approximation of the $\gamma^*_{\min}$.

Let $N_w$ be the total number of samples we used in Algorithm 10. Then, the next theorem shows that the estimate returned by our algorithm is both a valid lower bound to $\gamma^*_{\min}$ and sufficiently accurate.

---

**Algorithm 10** Learning $\lambda_{\min}(\Gamma)$

---

**Input:** Arm set $\mathcal{Z}$, rounding procedure $\text{ROUND}_E$ for the E-optimal design, rounding accuracy $\omega$
**Initialize:** $j = 1, t = 0$.
Compute the E-optimal design $\lambda_E^*$ for $\mathcal{Z}$.
Compute $\{m_z\}_{z\in\mathcal{Z}}$ by $\text{ROUND}_E(\lambda_E^*, \lceil r(\omega) \rceil)$.
**while** $\text{LCB}(t) \leq \frac{1}{2}\text{UCB}(t)$ **do**
  $t \leftarrow 2^{j-1}\lceil r(\omega) \rceil$.
  For each arm $z \in \mathcal{Z}$, sample $2^{j-1}m_z - \mathbb{1}\{j \neq 1\}\, 2^{j-2}m_z$ times.
  Estimate $\hat{\Gamma}_t$ using all samples collected so far (total $t$ data points)
  $j \leftarrow j + 1$.
**end while**
**Output**: $\text{LCB}(t)$

---

**Theorem K.2.** *(Correctness of Algorithm 10) The total number of samples denoted by $N_w$ used in Algorithm 10 satisfies that, with probability at least $1 - \delta$,*

$$\frac{\gamma_{\min}^*}{2} < \text{LCB}(N_w) \leq \gamma_{\min}^*$$

We next analyze the sample complexity of the algorithm, which essentially shows the scaling of $\frac{1}{(\gamma_{\min}^*)^2\kappa_0}$ even if the algorithm does not need knowledge of $\gamma_{\min}^*$.

**Theorem K.3.** *(Sample complexity of Algorithm 10) Then, with $\omega = 1$, we have, with probability at least $1 - \delta$,*

$$N_w = O\left(r(1) + (\gamma_{\min}^*)^{-2}\kappa_0^{-1}\cdot\left(d\cdot\text{polylog}((\gamma_{\min}^*)^{-2}, \kappa_0^{-1}, d) + \ln(2/\delta)\right)\right)$$

We remark that Allen-Zhu et al. [2] provides a rounding procedure with $r(\varepsilon) = O(d/\varepsilon^2)$.

***Proof of Lemma K.1.*** Note that $\hat{\Gamma} = \Gamma + V_t^{-1}Z^\top S$ where $Z, S \in \mathbb{R}^{t\times d}$ is the design matrices with $s$-th row being $z_s^\top$ and $\eta_s^\top$ respectively. Using Lemma G.4, we have, with probability at least $1 - \delta$,

$$\begin{aligned}
\forall t \geq 1, \|\hat{\Gamma}_t - \Gamma\|_{\text{op}}^2 &= \|V_t^{-1}Z^\top S\|_{\text{op}}^2 \\
&\leq \|V_t^{-1/2}\|_{\text{op}}^2 \|V_t^{-1/2}Z^\top S\|_{\text{op}}^2 \\
&= \frac{1}{t\sigma_{\min}(\frac{1}{t}V_t)}\|V_t^{-1/2}Z^\top S\|_{\text{op}}^2 \\
&\leq \frac{1}{t\sigma_{\min}(\frac{1}{t}V_t)}\cdot\left(8d\ln\left(1 + \frac{2t}{d(2\wedge\sigma_{\min}(V_t))}\right) + 16\ln\left(\frac{2\cdot 6^d}{\delta}\cdot\log_2^2\left(\frac{4}{2\wedge\sigma_{\min}(V_t)}\right)\right)\right) \\
&\qquad\qquad (\text{Lemma G.4}) \\
&= \frac{1}{t}\psi_t
\end{aligned}$$

The well-known Weyl's theorem implies that $\max_k |\lambda_k(\hat{\Gamma}_t) - \lambda_k(\Gamma)| \leq \|\hat{\Gamma}_t - \Gamma\|_{\text{op}}$ where $\lambda_k(A)$ is the $k$-th largest singular value of the matrix $A$. Choosing $k = d$ and combining it with the display above conclude the proof. $\square$

***Proof of Theorem K.2.*** We assume $\forall t \geq 1, \text{LCB}(t) \leq \gamma_{\min}^* \leq \text{UCB}(t)$, which happens with probability at least $1 - \delta$. Then, it is trivial to see that $\text{LCB}(N_w) \leq \gamma_{\min}^*$.

For the other inequality, we use the fact that the stopping condition was satisfied with $N_w$:

$$\text{LCB}(N_w) > \frac{1}{2}\text{UCB}(N_w) \geq \frac{\gamma_{\min}^*}{2}\ .$$

This concludes the proof. $\square$

***Proof of Theorem K.3.*** We assume $\forall t \geq 1, \mathrm{LCB}(t) \leq \lambda_{\min}(\Gamma) \leq \mathrm{UCB}(t)$, which happens with probability at least $1 - \delta$. In this proof, we let $\hat{\gamma}_{\min} := \lambda_{\min}(\hat{\Gamma}_{N_w})$ and $\gamma^*_{\min} := \lambda_{\min}(\Gamma)$ for brevity.

If $N_w = \lceil r(1) \rceil$, then there is nothing to prove. Otherwise, the loop in the algorithm was iterated more than once. Then, since the stopping condition was satisfied with $N_w$, we have that in the previous iteration where $t = N_w/2$ the stopping condition was not satisfied. Thus,

$$2\mathrm{LCB}(N_w/2) \leq \mathrm{UCB}(N_w/2) \ .$$

Using the following two inequalities:

$$2\mathrm{LCB}(N_w/2) \geq 2\left(\hat{\gamma}_{\min} - \sqrt{\frac{\psi_{N_w/2}}{N_w/2}}\right) \geq 2\left(\gamma^*_{\min} - 2\sqrt{\frac{\psi_{N_w/2}}{N_w/2}}\right)$$

$$\mathrm{UCB}(N_w/2) \leq \hat{\gamma}_{\min} + \sqrt{\frac{\psi_{N_w/2}}{N_w/2}} \leq \gamma^*_{\min} + 2\sqrt{\frac{\psi_{N_w/2}}{N_w/2}} \ ,$$

we have

$$\gamma^*_{\min} \leq 6\sqrt{\frac{\psi_{N_w/2}}{N_w/2}} \implies N_w \leq \frac{72}{(\gamma^*_{\min})^2}\psi_{N_w/2} \ .$$

On the other hand, with the rounding procedure, we have

$$\sigma^{-1}_{\min}\left(\sum_{t=1}^{N} z_t z_t^\top\right) = \frac{1}{N}\sigma^{-1}_{\min}\left(\frac{1}{N}\sum_{t=1}^{N} z_t z_t^\top\right) = \frac{1}{N}\sigma_{\max}\left(\left(\frac{1}{N}\sum_{t=1}^{N} z_t z_t^\top\right)^{-1}\right) \leq \frac{1}{N}(1+\omega)\kappa_0^{-1} = \frac{2}{N}\kappa_0^{-1}$$

since $\omega = 1$. Using this and the fact that $N_w \geq d$, it is easy to see that there exists an absolute constant $c_1$ such that

$$\psi_{N_w/2} \leq c_1 \kappa_0^{-1}\left(d\ln\left(1 + N_w + \kappa_0^{-1}\right) + \ln\left(\frac{2}{\delta}\right)\right) \ .$$

Then, there exists an absolute constant $c_2$ such that

$$N_w \leq (\gamma^*_{\min})^{-2} \cdot c_2 \kappa_0^{-1}\left(d\ln\left(1 + N_w + \kappa_0^{-1}\right) + \ln\left(\frac{2}{\delta}\right)\right)$$

We have $N_w$ on both sides. We invoke Lemma K.5 with $r = 1 + N_w$ to obtain

$$N_w \leq 1 + 2c_2(\gamma^*_{\min})^{-2}\kappa_0^{-1}\left(d\ln\left(1 + 2\kappa_0^{-1}(1 + c_2 d(\gamma^*_{\min})^{-2})\right) + \ln\left(\frac{2}{\delta}\right)\right) \ .$$

$\square$

**Lemma K.4.** *Let $A, \Gamma \in \mathbb{R}^{d \times d}$ where $A$ is symmetric positive semi-definite. Then,*

$$\sigma_{\min}\left(\Gamma^\top A \Gamma\right) \geq \sigma_{\min}(\Gamma)^2 \sigma_{\min}(A)$$

*Proof.*

$$
\begin{aligned}
(\sigma_{\min}(\Gamma^\top A\Gamma))^{-1} &= \|\Gamma^{-1}A^{-1}\Gamma^{-T}\|_{\mathrm{op}} \\
&\leq \|\Gamma^{-1}\|_{\mathrm{op}}\|A^{-1}\|_{\mathrm{op}}\|\Gamma^{-T}\|_{\mathrm{op}} \qquad \text{(submultiplicity of the operator norm)} \\
&= \sigma_{\min}(\Gamma)^{-2}\sigma_{\min}(A)^{-1}
\end{aligned}
$$

Taking the inverse on both sides concludes the proof. $\square$

## K.1 Proof of Lemma 3.2

Define $\Lambda = \sum_{i=1}^{d} \lambda_i e_i e_i^\top$, i.e. the diagonal matrix with $\lambda$ on the diagonal.

Note that

$$\min_\lambda \max_{i,j} (e_i^\top - e_j)^\top (\Gamma \Lambda \Gamma^\top)^{-1} (e_i - e_j) = \min_\lambda \max_{i,j} \sum_{k=1}^{d} \frac{(\Gamma_{k,i}^{-1} - \Gamma_{k,j}^{-1})^2}{\lambda_k}$$

So for an upper bound, we consider $\lambda = 1/d\mathbf{1}$,

$$\min_\lambda \max_{i,j} (e_i^\top - e_j)^\top (\Gamma \Lambda \Gamma^\top)^{-1} (e_i - e_j) \leq d \max_{i,j} \|\Gamma^{-1}(e_i - e_j)\|_2^2$$

The result about $\rho^*$ follows immediately.

When $\Gamma = (1-\varepsilon)/d\mathbf{1}\mathbf{1}^\top + \varepsilon I$, a computation using Sherman-Morrison shows that $\Gamma^{-1} = 1/\varepsilon[I - (1-\varepsilon)/d\mathbf{1}\mathbf{1}^\top]$. Thus

$$
\begin{aligned}
\min_\lambda \max_{i,j} (e_i^\top - e_j)^\top (\Gamma \Lambda \Gamma^\top)^{-1} (e_i - e_j) &= \varepsilon^{-2} \min_\lambda \max_{i,j} (e_i - e_j)^\top \Lambda^{-1}(e_i - e_j) \\
&= \varepsilon^{-2} \min_\lambda \max_{i,j} e_i\top\Lambda^{-1}e_i + e_j\Lambda^{-1}e_j \\
&= \varepsilon^{-2} \min_\lambda \max_{i,j} e_i\top\Lambda^{-1}e_i + e_j\Lambda^{-1}e_j \\
&\geq \varepsilon^{-2} \min_\lambda \max_i e_i\top\Lambda^{-1}e_i \qquad\qquad = \varepsilon^{-2}d
\end{aligned}
$$

where the last line follows from the Kiefer-Wolfowitz Theorem [26].

## K.2 lemma for solving x less than ln(x)

**Lemma K.5.** *Let $a, b, c, d > 0$. Then, for every $r$,*

$$r \leq a + b\ln(c + dr) \implies r \leq 2a + 2b\ln(1 + 2c + 2bd)$$

*Proof.* If $dr \leq c$, then

$$r \leq a + b\ln(2c) \leq a + b\ln(1 + 2c)$$

If $dr > c$, then,

$$
\begin{aligned}
r &\leq a + b\ln(2dr) \\
&= a + b\ln\left(2d\frac{r}{2b}\cdot 2b\right) \\
&\leq a + b\left(\frac{r}{2b} - 1 + \ln(4bd)\right) \qquad\qquad (\ln(x) \leq x - 1) \\
\implies r &\leq 2a + 2b\ln\left(\frac{4}{e}bd\right) \\
&\leq 2a + 2b\ln(2bd)
\end{aligned}
$$

Either case, we have

$$r \leq 2a + 2b\ln\big((1 + 2c) \vee 2bd\big) \leq 2a + 2b\ln(1 + 2c + 2bd)$$

$\square$

**Lemma K.6.** *Let $\alpha, \beta > 0$. Then, for any $r$,*

$$r \leq \alpha\ln(1 + r) + \beta \implies r \leq 2\alpha\ln(e + \alpha) + 2\beta$$

*Proof.*

$$r \leq \alpha \ln\left(r(\frac{1}{r} + 1)\right) + \beta$$

$$\leq \alpha \ln\left(\frac{r}{2\alpha} \cdot 2\alpha(\frac{1}{r} + 1)\right) + \beta$$

$$\leq \alpha \left(\frac{r}{2\alpha} - 1 + \ln\left(2\alpha(\frac{1}{r} + 1)\right)\right) + \beta \qquad\qquad (\forall x, \ln(x) \leq x - 1)$$

$$\implies r \leq 2\alpha \ln\left(\frac{2}{e}\alpha(\frac{1}{r} + 1)\right) + \beta$$

If $r \leq 2\alpha$, then there is nothing to prove. If $r > 2\alpha$, then $r \leq 2\alpha \ln(1 + \alpha) + 2\beta$. Either case, we have $r \leq 2\alpha \ln(e + \alpha) + 2\beta$. $\qquad\square$

