# OpenReview forum: "Adaptive Experimentation When You Can't Experiment"
_NeurIPS.cc/2024/Conference — NeurIPS 2024 poster_

### Official Review · Reviewer_byyL · 2024-07-11

**Soundness:** 3
**Presentation:** 2
**Contribution:** 3
**Rating:** 6
**Confidence:** 2

**Summary:**

This article studies the pure exploration transductive linear bandit problem in the presence of instrumental variables. The authors assume a linear structural equation model on the instrument, treatment and outcome. The proposed method attempts to estimate the parameters in the structural model while optimally designing to learn the best arm (i.e. treatment).  The paper is mainly theoretical, but some simulations are provided to demonstrate the method.

**Strengths:**

1. The combination of instrument variables and pure exploration appears novel.
2. The proposed method is guided by finite-time confidence bounds for two-stage least squares. The lower and upper bounds of sample complexity  are provided in the paper.
3. The method outperforms the standard methods UCB-OLS and UCB-IV.

**Weaknesses:**

1. Section 3 is very dense. The authors can consider shortening the content before Section 2.2 and use more room to explain Section 3.
2. No real-data demonstration of the proposed method.

**Questions:**

N.A.

**Limitations:**

N.A.

---

> ### Author Rebuttal · Authors · 2024-08-06
>
> We thank the reviewer for acknowledging the novelty of our problem setting and ideas, as well as our theoretical contributions.
>
> Regarding your comments:
>
> 1. We agree that Sec 3 is dense, making this paper a heavy lift with many theoretical results. We appreciate the suggestion to shorten the content. The camera-ready version allows an additional page, giving us the opportunity to make these improvements.
>
> 2. We would have loved to examine our approach on real data and compare it against alternative methods. Unfortunately, to the best of our knowledge, there is no public dataset available for this specific problem.

---

### Official Review · Reviewer_4vBb · 2024-07-11

**Soundness:** 3
**Presentation:** 2
**Contribution:** 3
**Rating:** 6
**Confidence:** 4

**Summary:**

The paper introduces the confounded pure exploration transductive linear bandit (CPET-LB) problem, which addresses the challenges of conducting adaptive experimentation in environments where direct randomization is not possible. From my understanding, the paper studies the best arm identification problem under linear structural model with non-compliance. The paper proposes the algorithms with nearly optimal sample complexity guarantee.

**Strengths:**

1.	The paper provides a thorough theoretical analysis, including proofs of finite-time confidence intervals for the estimators and sample complexity bounds. The theoretical contributions are significant and appear to be solid.
2.	The problem is practical-relevant and important.

**Weaknesses:**

1.	The connections with two streams of the literature should be more clearly stated. The basic problem set up is very classical econometric setting where the non-compliance exists. The analysis framework and some of the tools are very standard in transductive linear bandit problem.
2.	The presentation of Section 1 is not easy to follow. I feel the authors try to manage the terminology from causal inference and pure exploration. For example, in Line 78, “measurement” and ”evaluation” are new terminology of the paper and a bit hard to connect with the example proposed in introduction.
3.	The authors might want to reconsider the title. The title, from my perspective, is a bit confusing and misleading.

**Questions:**

1.	Could you please elaborate a bit more on why the confidence interval in Section 2.2 is novel? I know the traditional 2SLS always using asymptotic normality to construct CI. Is non-asymptotic or asymptotic the key difference?

**Limitations:**

See above.

---

> ### Author Rebuttal · Authors · 2024-08-06
>
> We thank the reviewer for acknowledging our thorough theoretical analysis, noting its significance and solid foundation, as well as the strong practical value of our work.
>
> Regarding your comments on weaknesses, we appreciate your detailed concerns:
>
> 1. As the reviewers kindly pointed out, we did take on the challenge of connecting fairly disparate settings in the literature. We addressed these aspects in the introduction, page 3 footnote, and the related work section of the appendix. We will conduct another round of editing to more clearly delineate the two streams of topics.
>
> 2. Thank you for this very insightful comment." We will link the terminology back to the example proposed in the introduction, where $Z$ refers to  encouragement and $X$ refers to treatment. The terms "measurement" and "evaluation" come from the transductive linear bandit literature. We will clarify this further in the revision.
>
> Thanks for your question:
>
> Non-asymptotic confidence intervals are one of our key novelties. While asymptotic intervals for the 2SLS estimator are known, they are not useful for algorithm design because they do not allow for identifying and eliminating bad treatments in the non-asymptotic regime with a precise correctness guarantee.
>
> Our second novelty addresses the limitations of the only other known non-asymptotic interval, which requires simultaneous control over Zs and Xs, this is challenging as X is a random result of Z. Our confidence interval solves this by using data splitting in two phases. This discussion is included in the paper.

---

> > ### Comment · Reviewer_4vBb · 2024-08-11
> >
> > I have read the rebuttal carefully. Thanks for the clarification. I really appreciate your efforts!

---

### Official Review · Reviewer_oap7 · 2024-07-13

**Soundness:** 4
**Presentation:** 3
**Contribution:** 3
**Rating:** 7
**Confidence:** 4

**Summary:**

This paper addresses the issue of adaptive experimentation using "encouragement" rather than "compulsion instruction," a scenario commonly encountered in industrial applications. The proposed solution integrates linear bandit algorithms with instrumental variables regressions. The authors provide rigorous theoretical guarantees for their method and demonstrate its superior performance compared to traditional A/B testing and conventional linear bandit algorithms.

**Strengths:**

The scenarios examined by the authors are well-motivated. In practice, numerous situations exist where only encouragement can be employed to influence user decisions. Considering the shift in industry from traditional A/B testing to adaptive experimentation, the study presented here has the potential for significant industry impact.

In addition, the proposed algorithm is intuitive, and the theoretical guarantees provided are robust.

**Weaknesses:**

The authors could enhance their study by conducting additional experiments to verify the robustness of their results.

Furthermore, there should be a more detailed discussion of the p-values and confidence intervals to strengthen the statistical analysis.

**Questions:**

How to think of p-values/confidence intervals as it is important in an experimentation setting.

**Limitations:**

The authors adequately addressed the limitations.

---

> ### Author Rebuttal · Authors · 2024-08-06
>
> We are happy to see the reviewer likes our work and find its strong practical value.
>
> Regarding your comment on additional experiments:
>
> Our work is primarily theoretical and to provide an initial solution, which makes the surprising connection between encouragement designs and pure exploration linear bandits. This highlights an area that the adaptive experimentation/bandits community has largely overlooked. We have included experiments in the Appendix to demonstrate the impact of confounding and the effectiveness of our solution. We acknowledge that more experiments can be conducted and hope our work inspires further study in this area. We will update our conclusion to reflect this.
>
> Regarding the question on p-values and confidence intervals in an experimentation setting:
>
> they correspond 1-to-1. For example, we can construct a confidence interval by considering the set of $\theta$ that are not rejected under the null hypothesis for a given $p$-value. A good reference is Johari, Ramesh, et al. "Always valid inference: Continuous monitoring of a/b tests." Operations Research 70.3 (2022): 1806-1821. Thus our confidence intervals in Section 2 could be used to provide inference. We will add a small discussion of this correspondence to the final draft.

---

> > ### Comment · Reviewer_oap7 · 2024-08-12
> >
> > Thank you for addressing my questions on the confidence interval. I will keep my score unchanged.

---

### Official Review · Reviewer_Y9pp · 2024-07-16

**Soundness:** 3
**Presentation:** 3
**Contribution:** 3
**Rating:** 5
**Confidence:** 3

**Summary:**

This paper addresses the problem of pure exploration bandits in the setting of encouragement designs. The authors describe the problem in terms of online instrumental variable regression. Toward this end, the authors derive a finite time confidence interval. Using this as the main tool, the authors then describe a pure exploration transductive bandit. Algorithms are provided for optimizing the pure exploration problem. A number of theoretical results are provided describing the entailed sample complexity of the proposed approach. Empirical results are provided which show strong performance against other baselines.

**Strengths:**

This is a problem that has practical relevance in both industrial and social scientific settings. The task, to my knowledge, is novel in its formulation, and the authors do a nice job of delineating this work from prior art. The authors do a nice job of presenting algorithms and analysis in both the settings of a known and unknown structural models. Further, there is thorough analysis of each of the proposed procedures' properties.

**Weaknesses:**

My concerns are largely around two things:
1. There is a fairly limited experimental evaluation here. It would be helpful if the authors provided a more thorough evaluation of the proposed approaches' behavior across a wider range of settings.

2. The text is a little meandering at times and as a result was a little hard to follow on first read. I would suggest a round of editing in order to improve the narrative and organization of the paper.

**Questions:**

Throughout an E-optimal design is also employed. My main question is whether the choice of E-optimality is done as a matter of convenience, or if it is motivated by aspects of the problem that would favor this criterion over other choices.

**Limitations:**

See above.

---

> ### Author Rebuttal · Authors · 2024-08-06
>
> We thank the reviewer for acknowledging the novelty of our problem formulation and the thorough analysis throughout.
>
> Concern on limited experimental evaluation:
>
> Our work is primarily theoretical and to provide an initial solution, which makes the surprising connection between encouragement designs and pure exploration linear bandits. This highlights an area that the adaptive experimentation / bandits community has largely overlooked. We have included experiments in the Appendix to demonstrate the impact of confounding and the effectiveness of our solution. We acknowledge that more experiments can be conducted and hope our work inspires further study in this area. We will update our conclusion to reflect this.
>
> Concern on readability of results:
>
> We will undertake additional rounds of editing and polishing to enhance readability. The camera-ready version will allow us to add one more page, which will help us improve clarity.
>
> Regarding the question:
>
> Thank you for your question. We chose the E-optimal design to ensure the covariance matrix of the collected data is well-conditioned by maximizing the minimal eigenvalue, a requirement not necessarily met by other objectives.
> This well-conditioning is crucial for efficiently meeting the stopping condition of the $\Gamma$ estimation phase.
> That said, we believe E-optimal design is sufficient but may not be necessary -- perhaps a more sophisticated design could be more useful, which remains to be a future direction.
> A significant algorithmic analysis contribution is determining the number of samples for E-optimal design to maintain these properties without significantly increasing sample complexity.

---

> > ### Author Response · Authors · 2024-08-13
> >
> > Dear reviewer Y9pp,
> >
> > As we approach the end of the rebuttal period, we sincerely appreciate this last opportunity to further engage with the reviewer and clarify any outstanding questions or concerns. We thank the reviewer again for acknowledging the novelty of our problem formulation and the thorough analysis throughout. If our rebuttal has adequately addressed the points raised, we would kindly request that the reviewer re-evaluate the score accordingly. We remain open to any additional feedback.

---

> > ### Comment · Reviewer_Y9pp · 2024-08-14
> >
> > Thank you for your answers and clarifications to my questions/concerns. I will leave my score unchanged, but appreciate the points raised by the authors.

---

### Author Rebuttal · Authors · 2024-08-07

We thank the reviewers for providing insightful comments. As the strengths of our paper,
reviewers Y9pp, oap7, and 4vBb have acknowledged the practical relevance and importance of the problem we introduced, and reviewers Y9pp and byyL acknowledged that our formulation is novel.
Finally, reviewer Y9pp, oap7, and 4vBb have mentioned that our analysis is through and solid.

---

### Decision · Program_Chairs · 2024-09-25

**Decision:**

Accept (poster)

**Comment:**

I concur with the reviewers' positive view of the submission and think the paper would makes a nice addition to the nascent literature on this topic. However, I find deferring all discussion of related work to the appendix to be very inappropriate. Remarking that there are "only a handful of works in the multi-armed bandit literature that even consider confounding" is not informative of the works and how their technical content relates to the present paper -- the number of works is not relevant. I therefore ask the authors to discuss the relevant related works in the main text and to even prioritize this over technical details or considering different settings (eg Sec 3.1 and 3.2). It is ok to have an even more extended discussion of tangentially related work in the appendix, but directly related works studying the same problem such as [5,21,24,11,36,17] must be discussed in full to properly set up the context for the work and its contribution, especially as there are close similarities such as remarked on line 497 ("We remark that their approach is similar to ours ..."). I trust that the authors can undertake this appropriately within the preparation of a camera ready version as this mostly involves moving content around from/to the appendix.